# REGULARIZATION VIA INVARIANT PATTERNS: TEMPORAL DOMAIN RANDOMIZATION FOR HUMAN ACTIVITY RECOGNITION

## ABSTRACT

Synthetic data has become a common strategy to address data scarcity in Human Activity Recognition (HAR). However, models trained on synthetic samples often overfit to spurious features, leading to a substantial domain gap when transferred to real-world data. To address this challenge, we propose Regularization via Invariant Patterns (RIP), a novel data-centric method that extends domain randomization to the temporal domain. RIP augments time-series windows by "framing" them with invariant (constant-valued) patterns, compelling models to focus on informative signals rather than irrelevant temporal context. Evaluated across five HAR datasets, four classifiers, and more than 2,000 experiments, RIP consistently improves F1 scores, achieving gains of up to +53 percentage points (over +160% relative improvement) compared to synthetic baselines — often matching or surpassing real-data baselines. Beyond synthetic scenarios, RIP also boosts performance in real-only training settings, highlighting its broad applicability. Both theoretical analysis and empirical results show that RIP stabilizes weight updates and enhances calibration, all without modifying model architectures.

## 1 INTRODUCTION

Human Activity Recognition (HAR) increasingly uses synthetic samples to mitigate data scarcity, privacy constraints, and inter-subject variability, especially with wearable time-series data. However, models trained on synthetic data often rely on generator-specific cues, creating a significant synthetic-to-real gap at inference time Seib et al. (2020); Sankaranarayanan et al. (2018). Prior work indicates that the use of synthetic samples is most effective when combined with principled, data-centric regularization rather than used as a mere data augmentation strategy Souza et al. (2023); Lupión et al. (2024).

**Background and Related Work.** (1) *Domain randomization (DR).* In computer vision, it improves simulated-to-real transfer by randomizing non-essential factors (e.g., backgrounds) Tremblay et al. (2018). A principled *temporal* analogue for wearable HAR remains underexplored. (2) *Time-series/HAR regularization.* Common data-centric approaches include jitter, scaling, time-warping, permutation/cropping, and *temporal masking/cutout* (SpecAugment-style), as well as Mixup/Cutmix and optimization-level methods like SAM/DRO Zhang et al. (2018); Yun et al. (2019); Foret et al. (2021); Kuhn et al. (2024); Bento et al. (2023). These techniques perturb local dynamics or alter the loss, but generally do *not* enforce invariance to non-informative temporal context. (3) *Domain generalization (DG) and calibration.* In the HAR domain, only a limited number of works explicitly address DG, such as Napoli & Borin (2025); Qin et al. (2023). Most prior research has focused on modalities like images and text, while applications to time-series data remain comparatively underexplored.

**Our idea.** We present *Regularization via Invariant Patterns (RIP)*, a simple, architecture-agnostic *temporal* DR mechanism: each training window is "framed" by *constant-valued* segments. RIP discourages reliance on spurious context and biases learning toward class-relevant dynamics by inducing class-agnostic invariance in the surrounding temporal context. Unlike zero/edge padding or random masking, RIP uses *structured* invariant patterns drawn from a small set of $\gamma$ values, explicitly operationalizing DR in time.

**Contributions.** (i) We introduce RIP, a data-centric regularizer that, to our knowledge, brings domain randomization to the temporal axis for HAR without modifying architectures or losses. (ii) Across five datasets, six classifiers, and more than 2,000 runs, RIP consistently improves Synthetic → Real transfer (average macro-F1 gains ≈+53 pp up to ≈+81 pp) and also boosts real-world performances, with tighter confidence intervals. (iii) We analyze why RIP stabilizes learning—reduced hidden-state variance and better probability calibration. (iv) We position RIP against time-series augmentations and DG baselines, and provide ablations on duplication factor $i$ and constant design $\gamma$. Extended related work appears in Appendix B.

## 2 REGULARIZATION VIA INVARIANT PATTERNS (RIP)

Regularization via Invariant Patterns (RIP) introduces a new form of data-centric regularization inspired by the domain randomization principle Tremblay et al. (2018). In their work, Tremblay et al. (2018) demonstrated that training on synthetic images with randomized backgrounds forces a model to become invariant to non-essential features, thus bridging the sim-to-real domain gap. We translate this core idea from the spatial domain of images to the temporal domain of HAR data. For a time-series window, we treat the surrounding temporal context as the "background." RIP implements this concept of temporal domain randomization by strategically "framing" the core signal with constant-valued windows. These invariant patterns, defined by a scalar $\gamma$, compel the model to focus on the dynamic, informative part of the signal, much like a picture frame draws attention to the image it contains. This process encourages learning more robust and generalizable representations from synthetic data (We discuss the DR challenge in Appendix A).

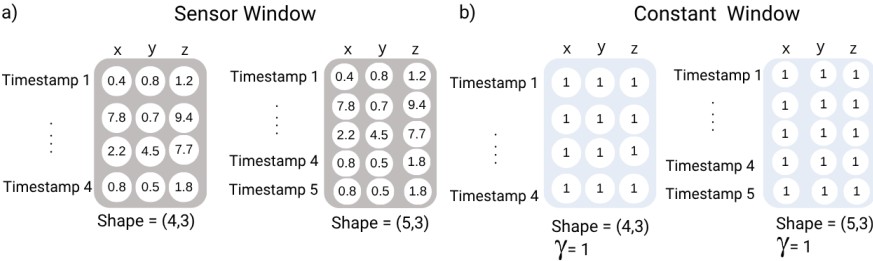

Figure 1: **Sensor vs. constant windows:** (a) Varying signals over time. (b) Fixed values across time and attributes are used for regularization. Each window has $\omega$ timestamps and 3 attributes.

**Preliminary Concepts.** Let $\mathcal{D}$ be a dataset of time-series samples, where each sample is a temporal window $x \in \mathbb{R}^{\omega \times 3}$. Here, $\omega \in \mathbb{N}$ is the window length, and the second dimension corresponds to the three sensor axes. We define two types of windows: a **sensor window**, containing dynamic sensor readings, and a **constant window**, where all values are fixed to a scalar $\gamma \in \mathbb{Z}$. This invariant pattern serves as the non-informative "temporal background" that regularizes the learning process. Figure 1 illustrates the structural difference between these two window types. The value of $\gamma$ is a hyperparameter subject to tuning.

**RIP Formalization.** Given a dataset $\mathcal{D} = \{(x_1, y_1), \ldots, (x_n, y_n)\}$, the RIP method produces an augmented dataset $\mathcal{D}'$. For each sensor window $x_i$, we generate a new sample $x_i'$ by creating a sequence of information where constant windows frame the original window. This operation is controlled by two hyperparameters: the constant value $\gamma$ and a duplication factor $i \in \mathbb{N}$. The number of constant windows prepended and appended to form the sequence is defined as $2i$. The final augmented sample $x_i'$ is a tensor of shape $(4i + 1, \omega, 3)$, where $\omega$ is the original window length. The corresponding label $y_i'$ is the original label repeated $4i + 1$ times. The whole procedure is formally described in Algorithm 1 and illustrated in Fig.2.

By explicitly introducing invariant information into the training data, RIP compels the model to learn from the contrast between constant and dynamic temporal patterns. This strategy reinforces the focus on meaningful signal patterns, reducing variance in the learned weights and enhancing the model's ability to generalize across both synthetic and real data distributions[1]

---

[1]The source code for the proposed method is publicly available at after_review_process.

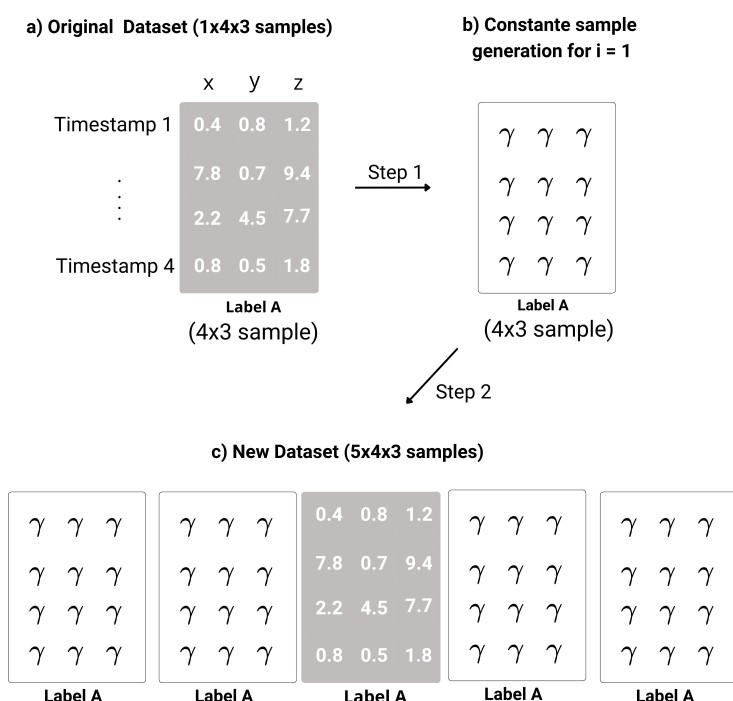

Figure 2: RIP dataset construction process. For each sample in the original dataset, we generate a constant sample using a distribution parameterized by $\gamma$ and a duplication factor $i$, ensuring that the generated sample has the same shape as the original input. This constant sample is assigned the same label as the original one. We then store the following ordered sequence in the dataset: **(1) constant sample, (2) constant sample, (3) real sample, (4) constant sample, (5) constant sample**. Our ablation studies show that this order is crucial, as it defines the real window within the constant windows, a key component of the RIP regularization mechanism. This process is applied to every sample in the dataset.

## 3 EXPERIMENTAL SETUP

To evaluate the effectiveness of our proposed RIP method, we conducted extensive experiments on both real and synthetically generated datasets. Our setup is designed to isolate the impact of RIP on synthetic data quality and assess its broader applicability.

**Data and Models** We used the Time-LogCosh-GAN (TLCGAN) Souza et al. (2023) to generate synthetic tri-axial accelerometer data for five publicly available HAR datasets: MHAD1 Chen et al. (2015), MHAD2 Chen et al. (2015), MHEALTH Banos et al. (2014), WISDM Weiss, and WHARF Bruno et al. (2013). For classification, we employed a diverse set of four models: Deep ConvLSTM (DClassifier) Singh et al. (2020), TS-Classifier hfawaz (2020), Time Series Random Forest (TSRF) for, Time Series Bag-of-Features (TSBF) Baydogan et al. (2013), RevTransformer Pramanik et al. (2023), and TimeTransformer Team (2020). Full details on datasets, preprocessing steps, and model implementations are provided in Appendix C.

**Evaluation Protocol** Our primary evaluation follows the Train on Synthetic, Test on Real (TSTR) protocol, which is well-suited to assess whether synthetic data, when enhanced by RIP, effectively improves generalization to real-world distributions. As a practical baseline, we also employ the conventional Train on Real, Test on Real (TRTR) protocol. In addition, to ensure robustness across subjects and to evaluate cross-user generalization, we include the standard Leave-One-Subject-Out (LOSO) protocol (See Appx D). This setting is particularly relevant for HAR, where inter-subject variability is high. Given the class imbalance inherent to HAR datasets, we report multiple metrics, with emphasis on the F1 score due to its robustness. Performance gains are expressed in percentage

Table 1: Performance comparison of RIP and the baseline under the Train on Synthetic, Test on Real (TSTR) protocol. For each dataset, the best-performing RIP configuration is reported. Best results per metric are shown in bold.

| Model | Dataset | RIP Config. | Accuracy (%) | | F1-score (%) | |
|---|---|---|---|---|---|---|
| | | $(\gamma, i)$ | Baseline | RIP | Baseline | RIP |
| DClassifier | MHEALTH | (0, 5) | 55.79±2.44 | **97.75±0.37** | 52.56±2.33 | **97.92±0.34** |
| | MHAD1 | (0, 16) | 33.86±1.46 | **86.03±0.59** | 32.85±1.66 | **86.03±0.58** |
| | MHAD2 | (0, 16) | 46.97±3.70 | **81.66±0.63** | 43.52±3.78 | **81.10±0.73** |
| | WHARF | (-1, 16) | 15.46±3.04 | **89.99±0.87** | 6.14±1.78 | **87.26±1.26** |
| | WISDM | (1, 5) | 53.03±3.04 | **99.79±0.05** | 44.94±2.51 | **99.66±0.08** |
| TS-Classifier | MHEALTH | (0, 1) | **61.00±4.09** | 58.83±2.34 | **57.42±4.38** | 53.99±2.72 |
| | MHAD1 | (-1, 16) | **35.62±1.98** | 29.92±1.02 | **32.58±2.39** | 25.99±1.12 |
| | MHAD2 | (-1, 16) | 48.31±3.62 | **51.44±2.85** | 41.97±4.12 | **41.50±3.32** |
| | WHARF | (-1, 1) | 20.94±3.16 | **44.90±4.10** | 11.41±2.47 | **26.61±2.07** |
| | WISDM | (1, 5) | 50.07±3.78 | **93.14±1.50** | 47.38±2.49 | **92.42±1.76** |
| TSBF | MHEALTH | (1, 1) | 31.44±2.30 | **31.50±2.38** | 26.63±2.36 | **26.79±2.10** |
| | MHAD1 | (1, 5) | 19.94±0.98 | **20.32±0.71** | 18.80±0.97 | 18.80±0.70 |
| | MHAD2 | (-1, 5) | 37.46±2.03 | **38.00±1.64** | 33.19±2.27 | **33.96±1.89** |
| | WHARF | (-1, 5) | **15.47±3.77** | 15.31±3.61 | **5.53±1.76** | 5.43±1.52 |
| | WISDM | (0, 16) | 39.10±6.34 | **39.39±6.54** | 29.22±2.46 | **30.58±2.88** |
| TSRF | MHEALTH | (-1, 5) | 29.72±1.68 | **29.91±2.42** | 25.98±1.07 | **26.52±1.75** |
| | MHAD1 | (-1, 1) | 21.51±0.84 | **21.75±0.82** | 19.59±0.97 | **19.85±0.96** |
| | MHAD2 | (-1, 5) | 34.65±2.95 | **36.15±2.43** | 31.13±3.08 | **32.77±2.65** |
| | WHARF | (0, 16) | 12.60±3.39 | **12.82±3.37** | 4.05±1.09 | **4.41±1.22** |
| | WISDM | (-1, 16) | 29.58±6.75 | **30.20±7.01** | 24.14±4.21 | **24.60±4.69** |

Table 2: Performance comparison of RIP and the baseline on real-world data (TRTR). Best metric values per dataset are shown in bold. RIP results correspond to the best hyperparameter configuration.

| Model | Dataset | RIP Config. | Acc. Base | Acc. RIP | F1 Base | F1 RIP |
|---|---|---|---|---|---|---|
| DClassifier | MHEALTH | (0,16) | 91.21±1.61 | **97.84±0.21** | 90.46±2.84 | **98.01±0.20** |
| | MHAD1 | (0,16) | 58.25±2.06 | **86.12±0.55** | 67.13±1.14 | **86.13±0.53** |
| | MHAD2 | (0,16) | 68.32±1.05 | **82.12±0.72** | 67.58±1.31 | **81.68±0.74** |
| | WHARF | (1,16) | 83.11±1.85 | **90.25±1.19** | 78.62±2.08 | **87.48±1.70** |
| | WISDM | (0,5) | 99.47±0.09 | **99.77±0.03** | 99.20±0.15 | **99.46±0.05** |
| TS-Classifier | MHEALTH | (1,1) | 32.37±2.40 | **59.98±2.03** | 24.08±2.59 | **55.54±2.42** |
| | MHAD1 | (-1,5) | 20.45±1.15 | **31.67±0.73** | 15.67±1.41 | **26.08±0.70** |
| | MHAD2 | (1,16) | 31.25±1.64 | **52.49±3.16** | 24.46±1.72 | **43.22±3.91** |
| | WHARF | (1,1) | 19.19±6.71 | **46.06±4.34** | 10.28±3.63 | **28.63±2.24** |
| | WISDM | (-1,5) | 90.32±1.63 | **93.12±1.21** | 87.19±2.62 | **92.30±1.46** |

points (p.p.) relative to their respective baselines. For example, a baseline F1 score of 10% with an improvement of 4 p.p. corresponds to a final score of 14%. Additional details and full results are provided in Appendix D.

**Hyperparameters** Our method introduces two hyperparameters: the constant $\gamma$ and the duplication factor $i$. In the synthetic-data experiments, we evaluated $\gamma \in \{-1, 0, 1, 5\}$ and $i \in \{1, 5, 16\}$. The same configurations were tested on the real datasets to assess the general applicability of RIP. Values of $\gamma$ in the range $[-1, 1]$ were selected to match the natural scale of the data, whereas $\gamma = 5$ was included as an out-of-range control to test robustness beyond this interval. To identify the best-performing configuration, we incorporated a greedy search using these predefined ranges. A complete description of all hyperparameter settings and additional experimental details is provided in Appendix C.

## 4 RESULTS AND DISCUSSION

**Results on TSTR.** Table 1 summarizes the best-performing RIP configurations, compared to the baselines. Our analysis, detailed in the Appendix D, reveals four key findings: (1) RIP disproportionately benefits deep learning models that rely on representation learning (DClassifier and TS-Classifier); (2) it enhances model fairness and robustness, not just predictive accuracy; (3) its effectiveness is highly dependent on the dataset characteristics; and (4) it can help bridge the synthetic-to-real performance gap. Results under RevTransformer, TimeTransformer, and LOSO are shown in Appendix D.

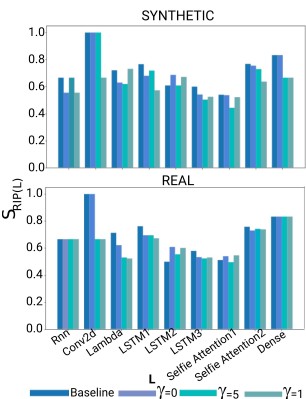

Figure 3: Layer-wise KS statistics for DClassifier weights trained on real and synthetic MHAD2 data, tested against a uniform reference distribution. The x-axis shows the model layer, and the y-axis reports the test statistic S.

A clear pattern emerges across all experiments: RIP provides stronger improvements for deep architectures (DClassifier and TS-Classifier) than their statistical counterparts (TSBF and TSRF). In some cases, RIP led to significant gains, such as over +53 percentage points in F1 (corresponding to more than +160% relative improvement) and even extreme gains of over +1300% relative improvement. This performance suggests that RIP's regularization mechanism is particularly beneficial for models engaged in representation learning, possibly by preventing overfitting to spurious features in the synthetic data. The datasets respond differently to RIP. For example, MHAD2 shows greater instability, with gains that depend on the adopted configuration. In contrast, WHARF consistently benefits, reaching some of the most significant relative improvements observed. Regarding the hyperparameter $i$, the distribution of its optimal value (40% for $i = 16$, 40% for $i = 5$, and 20% for $i = 1$) underscores that there is no single best setting. Instead, the required amount of regularization depends on the dataset's complexity and the model's capacity, highlighting the need to treat $i$ as a crucial hyperparameter to be tuned for each context.

**Results on TRTR.** Table 2 highlights the impact of RIP on real-world data, particularly for deep learning models. The results show consistent performance gains, alongside improvements in robustness and fairness. Results under RevTransformer and TimeTransformer are shown in Appendix D.

In general, RIP significantly improved the models, though in distinct ways. For DClassifier, which already had strong baselines, RIP consistently improved results to higher levels—for example, on MHAD1, Accuracy increased by +27.9 percentage points (pp) and F1 by +19.0 pp. On MHEALTH, F1 rose from 90.5% to 98.0%, setting a new performance bound. Despite weaker baselines, TS-Classifier achieved the most significant relative improvements: on MHEALTH, the F1 increased by +31.5 pp (a relative gain of 131%), and on WHARF by +18.4 pp ( 179%). These findings suggest that RIP is particularly effective at regularizing models that struggle to generalize. Analysis of the optimal configurations shows that the duplication factor $i$ is a dominant parameter: $i = 16$ was optimal in half of the cases. No single value of $\gamma$ consistently prevailed, with $\gamma = 0$ and $\gamma = 1$ each appearing in 40% of the best cases, indicating sensitivity to the model–dataset interaction. Tree-based models (TSRF and TSBF), reported in the Appendix, already achieved near-perfect performance ($> 99\%$ F1) under TSTR. RIP maintained or slightly improved these results, demonstrating that it does not degrade performance even in scenarios with little room for improvement. Overall, RIP emerges as a safe and effective regularizer that improves weaker models, stabilizes stronger ones, and contributes to fairer and more robust performance across datasets.

**Computational Cost.** Applying RIP is primarily influenced by the dataset size and the duplication factor $i$. As an illustrative example, applying RIP with $i = 16$ to the WISDM dataset on a CPU-based model increased runtime from 2 hours to 4 hours and memory usage from 4 GB to 10 GB. In contrast, using a GPU (e.g., NVIDIA RTX 3090) for DL models reduced the training time for the same configuration to approximately 30 minutes, with a comparable memory usage. More details in Appendix D.

**Robustness and Class Balance.** Beyond improving overall accuracy, RIP reduces the discrepancy between Accuracy and macro F1, indicating more balanced treatment of underrepresented classes. For synthetic MHEALTH, TS-Classifier narrowed this gap from roughly 8 pp to about 4 pp with RIP, demonstrating that the method alleviates class imbalance effects. RIP also improves stability: standard deviations consistently decrease across datasets and models. For example, on MHEALTH with DClassifier, the Accuracy standard deviation drops from ±1.61 to ±0.21, yielding more reliable outcomes. Together, these results show that RIP enhances robustness, reduces prediction bias across classes, and provides more trustworthy model behavior.

**Regularization Effect.** To characterize RIP as a regularizer, we performed a layer-wise analysis of DClassifier weight distributions using Kolmogorov–Smirnov and Wasserstein distances (Appendix E). Figures 3 and 4 show that RIP—especially with $\gamma = 1$—consistently pushes weights toward a more uniform distribution, with the strongest effects in contextual layers such as Self-Attention and LSTM. Unlike $\ell_1$ and $\ell_2$, which enforce sparsity or shrink weights toward zero, RIP promotes uniformity across the weight range. This encourages broader use of model weights, reducing overfitting and improving confidence calibration. Higher perturbation strengths ($\gamma = 5$) produce stronger but less stable effects, whereas $\gamma = 0$ yields moderate regularization. Overall, RIP introduces a distinct and controllable regularization behavior: it improves local weight uniformity without collapsing the global structure when $\gamma = 1$, offering the most stable balance across datasets. Additional layer-wise dynamics are provided in Appendix E.

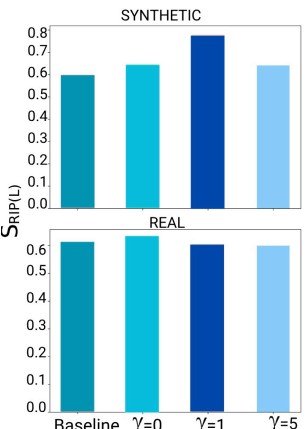

Figure 4: KS statistics between model output logits (real vs. synthetic MHAD2) and a uniform reference distribution. Lower values indicate outputs closer to uniformity. Each bar corresponds to a different model setting.

**On the Hyperparameter and Stability** To make RIP practical and reproducible, we provide an

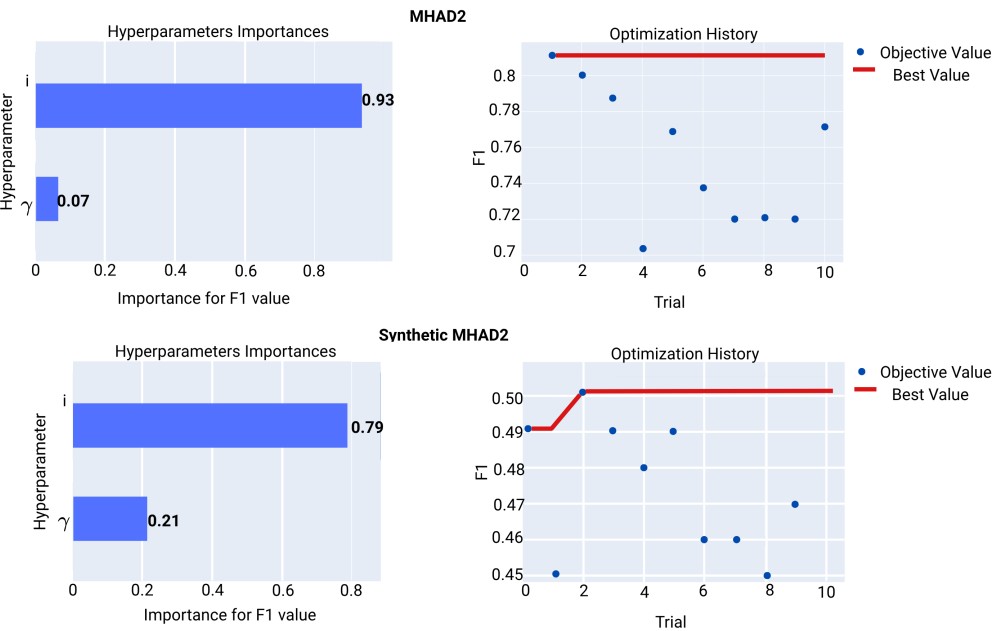

Figure 5: Hyperparameter optimization for RIP on real (top) and synthetic (bottom) MHAD2. Left: importance analysis showing $i$ as the dominant parameter. Right: trial history with rapid and stable convergence across 10 trials.

automatic hyperparameter search procedure based on Optuna Akiba et al. (2019). The goal is to minimize manual tuning while identifying suitable parameters via $(\gamma_{\mathrm{opt}}, i_{\mathrm{opt}}) = \arg\max_{\gamma,i} \mathrm{F1}(\gamma, i)$.

Table 3: Performance of RIP compared with Baseline, $\ell_1$, and $\ell_2$ regularization on the TS-Classifier across four real-world datasets. Best results are in bold.

| Dataset | Method | Accuracy (%) | F1 (%) |
|---|---|---|---|
| MHEALTH | Baseline | 32.37±2.40 | 24.08±2.59 |
| | RIP ($\gamma = 1, i = 1$) | **59.98**±2.03 | **55.54**±2.42 |
| | $\ell_1$ ($\epsilon = 1.0$) | 57.71±2.60 | 51.67±3.56 |
| | $\ell_2$ ($\epsilon = 0.1$) | 58.25±3.37 | 51.61±4.45 |
| MHAD1 | Baseline | 20.45±1.15 | 15.67±1.41 |
| | RIP ($\gamma = -1, i = 5$) | 31.67±0.73 | 26.08±0.70 |
| | $\ell_1$ ($\epsilon = 0.001$) | 31.49±1.95 | 27.62±2.00 |
| | $\ell_2$ ($\epsilon = 0.001$) | **33.42**±1.95 | **29.70**±2.29 |
| MHAD2 | Baseline | 31.25±1.64 | 24.46±1.72 |
| | RIP ($\gamma = 1, i = 16$) | **52.49**±3.16 | **43.22**±3.91 |
| | $\ell_1$ ($\epsilon = 0.001$) | 47.13±6.13 | 40.94±6.24 |
| | $\ell_2$ ($\epsilon = 0.001$) | 44.65±6.04 | 37.84±6.03 |
| WHARF | Baseline | 19.19±6.71 | 10.28±3.63 |
| | RIP ($\gamma = 1, i = 1$) | **46.06**±4.34 | **28.63**±2.24 |
| | $\ell_1$ ($\epsilon = 0.001$) | 19.51±2.09 | 10.92±1.98 |
| | $\ell_2$ ($\epsilon = 0.001$) | 20.04±3.16 | 10.90±2.73 |
| WISDM | Baseline | 90.32±1.63 | 87.19±2.62 |
| | RIP ($\gamma = -1, i = 5$) | **93.12**±1.21 | **92.30**±1.46 |
| | $\ell_1$ ($\epsilon = 0.001$) | 43.68±5.46 | 43.94±4.06 |
| | $\ell_2$ ($\epsilon = 0.001$) | 47.88±6.81 | 46.12±3.21 |

We restricted the search space to $i \in \{1, 5\}$ and $\gamma \in \{1, 5, 16\}$. Even within this reduced domain, the best-performing configuration remained close to the full-search optimum (the optimal $\gamma$ was not included in the reduced set) and, crucially, the duplication factor consistently converged to $i = 16$. Figure 5 summarizes the optimization process for both real (top) and synthetic (bottom) MHAD2. The hyperparameter importance analysis (left panels) shows that $i$ accounts for 93% of the variance in F1 for the real dataset and 79% for the synthetic dataset. The optimization history (right panels) presents the trial-by-trial F1 values (blue) and the best-so-far curve (red): in both settings, Optuna converges rapidly, with stable optima emerging within the first few trials. Overall, although the optimal pair $(\gamma, i)$ is dataset-dependent—as expected—the RIP hyperparameter landscape is stable, smooth, and dominated by a single parameter ($i$), as previously commented.

**Why RIP Improvements Are More Noticeable in Synthetic Data?** Although TLC-GAN is designed for wearable-sensor signals, its generated samples still exhibit a measurable synthetic–real distributional gap. This gap is consistently reflected in our experiments: models trained or evaluated exclusively on synthetic data achieve markedly lower F1 scores. This limitation aligns with common observations in generative modeling (synthetic → Real gap) Tremblay et al. (2018). RIP operates precisely within this residual space. RIP functions as a refinement mechanism that enhances the structure of synthetic sequences by enforcing temporal coherence and suppressing artifact-like fluctuations. Consequently, RIP shifts synthetic samples closer to the real-data distribution. Figure 6 illustrates this behavior. In the first panel (Real vs. Synthetic), the synthetic samples are clearly offset from the real distribution. After applying RIP (left panel), the synthetic+RIP features exhibit substantially greater overlap with the real samples. Additional per-axis visualizations and temporal examples are included in the Appendix E.

## 5 ADDITIONAL EXPERIMENTS

**RIP vs. Traditional Regularization.** To assess the value of RIP as a general-purpose regularization method for HAR, we compared it against standard $\ell_1$ and $\ell_2$ techniques across both real and synthetic datasets, using two architectures (TS-Classifier and DClassifier). Table 3 consistently showed that RIP either matched or outperformed traditional methods, particularly in challenging or high-baseline scenarios. RIP was more robust across datasets with varying complexity, preserving or improving performance even where $\ell_1$ and $\ell_2$ caused degradation—up to 50 percentage points in some cases. It also proved more versatile, delivering consistent gains across a wide range of $\gamma$ and $i$ configurations, without requiring extensive tuning. RIP demonstrated generalization across archi-

tectures, showing both models' effectiveness, while traditional regularizers offered only marginal or inconsistent improvements. RIP provides a stable and straightforward alternative in scenarios where regularization is needed but domain-specific tuning is impractical. Due to space limitations, we have presented the results for the TS-Classifier exclusively in the main body of this paper, as they most accurately reflect the overall behavior observed in our experiments. Comprehensive results, methodological justifications, and all corresponding tables are available in Appendix G.

**RIP vs. HAR approaches.** Our literature review revealed a scarcity of research addressing regularization and domain generalization specifically for HAR. The most relevant prior work, Bento et al. (2023), explored Mixup and Distributionally Robust Optimization (DRO) for accelerometer-based HAR, partially aligning with our objectives. To ensure a fair comparison, we benchmarked RIP against these approaches, along with Cutmix Yun et al. (2019), DRO, and Mixup Zhang et al. (2018) using the MHAD2 dataset. For brevity, only the best-performing configuration of each competing method is reported in Tab. 4. A complete set of results, parameter sweeps, and detailed commentary is provided in the Appendix G.

Across all metrics, RIP outperforms both the baseline and compared approaches. In addition to achieving higher Accuracy and F1-scores, RIP exhibits substantially narrower confidence intervals (Table 4), indicating more stable training dynamics and more reliable predictions. Moreover, RIP preserves a tight alignment between Accuracy and macro F1, producing more balanced outputs and reducing sensitivity to underrepresented classes. Although this effect is less pronounced on real datasets, RIP remains competitive with the strongest methods, reinforcing its ability to achieve fairer, more consistent outcomes. While techniques such as Cutmix achieve moderate improvements on synthetic data, their performance degrades considerably when transferred to real data, amplifying the synthetic–real gap. In contrast, RIP consistently delivers state-of-the-art results in both domains, underscoring its unique ability to bridge this critical distributional divide.

**RIP in Other Domains.** We conducted preliminary experiments on general time-series and tabular datasets to explore RIP's applicability beyond wearable sensor data. Results suggest that RIP's most substantial benefits arise in structured, repetitive sensor settings, while improvements in other domains are modest or inconsistent. Significantly, performance rarely degrades substantially, indicating that RIP is a safe-to-try regularizer even outside its primary target. These findings reinforce RIP's specialization for HAR while also pointing toward future research opportunities, such as adapting the invariant framing principle to multimodal or irregular time-series domains. For completeness, we report and discuss detailed per-domain results in the Appendix G.

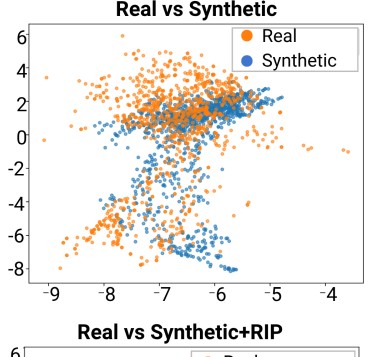

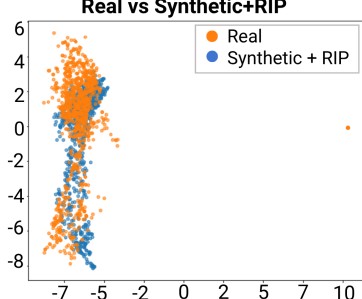

Figure 6: Effect of RIP on the synthetic–real distributional gap. PCA projections comparing (left) Real vs. Synthetic and (right) Real vs. Synthetic+RIP.

## 6 THEORETICAL ANALYSIS

Empirically, context-based architectures are the most sensitive to RIP. To ground these findings, we provide a formal analysis. We begin with a simple Recurrent Neural Network (RNN) with bias $b \in \mathbb{R}^h$ and identity activation $\varphi(x) = x$. A corresponding derivation for Transformer layers, as well as the case of duplicated spans $i \neq 1$, is presented in Appendix F. Additional details and full formalizations are also provided in the appendix. At time step $t$, the input is $x_t \in \mathbb{R}^d$ with target $y_t \in \mathbb{R}^q$. The RNN evolves as $h_t = W x_t + U h_{t-1} + b$, $o_t = V h_t + C$, $\hat{y}_t = \text{softmax}(o_t)$, where $W \in \mathbb{R}^{d \times h}$, $U \in \mathbb{R}^{h \times h}$, $V \in \mathbb{R}^{h \times q}$ and $C \in \mathbb{R}^q$. We simulate a sequence of 5 steps, with $h_0 = 0$, where constant samples $\gamma$ are injected at $t = 1, 2, 4, 5$ ($x_1 = x_2 = x_4 = x_5 = \gamma$). The hidden state

at $t = 5$ under RIP is

$$h_5^{(\text{RIP})} = (U^4 + U^3 + U + I)W\gamma + U^2 W x_3 + \sum_{i=0}^{4} U^i b, \tag{1}$$

In contrast, without RIP, we have

$$h_5^{(\text{noRIP})} = \sum_{i=0}^{4} \left( U^i W x_{5-i} + U^i b \right). \tag{2}$$

Equations 1, 2 show that RIP substantially constrains $h_5$: most of its variability now arises from a single non-repeated input ($x_3$), while all other terms collapse to the deterministic component $W\gamma$ propagated through powers of $U$ Assuming i.i.d. inputs with mean $\mu$ and covariance $\Sigma$, the expectations and variances are

$$\mathbb{E}[h_5^{(\text{noRIP})}] = \sum_{i=0}^{4} U^i W \mu + \sum_{i=0}^{4} U^i b, \quad \text{Var}(h_5^{(\text{noRIP})}) = \sum_{i=0}^{4} U^i W \Sigma W^\top (U^i)^\top,$$

$$\mathbb{E}[h_5^{(\text{RIP})}] = (U^4 + U^3 + U + I)W\gamma + U^2 W \mu + \sum_{i=0}^{4} U^i b, \quad \text{Var}(h_5^{(\text{RIP})}) = U^2 W \Sigma W^\top (U^2)^\top.$$

Thus, while $\text{Var}(h_5^{(\text{noRIP})})$ aggregates variability from five independent sources, $\text{Var}(h_5^{(\text{RIP})})$ depends only on $x_3$. RIP therefore reduces temporal diversity and constrains the hidden representation to a lower-variance subspace. This implicit regularization yields smoother gradients and more stable weight updates: $\Omega \leftarrow \Omega - \alpha \frac{\partial \ell}{\partial \Omega}, \quad \Omega \in \{W, U, V, b\}$, where $\frac{\partial \ell}{\partial \Omega}$ inherits the reduced variability of $h_5$. While this promotes generalization, excessive repetition (a large duplication factor $i$) can over-constrain the model and limit its representational capacity. Let $\varphi$ be Lipschitz-continuous with constant $L$ (e.g., ReLU, tanh, GELU). Then:

$$\text{Var}(h_t) = \text{Var}(\varphi(z_t)) \leq L^2 \text{Var}(z_t), \qquad z_t = W x_t + U h_{t-1} + b.$$

This yields the recursive bound:

$$\text{Var}(h_t) \leq L^{2t} \sum_{i=0}^{t-1} U^i W \text{Var}(x_{t-i}) W^\top (U^i)^\top.$$

Under RIP, only one input window contributes stochasticity, leading to:

$$\text{Var}(h_t^{(\text{RIP})}) \leq L^{2t} U^k W \Sigma W^\top (U^k)^\top,$$

whereas without RIP:

$$\text{Var}(h_t^{(\text{noRIP})}) \leq L^{2t} \sum_{i=0}^{t-1} U^i W \Sigma W^\top (U^i)^\top.$$

Thus, nonlinearities introduce a global factor $L^{2t}$ but preserve the core effect: RIP suppresses variance by limiting the number of independent stochastic inputs.

## 7 ABLATIONS

To better understand the core mechanisms behind our proposed RIP method, we conducted a series of ablation studies addressing three key questions. While we report high-level findings here, complete experimental setups and extended results are detailed in Appendix H.

**Can naive data duplication achieve similar effects to invariant patterns?** Not entirely. Duplicating input windows (e.g., $i = 5$) provides marginal improvements over the TSTR baseline, but these gains are not statistically significant and vanish for larger $i$. This indicates that naive repetition may lead to overfitting, limiting generalization—especially in time-series, where subtle variations are critical.

**Is $i$ as a duplication factor necessary?** Yes. When using a minimal RIP structure with only one central window (i.e., $\tilde{i} = \frac{1}{2}i$), performance drops below the synthetic baseline. Full duplication patterns (e.g., $i = 5$) result in substantial gains—up to 4 percentage points in F1 score—confirming that structural repetition enhances the contextual framing effect and model focus.

**Can random distributions replace constants?** No. Replacing $\gamma$ with non-stationary random values (e.g., $\text{rand}(0,1)$ or $\text{rand}()$) consistently degrades performance. Even the best randomized setup merely matches the baseline. These results suggest that randomness introduces spurious patterns, confusing the model rather than improving robustness.

**Does structure and $\gamma$-design matter?** Absolutely. Experiments with unordered or non-integer $\gamma$ (e.g., $\gamma = \text{Avg(features)}$) led to performance degradation. This confirms that the contextual frame's order and fixed design are essential for RIP's effectiveness. These findings highlight that the benefits of RIP arise not from trivial data augmentation or randomness, but from the careful design of its structure and components. Additional experiment, tables, and visualizations in Appendix H.

Table 4: Comparison of RIP with standard data-driven regularization methods under both TSTR and TRTR protocols. Best results for each protocol are highlighted in bold.

| Method | Accuracy (%) | F1-Score (%) | Epsilon | Protocol |
|---|---|---|---|---|
| Baseline | 46.97±3.70 | 43.52±3.78 | - | |
| Cutmix | 45.38±2.83 | 41.80±4.05 | 0.1 | |
| DRO | 32.70±4.71 | 26.88±4.99 | 0.3 | TSTR |
| Mixup | 39.88±6.14 | 32.33±9.63 | 0.1 | |
| RIP ($\gamma$=0, $i$=16) | **81.66 ±0.63** | **81.10 ±0.73** | - | |
| Baseline | 68.32±1.05 | 67.58±1.31 | - | |
| Cutmix | 63.10±1.18 | 62.03±1.47 | 0.1 | |
| DRO | 45.75±2.08 | 38.91±2.76 | 0.3 | TRTR |
| Mixup | 39.98±12.17 | 30.89±16.12 | 0.1 | |
| RIP ($\gamma$=0, $i$=16) | **82.12 ± 0.72** | **81.68 ± 0.74** | - | |

## 8 CONCLUSION

We introduced RIP, a data-centric regularization strategy tailored for tri-axial wearable sensor data. By augmenting training batches with invariant patterns, RIP improves model generalization without requiring architectural modifications or changes to the loss function. Our experiments demonstrate that RIP consistently enhances both performance and calibration in deep learning models for HAR, yielding gains across synthetic and real datasets. RIP directly addresses core challenges in wearable-sensor data—such as scarcity, variability, and noise—by reducing weight variance, mitigating overconfidence, and promoting more uniform and stable weight distributions. We show that RIP reshapes the optimization landscape through its batch-level update dynamics, and that its regularization effects cannot be reproduced by simply holding most samples constant; the structure of RIP is essential. Among its hyperparameters, the duplication factor $i$ proved most influential, with $i = 16$ providing robust improvements across settings. While RIP shows limited benefits outside its target domain, it remains a lightweight, architecture-agnostic technique with strong practical value for sensor-based systems. It requires no additional computational cost beyond batch construction and integrates seamlessly into existing training pipelines. RIP thus extends the applicability of domain randomization approaches and offers a promising avenue for applications in healthcare monitoring, sports analytics, and eldercare, where reliable HAR is critical. **Limitations and Future Work.** Although the SOLO protocol controls key confounding factors, our study focuses mainly on inter-subject generalization and on a single-sensor setup. Future work may extend RIP to multimodal sensors and broader intra-subject evaluations, and refine the search for optimal hyperparameters.

## REPRODUCIBILITY STATEMENT

The source code associated with this work will be released on GitHub upon completion of the review process. Our implementation is developed on top of the TensorFlow framework, and we

explicitly reference any external code utilized, including the classifiers incorporated, which were not reimplemented from scratch. A comprehensive description of all hyperparameters is provided in the Appendix, and the corresponding configuration files are included in the source code repository to facilitate reproducibility.

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
