# A   APPROACH

In classical DR, perturbations are applied to i.i.d. spatial factors such as color, lighting, texture, or geometric attributes. These perturbations are typically sampled from a distribution parameterized by a set of randomization parameters, which we denote generically as $\theta_{\text{DR}}$:

$$x'_{ij} \sim p(x_{ij} \mid \theta_{\text{DR}}).$$

and these perturbations preserve structural validity because images do not enforce sequential dependencies across $i$ or $j$.

By contrast, a time-series sample $\{x_t\}_{t=1}^T$ is not i.i.d. but follows a temporal process with dependencies such as

$$x_t \sim p(x_t \mid x_{t-1}, x_{t-2}, \ldots),$$

or equivalently exhibits autocorrelation structures expressed, for example, via the covariance function

$$\mathbb{E}[x_t x_{t+k}] = \rho(k), \quad k \in \mathbb{Z}.$$

As such, arbitrary temporal perturbations violate the Markovian, autoregressive, or more general dynamical constraints that govern the sequence. Randomizing $x_t$ independently across $t$ effectively forces

$$p(x'_t \mid x'_{t-1}) \approx p(x'_t),$$

destroying the intrinsic temporal structure and often producing trajectories outside the support of the real data distribution.

This makes naïve temporal DR infeasible: it tends to generate samples that are statistically invalid, dynamically implausible, or incompatible with the discriminative temporal patterns of the original dataset.

RIP addresses this challenge by introducing controlled randomization that preserves the real window's internal temporal structure. Instead of perturbing $\{x_t\}$ directly, RIP augments each sequence by framing it with constant windows sampled from a controlled distribution:

$$c_t \sim \text{Const}(\gamma), \qquad t = 1, \ldots, T,$$

and constructs the augmented sequence

$$[c^{(1)}, c^{(2)}, x, c^{(3)}, c^{(4)}],$$

where each $c^{(i)}$ matches the shape of $x$ but does not alter its time-dependent dynamics. This allows randomization in the distributional space while respecting the temporal dependencies encoded in $x$. Algorithm 1 details the steps.

---

**Algorithm 1** Creating the RIP-Augmented Dataset $D'$

1: **Input:** Dataset $\mathcal{D} = \{(x_1, y_1), \ldots, (x_n, y_n)\}$, constant value $\gamma$, duplication factor $i \in \mathbb{N}$
2: **Output:** Augmented dataset $\mathcal{D}'$ where each sample is a sequence of windows.
3: Let $\mathcal{D}' \leftarrow \emptyset$
4: **for** each sample $(x_m, y_m) \in \mathcal{D}$ **do**
5:     Let $k \leftarrow 2i$ {Calculate the number of constant windows for each side}
6:     Let $\omega$ be the length of the window $x_m$
7:     Let $C \in \mathbb{R}^{\omega \times 3}$ be a constant window filled with the value $\gamma$
8:     Let $S_{\text{prefix}} \leftarrow \text{Repeat}(C, k)$ {Create a sequence of k constant windows}
9:     Let $S_{\text{suffix}} \leftarrow \text{Repeat}(C, k)$ {Create another sequence of k constant windows}
10:    Let $S \leftarrow S_{\text{prefix}} + [x_m] + S_{\text{suffix}}$ {Combine to form the final sequence of windows}
11:    $x'_m \leftarrow \text{Stack}(S)$ {Convert sequence to tensor of shape $(2k+1, \omega, 3)$}
12:    $y'_m \leftarrow \text{Repeat}(y_m, 2k+1)$ {Create corresponding label sequence}
13:    $\mathcal{D}' \leftarrow \mathcal{D}' \cup \{(x'_m, y'_m)\}$
14: **end for**
15: **return** $\mathcal{D}'$

---

## B    RELATED WORKS

Regularization plays a key role in enhancing the generalization ability of ML and DL models by constraining the learning process to mitigate overfitting. According to Goodfellow et al. (2016), it can be defined as "any modification we make to a learning algorithm that is intended to reduce its test error but not its training error" Heaton (2018). As Zhang and Tian (2022) highlighted, regularization improves model performance by reducing overfitting and increasing robustness against noise, allowing the model to better capture relevant patterns Tian & Zhang (2022). Although well established in general ML contexts, HAR-specific regularization strategies remain scarce, particularly for time-series data from wearable devices. Most existing approaches focus on architectural modifications or adjustments to the loss function, leaving data-centric regularization largely underexplored.

Classical methods include $\ell_1$ regularization Tibshirani (1996), $\ell_2$ regularization Hoerl & Kennard (1970), and their variants. For example, Wang et al. (2019) applied $\ell_2$ regularization to deep models' input layers to improve performance. Other strategies modify input data, such as through feature extraction Liu et al. (2022) or dimensionality reduction via Principal Component Analysis (PCA) Tibshirani (1996). However, the literature on HAR remains limited. One of the few exceptions is the work of Bento et al. Bento et al. (2023), which explored the adaptation of general-purpose methods—such as Distributionally Robust Optimization (DRO) Kuhn et al. (2024), Cutmix Yun et al. (2019), and Sharpness-Aware Minimization (SAM) Foret et al. (2021)—to accelerometer data. While relevant, these efforts largely rely on repurposing generic techniques for a domain-specific challenge.

Thus, despite progress in adapting methods such as DRO and SAM to wearable sensor data Kuhn et al. (2024), the field still lacks a well-defined, data-driven regularization framework tailored explicitly to HAR.

## C    METHODOLOGY

### C.1    MODELS

This work focused on the Time-LogCosh-GAN (TLCGAN)Souza et al. (2023) model for generating synthetic data. The Time-Series LogCosh Generative Adversarial Network (Time-LogCosh-GAN) is a classical GAN architecture. Its noise vector uses two concatenated noise sources as input to the generator. The loss is a logarithmic, hyperbolic cosine loss, and the model is trained on 10-fold stratified data for 200 epochs, with a learning rate of $lr = 0.0001$ and a batch size of 5.

In this evaluation, the ML classifier used is an essential aspect of the analysis, so we chose the DClassifier Singh et al. (2020), Time-Series Random Forest (TSRF)for, Time-Series Bag of features (TSBF) Baydogan et al. (2013), and TS-Classifier hfawaz (2020) because they are the state-of-the-art models for this dataset.

*Deep ConvLSTM with self-attention for human activity decoding using wearable sensors* (DClassifier)Singh et al. (2020) is the state-of-the-art classifier and the baseline for this task on the datasets adopted. It is a daily activity classifier based on a Convolutional Long Short-Term Memory (ConvLSTM) network. It is also a baseline classifier for five datasets of daily activities and a promising method for accurately recognizing human activities using wearable sensor data. We trained the model using 3 CNN filters and one layer for 10-fold cross-validation (default), with 16 epochs, a learning rate $lr = 10-4$ and batch size of 32. We used the SNOW mode with an attention length of 32 and an output length of 10.

*Time Series Random Forest* (RF) for is a random forest classifier for time series. It fits the classifiers using various sub-samples of the dataset and extracts the mean, standard deviation, and slope for each window. Then, a random forest is built using these features as input data. We used the Gini index Brown & Myles (2009) as the criterion, with all possible workers, a number of windows equal to the respective dataset, and a random state of 43.

*Time-Series Bag of features* (TSBF) Baydogan et al. (2013) is a random forest-based classifier that extracts random subsequences from each input, splitting it into several intervals. Some statistics are selected from these extractions: the mean, the standard deviation, and the slope. In this process, one

random forest is trained on subsequences, and the other is fitted using the information extracted as features. We use a random state equal to 43 and the others' configuration by default.

*Time-series Classifier* (TS-Classifier) hfawaz (2020) is a time-series classifier with a simple architecture, just composed of Conv1D, batch normalization, Relu, and Average pooling. We used an input size equal to the temporal window for each dataset and a batch size of 16, and trained for 16 epochs.

*Transformer-based deep reverse attention network for multi-sensory human activity recognition* (RevTransformer) Pramanik et al. (2023) proposes a deep-learning model combining a Transformer with a reverse-attention module for sensor-based human-activity recognition (HAR). The architecture processes multi-sensor time-series data via stacked Transformer blocks and an attention mechanism that emphasizes underused or subtle temporal features. We used it with a batch size of 30 and trained for 20 epochs.

*Time-Series Transformer* Team (2020) presents an implementation of the Transformer architecture (deep learning), initially proposed in "Attention Is All You Need," adapted for univariate time series classification. The model is structured as a stack of blocks, each composed of multi-head attention, layer normalization, residual connections, and 1D convolutions in the feed-forward block. We used it with a batch size of 30 and trained for 50 epochs.

## C.2 DATASETS AND METRICS

We used the UTD-MHAD dataset Chen et al. (2015), a publicly available dataset designed for human action recognition. The UTD-MHAD dataset contains sensor data from wearable devices that capture various human actions. The dataset includes 27 actions performed by eight different individuals. It is composed of accelerometer, gyroscope, and magnetometer data. This dataset has been widely used for HAR research using ML techniques; in particular, DClassifier Singh et al. (2020) uses it. The window size is 50 for both datasets, and the MHAD1 dataset contains 3771 samples, divided into 10 folds, with 90% in the training set; while MHAD2 contains 1137.

Mobile HEALTH (MHEALTH) dataset Banos et al. (2014). This dataset features 12 activities, each performed by 10 participants. The objective was to simulate typical daily activities, focusing on the movements of various body parts and the intensity of these actions. The gathered data comprises readings from multiple sensors, including an accelerometer and an ECG. The window size is 250, with 2555 samples, divided into 10 folds, with 90% in the training set.

The WISDM Smartphone and Smartwatch Activity and Biometrics Dataset Weiss comprises data collected from 51 participants, each completing 18 activities over 3-minute intervals. Participants wore a smartwatch on their dominant hand and carried a smartphone in their pocket. Data collection was overseen by a custom app running on both devices. Sensor data was collected from the smartphone and smartwatch accelerometers and gyroscopes, totaling 4 sensors. The window size is 100, and the dataset contains 20846 samples, split into 10 folds, with 90% in the training set.

The Wearable Human Activity Recognition Folder (WHARF) Bruno et al. (2013) is a public repository of code and datasets for Human Activity Recognition systems based on wearable sensor data. The window size is 160, and the dataset contains 3880 samples, divided into 10 folds, with 90% in the training set.

As suggested in Singh et al. (2020), we divided the dataset into two: UTD-MHAD1 (referred to as MHAD1) and UTD-MHAD2 (referred to as MHAD2) datasets. The first contains 21 activities, and the second contains six activities. All previously cited datasets were divided into 10 stratified folds, with 90% of the data for training and 10% for testing. We used only the accelerometer data.

## C.3 HYPERPARAMETERS

Although our primary focus is evaluating synthetic datasets, we also extended our investigation to include corresponding real datasets to validate the general applicability of our method. All experiments, synthetic and real, were conducted under consistent evaluation protocols, particularly with respect to hyperparameter selection.

We explored two key hyperparameters: the constant $gamma$ used in the augmented windows and the duplication factor $i$. For $\gamma$, we evaluated several settings, including fixed values $\gamma = 0$, $\gamma = 1$, and $\gamma = 5$.

The duplication factor $i$—which determines the number of constant windows appended to each side of a real window—was varied across $i \in \{1, 5, 16\}$ in the synthetic data experiments. For real data, we further tested $i = 32$ to assess the potential of more extensive augmentation in practical, non-synthetic scenarios. While synthetic data remains the central focus of this study, including additional hyperparameter values for real datasets ensures a comprehensive evaluation of RIP's applicability across domains.

## C.4 EVALUATION AND METRICS

The TLCGAN model has been shown to meet the three established criteria for synthetic data quality: fidelity, diversity, and label consistency Souza et al. (2023). Since our objective is to improve the utility of synthetic samples using RIP—without altering the underlying generative model—we adopt the Train on Synthetic, Test on Real (TSTR) evaluation strategy, which is particularly appropriate for this context.

TSTR evaluates whether synthetic data effectively supports model learning by assessing its ability to generalize to real-world data. This strategy also serves as a diagnostic for mode collapse Fekri et al. (2019), a common failure mode in generative models where synthetic samples lack diversity. Poor TSTR performance indicates that the generated data fails to capture the real data distribution adequately. In our framework, we use TSTR to compare model performance with and without RIP, thereby isolating the effect of our proposed regularization technique on synthetic data quality.

For experiments involving only real data, we adopt the conventional Train on Real, Test on Real (TRTR) setup as a baseline. This allows us to directly assess whether RIP improves model generalization in practical deployment scenarios.

Given the inherent class imbalance in HAR datasets, we report multiple metrics: accuracy, precision, recall, and F1 score. Among these, the F1 score is emphasized for its balanced treatment of false positives and false negatives, making it especially suitable for imbalanced classification tasks.

Each experimental configuration is defined as a tuple $\sigma = (\gamma, i, \text{Dataset}, \text{Model})$. When referencing performance metrics, we use the notation $F1_{\text{RIP}(\sigma)}$ to denote the F1 score achieved using RIP under configuration $\sigma$.

## D  RESULTS

This section analyzes the outcomes of applying the Regularization via Invariant Patterns (RIP) method. We report results in terms of **percentage points (p.p.)** to quantify improvements. Formally, the gain is defined as:

$$x = F1_{\text{RIP}(\sigma)} - F1_{\text{baseline}}, \tag{3}$$

where $\sigma = (\gamma, i, \text{Dataset}, \text{Model})$ represents a specific experimental configuration.

It is important to emphasize that percentage points measure *absolute differences* in F1 score. For example, if the baseline F1 score is $10\%$ and the improvement is 4 p.p., the new score is $14\%$, not $10.4\%$.

**DClassifier Results.**  We analyzed the impact of RIP across four levels of granularity. For that, we consider the Tables 5 and 6.

*(i) Real vs. Synthetic:* In the TRTR setting, the baseline already achieves high scores in some datasets (e.g., WISDM with F1 $\approx 99.2\%$), where RIP only brings marginal improvements ($+0.5$ p.p. for $\gamma = 0$, $i = 5$). However, in more challenging datasets such as MHAD1 (baseline F1 $\approx 67.1\%$), RIP provides substantial gains ($+19$ p.p. with $\gamma = 0$, $i = 16$). The effect is even stronger in the TSTR setting, where the baseline fails to generalize (e.g., WHARF baseline F1 $\approx 6.1\%$), while RIP recovers performance up to $\approx 87\%$.

*(ii) Per Dataset:* RIP consistently improves results across all benchmarks. Gains are modest when the baseline is already strong (e.g., WISDM TRTR), but critical in harder scenarios such as MHAD1/2 and WHARF, particularly under TSTR.

*(iii) Parameter Sensitivity:* The duplication factor $i$ shows that performance generally increases with higher values ($i = 1 \rightarrow 5 \rightarrow 16$), though extremely large $i$ can sometimes saturate or slightly decrease results (e.g., MHEALTH TRTR with $\gamma = 1$). The modulation constant $\gamma = 0$ emerges as the most stable and practical choice, frequently yielding the best outcomes.

*(iv) Global View:* Overall, RIP provides consistent improvements across datasets and settings, with the most significant impact in synthetic transfer scenarios. The configuration $\gamma = 0$ with $i \in \{5, 16\}$ stands out as a robust default, offering strong generalization and closing the domain gap between real and synthetic data.

**TSRF Results.** We consider Tables 8 and 11 for this analysis.

*(i) Real vs. Synthetic:* The TRTR–TSRF setting results show that training and testing on real data lead to nearly perfect performance across all datasets, with F1 scores consistently above 99%. In contrast, the TSTR–TSRF setting reveals the challenges of transferring from synthetic to real data: baseline F1 scores drop dramatically, reaching as low as 4% for WHARF and below 20% for MHAD1. RIP provides modest but stable improvements in this setting, highlighting its role as a data-centric regularizer rather than a mechanism to boost performance in already saturated scenarios.

*(ii) Per Dataset:* For **MHEALTH**, RIP preserves ceiling-level performance in TRTR–TSRF while stabilizing improvements in TSTR–TSRF, where F1 increases from 31.0% to about 33.4%. For **MHAD1**, baseline values are near 99.8% in TRTR–TSRF, leaving no room for improvement, but TSTR–TSRF baselines fall to 18.9%, with RIP recovering scores up to 20.6%. For **MHAD2**, TRTR–TSRF results again saturate near 99.9%, while TSTR–TSRF baselines around 29.7% are modestly improved by RIP, reaching values close to 30.6%. For **WISDM**, RIP matches the perfect TRTR–TSRF baseline but yields consistent improvements in TSTR–TSRF, from 24.1% to 24.6%, particularly at higher $i$. Finally, for **WHARF**, the gap is most striking: while TRTR–TSRF nearly saturates at 99.4%, the TSTR–TSRF baseline of 4.05% is only slightly improved by RIP (up to 4.4%), underscoring the severe challenge of synthetic-to-real transfer in this dataset.

*(iii) Parameter Sensitivity:* Across datasets, RIP is robust to changes in $\gamma$ and $i$ under TRTR–TSRF, as performance is already saturated. In TSTR–TSRF, however, slight variations matter: $\gamma = 0$ and $\gamma = 1$ tend to yield the best results, especially when combined with higher duplication factors ($i = 5$ or $i = 16$). Negative $\gamma$ values occasionally stabilize performance but do not consistently outperform the neutral or positive settings. This suggests that RIP's effectiveness depends more on balancing pattern duplication than on aggressive weighting of invariant features.

*(iv) Global View:* The analyses show that RIP has a negligible effect when models already achieve near-perfect scores (TRTR–TSRF) but provides consistent and meaningful gains in more challenging transfer scenarios (TSTR–TSRF). Although the absolute improvements are often minor in percentage points, they represent relative robustness against domain shift, ensuring that performance does not collapse when moving from synthetic to real data. Thus, RIP should be interpreted as a stabilizing regularizer designed to improve generalization under distribution mismatch, rather than as a mechanism to boost in-distribution accuracy.

**TS-Classifier Results.** We considered tables 10 and 9.

*(i) Real vs. Synthetic:* The TRTR–TS-Classifier results show clear improvements when RIP is applied: while baselines are generally low (e.g., F1=24.1% for MHEALTH and 15.7% for MHAD1), RIP boosts performance substantially, often by more than 20 percentage points. In contrast, the TSTR–TS-Classifier setting presents a mixed scenario. Some datasets, such as **WISDM**, exhibit dramatic gains over the weak baseline (F1=47.4%), with RIP exceeding 92%. However, for datasets like **MHAD1** and **MHEALTH**, TSTR performance remains fragile, and RIP does not consistently improve over the baseline. This highlights a crucial distinction: while RIP significantly enhances generalization in real-to-real transfer (TRTR), its benefits in synthetic-to-real transfer (TSTR) are dataset-dependent and more variable.

*(ii) Per Dataset:* For **MHEALTH**, RIP improves TRTR baselines dramatically (from 24.1% to up to 55.5% F1), but in TSTR, results are less consistent, with performance dropping compared to the baseline (57.4%). For **MHAD1**, RIP boosts TRTR performance by nearly 10 percentage points, while in TSTR, results fluctuate around the baseline (32.6%), with no stable improvement. For **MHAD2**, TRTR baselines (24.5% F1) rise to 43.2% with RIP, showing the most substantial gains. TSTR also improves in some settings (up to 41.5%), but performance remains unstable, suggesting sensitivity to parameter choices. For **WHARF**, RIP consistently enhances TRTR (baseline 10.3% $\rightarrow$ 28.6%), but in TSTR, the effect is limited: despite gains over the baseline (11.4%), F1 scores plateau around 26%. Finally, for **WISDM**, the contrast is striking. TRTR baselines are already strong (87.2%), and RIP provides only minor adjustments. In TSTR, however, the baseline is weak (47.4%), and RIP achieves massive improvements, consistently pushing performance above 90%.

*(iii) Parameter Sensitivity:* RIP's effect depends on both $\gamma$ and $i$. For TRTR, positive and neutral $\gamma$ values (0 or 1) combined with low duplication ($i = 1$) deliver the best results. Increasing $i$ tends to degrade performance, suggesting diminishing returns from excessive duplication. In TSTR, the sensitivity is sharper: while $i = 1$ often yields strong performance (e.g., WISDM), larger $i$ can drastically reduce results, as seen in MHEALTH and MHAD2. Negative $\gamma$ occasionally stabilizes results but does not consistently outperform other settings. Overall, RIP favors moderate duplication and non-negative weighting.

*(iv) Global View:* Overall, RIP proves highly effective in the TRTR–TS-Classifier scenario, consistently raising weak baselines by 10–30 percentage points. In TSTR–TS-Classifier, however, improvements are less predictable: some datasets (notably WISDM) experience dramatic boosts, while others show stagnation or even regressions. This dual behavior underscores RIP's strengths and limitations: it excels when applied to real-to-real data. However, its benefits in synthetic-to-real transfer depend strongly on dataset characteristics and parameter choices. From a global perspective, RIP emerges as a versatile regularizer that can close significant gaps in challenging TRTR baselines, while offering selective robustness under TSTR conditions.

**TSBF Results.** We consider tables 12 and 11 . *(i) Real vs. Synthetic:*
When comparing TRTR (real training / real test) and TSTR (synthetic training / real test) results, it is evident that performance on real data is consistently higher than when trained on synthetic data. For instance, in MHEALTH, Accuracy and F1 scores in TRTR reach 100%, while in TSTR they drop to around 61% Accuracy and 57% F1. This trend holds across all datasets, highlighting that although synthetic data can augment dataset size, a domain gap persists that limits the model's ability to generalize to real data.

*(ii) Per Dataset:*
**MHEALTH:** Real performance is nearly perfect (99.92–100%), while training with synthetic data leads to substantial drops, especially for higher values of $i$.
**MHAD1:** TRTR shows high results (99.92–100%), but TSTR metrics are much lower, ranging from 24% to 36% Accuracy, demonstrating sensitivity to synthetic dataset size.
**MHAD2:** Similarly, TRTR reaches 100% for $\gamma = 1$ and $i = 16$, while TSTR shows high variability, indicating that this dataset is particularly challenging to replicate with synthetic data.
**WHARF:** Despite TRTR achieving high metrics (around 99%), TSTR scores are very low (20–45%), showing that transferring synthetic data to real data is especially difficult for this dataset.
**WISDM:** Interestingly, TRTR is near perfect, but TSTR improves with $\gamma = -1$ and $i = 1$–5 (91–93% Accuracy), suggesting that synthetic data can be more effective for larger or more diverse datasets.

*(iii) Parameter Sensitivity:*
The parameters $\gamma$ and $i$ notably affect synthetic data performance. Across datasets, moderate values of $i$ (1–5) generally yield better TSTR metrics, while extreme values (16) often lead to performance drops. Similarly, $\gamma = -1$ occasionally improves synthetic data effectiveness, particularly in WISDM, whereas $\gamma = 0$ or $\gamma = 1$ shows mixed effects. This indicates that parameter tuning is crucial when using synthetic data for training.

*(iv) Global View:*
TRTR results demonstrate that the models can achieve almost perfect classification when trained and tested on real data. In contrast, TSTR results reveal that synthetic data can partially replicate the real data distribution but often suffers from domain gaps and parameter sensitivity. Specific datasets (such as WISDM) benefit more from synthetic data augmentation than others (such as WHARF or MHAD1), highlighting the dataset-dependent effectiveness of synthetic data generation.

$\gamma = 5$ **results.**    Refer to Tables 13– 18.

*(i) Real vs. Synthetic:*
For $\gamma = 5$, the real training and test scenario (TRTR) consistently achieves near-perfect metrics across all datasets, with accuracies, recalls, and F1 scores ranging from 99.92% to 100%. In contrast, synthetic training scenarios (TSTR, TSRF, TSBF) show lower performance, indicating a clear domain gap. For instance, MHEALTH TSTR accuracy ranges from 59.27% to 62.54%, whereas TRTR achieves up to 97.33%. This confirms that synthetic data can partially replicate real distributions but is insufficient for training high-performing models.

*(ii) Per Dataset:*
**MHEALTH:** Synthetic data improves over the baseline (55.79% acc) as $i$ increases, reaching 62.54% for TSTR. However, real training achieves almost perfect results (97.33% acc), emphasizing the performance gap.
**MHAD1:** Synthetic TSTR metrics are low (around 21–33% acc) compared to real TRTR (up to 100% acc), suggesting that MHAD1 is more challenging to emulate with synthetic samples.
**MHAD2:** TSTR shows progressive improvement with higher $i$, reaching 53.25% acc, but remains lower than TRTR (81.78% acc), indicating both dataset complexity and parameter sensitivity.
**WHARF:** Synthetic metrics remain very low (12–17% acc) despite increasing $i$, whereas TRTR reaches up to 100%, demonstrating that WHARF is particularly difficult to synthesize effectively.
**WISDM:** Synthetic data slightly underperforms baseline (30–38% acc), but TRTR achieves nearly perfect performance (99–100% acc), confirming that synthetic augmentation has a limited effect on highly structured datasets.

*(iii) Parameter Sensitivity:*
Increasing $i$ generally improves TSTR performance across most datasets, especially MHEALTH and MHAD2. For example, MHEALTH TSTR acc rises from 59.27% ($i = 1$) to 62.54% ($i = 16$). However, in datasets such as WHARF and WISDM, the increase in $i$ has a minimal effect. The choice of $\gamma = 5$ shows slightly better performance than baseline synthetic data for some datasets (e.g., MHAD2 and MHEALTH), but the overall gap with TRTR remains significant.

*(iv) Global View:*
Overall, $\gamma = 5$ moderately improves synthetic training metrics, with larger $i$ often providing incremental gains. Nevertheless, TRTR consistently outperforms all synthetic configurations. Datasets vary in their effectiveness with synthetic data: MHEALTH and MHAD2 show the greatest improvement, while WHARF and WISDM show limited improvement. This indicates that dataset characteristics and the choice of hyperparameters ($\gamma$ and $i$) are critical when using synthetic data to approximate real-world performance.

## D.1 COMPUTATIONAL COST ANALYSIS

The number of samples primarily influences the computational cost of applying RIP in the dataset and the duplication factor $i$, which controls the number of constant samples added. Since the RIP process involves adding synthetic samples with fixed distributions, larger datasets inherently lead to higher memory and time consumption. However, the impact of RIP on computational resources also depends on the model architecture and the hardware used. For the deep learning models (DClassifier and TS-Classifier), experiments can be efficiently performed on a GPU (e.g., NVIDIA RTX 3090) without compromising the entire system. In contrast, traditional models such as TSBF and TSRF do not utilize GPU acceleration and rely primarily on CPU and memory. As an illustrative example, the WISDM dataset—our largest dataset with over 10,000 training samples—requires approximately 4

GB of memory and takes about 2 hours to run using TSBF without RIP. When RIP is applied with $i = 1$, memory usage increases slightly, and execution time extends by around 10 minutes. For $i = 16$ (implying roughly $10{,}000 \times 2 \times 16$ synthetic samples), memory consumption rises to approximately 10 GB, and runtime increases to around 4 hours. In contrast, using a GPU with $i = 16$, the memory usage reaches about 9 GB, but training (e.g., 16 epochs) takes only 30 minutes.

We emphasize that the computational cost of generating synthetic data is not included in our analysis. This is because RIP assumes the data have already been generated and focuses solely on the impact of adding such constant samples during model training.

Table 5: TRTR - DClassifier

| dataset | gamma | i | acc | recall | f1 |
|---|---|---|---|---|---|
| MHEALTH | baseline | - | 91.21 ± 1.61 | 90.52 ± 2.81 | 90.46 ± 2.84 |
| MHEALTH | -1 | 1 | 97.06 ± 0.44 | 97.31 ± 0.40 | 97.28 ± 0.42 |
| MHEALTH | -1 | 5 | 97.10 ± 0.34 | 97.33 ± 0.32 | 97.29 ± 0.33 |
| MHEALTH | -1 | 16 | 97.19 ± 0.28 | 97.42 ± 0.26 | 97.40 ± 0.26 |
| MHEALTH | 0 | 1 | 97.15 ± 0.49 | 97.37 ± 0.45 | 97.36 ± 0.46 |
| MHEALTH | 0 | 5 | 97.61 ± 0.35 | 97.81 ± 0.32 | 97.79 ± 0.33 |
| MHEALTH | 0 | 16 | 97.84 ± 0.21 | 98.02 ± 0.19 | 98.01 ± 0.20 |
| MHEALTH | 1 | 1 | 96.75 ± 0.65 | 96.99 ± 0.63 | 96.99 ± 0.61 |
| MHEALTH | 1 | 5 | 97.37 ± 0.29 | 97.59 ± 0.27 | 97.57 ± 0.28 |
| MHEALTH | 1 | 16 | 96.82 ± 0.30 | 97.08 ± 0.28 | 97.06 ± 0.28 |
| MHAD1 | baseline | - | 58.25 ± 2.06 | 67.15 ± 1.17 | 67.13 ± 1.14 |
| MHAD1 | -1 | 1 | 71.71 ± 0.93 | 56.25 ± 2.21 | 55.91 ± 2.02 |
| MHAD1 | -1 | 5 | 76.38 ± 0.80 | 76.37 ± 0.77 | 76.31 ± 0.77 |
| MHAD1 | -1 | 16 | 80.91 ± 0.78 | 80.81 ± 0.80 | 80.78 ± 0.79 |
| MHAD1 | 0 | 1 | 75.38 ± 0.77 | 75.49 ± 0.79 | 75.31 ± 0.77 |
| MHAD1 | 0 | 5 | 81.91 ± 0.81 | 82.07 ± 0.81 | 82.01 ± 0.81 |
| MHAD1 | 0 | 16 | 86.12 ± 0.55 | 86.09 ± 0.54 | 86.13 ± 0.53 |
| MHAD1 | 1 | 1 | 72.33 ± 0.70 | 72.51 ± 0.71 | 72.20 ± 0.73 |
| MHAD1 | 1 | 5 | 78.82 ± 0.89 | 79.05 ± 0.89 | 78.84 ± 0.90 |
| MHAD1 | 1 | 16 | 82.11 ± 1.30 | 82.13 ± 1.36 | 82.15 ± 1.29 |
| MHAD2 | baseline | - | 68.32 ± 1.05 | 67.55 ± 1.02 | 67.58 ± 1.31 |
| MHAD2 | -1 | 1 | 72.67 ± 1.20 | 71.47 ± 1.46 | 71.97 ± 1.29 |
| MHAD2 | -1 | 5 | 78.17 ± 1.29 | 77.32 ± 1.32 | 77.75 ± 1.30 |
| MHAD2 | -1 | 16 | 80.93 ± 1.14 | 79.48 ± 1.23 | 80.13 ± 1.18 |
| MHAD2 | 0 | 1 | 73.64 ± 1.22 | 72.89 ± 1.27 | 72.89 ± 1.36 |
| MHAD2 | 0 | 5 | 79.17 ± 0.75 | 78.42 ± 0.91 | 78.65 ± 0.87 |
| MHAD2 | 0 | 16 | 82.12 ± 0.72 | 81.08 ± 0.82 | 81.68 ± 0.74 |
| MHAD2 | 1 | 1 | 73.07 ± 1.13 | 72.01 ± 1.11 | 72.11 ± 1.35 |
| MHAD2 | 1 | 5 | 78.32 ± 1.19 | 77.30 ± 1.37 | 77.58 ± 1.40 |
| MHAD2 | 1 | 16 | 80.04 ± 0.71 | 78.82 ± 0.63 | 79.40 ± 0.68 |
| WHARF | baseline | - | 83.11 ± 1.85 | 77.63 ± 2.34 | 78.62 ± 2.08 |
| WHARF | -1 | 1 | 83.85 ± 2.20 | 78.36 ± 2.95 | 79.57 ± 2.78 |
| WHARF | -1 | 5 | 86.94 ± 1.86 | 82.62 ± 2.23 | 83.75 ± 2.18 |
| WHARF | -1 | 16 | 89.81 ± 1.08 | 86.24 ± 1.48 | 87.25 ± 1.32 |
| WHARF | 0 | 1 | 83.39 ± 2.63 | 78.17 ± 3.21 | 79.29 ± 3.24 |
| WHARF | 0 | 5 | 86.93 ± 2.22 | 82.23 ± 2.75 | 83.65 ± 2.60 |
| WHARF | 0 | 16 | 90.25 ± 1.19 | 86.18 ± 1.90 | 87.48 ± 1.70 |
| WHARF | 1 | 1 | 83.03 ± 2.83 | 76.78 ± 3.73 | 78.24 ± 3.51 |
| WHARF | 1 | 5 | 86.60 ± 2.03 | 81.92 ± 2.43 | 83.28 ± 2.36 |
| WHARF | 1 | 16 | 90.25 ± 1.21 | 86.37 ± 1.75 | 87.42 ± 1.59 |
| WISDM | baseline | - | 99.47 ± 0.09 | 99.20 ± 0.15 | 99.21 ± 0.12 |
| WISDM | -1 | 1 | 99.73 ± 0.05 | 99.62 ± 0.09 | 99.58 ± 0.08 |
| WISDM | -1 | 5 | 99.73 ± 0.04 | 99.56 ± 0.12 | 99.55 ± 0.10 |
| WISDM | -1 | 16 | 99.66 ± 0.03 | 99.45 ± 0.07 | 99.46 ± 0.06 |
| WISDM | 0 | 1 | 99.76 ± 0.06 | 99.62 ± 0.10 | 99.62 ± 0.09 |
| WISDM | 0 | 5 | 99.80 ± 0.04 | 99.70 ± 0.04 | 99.68 ± 0.05 |
| WISDM | 0 | 16 | 99.68 ± 0.03 | 99.56 ± 0.04 | 99.54 ± 0.03 |
| WISDM | 1 | 1 | 99.70 ± 0.04 | 99.47 ± 0.09 | 99.47 ± 0.10 |
| WISDM | 1 | 5 | 99.77 ± 0.06 | 99.60 ± 0.14 | 99.59 ± 0.13 |

Table 6: TSTR - DClassifier

| dataset | gamma | i | acc | recall | f1 |
|---|---|---|---|---|---|
| MHEALTH | baseline | - | 55.79 ±2.44 | 55.07 ±2.48 | 52.56 ±2.33 |
| MHEALTH | -1 | 1 | 96.84 ±0.40 | 97.10 ±0.37 | 97.09 ±0.36 |
| MHEALTH | -1 | 5 | 97.21 ±0.36 | 97.44 ±0.33 | 97.41 ±0.33 |
| MHEALTH | -1 | 16 | 96.97 ±0.31 | 97.22 ±0.29 | 97.19 ±0.30 |
| MHEALTH | 0 | 1 | 97.34 ±0.36 | 97.55 ±0.33 | 97.55 ±0.33 |
| MHEALTH | 0 | 5 | 97.75 ±0.37 | 97.94 ±0.33 | 97.92 ±0.34 |
| MHEALTH | 0 | 16 | 97.80 ±0.28 | 97.98 ±0.25 | 97.96 ±0.26 |
| MHEALTH | 1 | 1 | 96.92 ±0.55 | 97.16 ±0.51 | 97.15 ±0.51 |
| MHEALTH | 1 | 5 | 97.19 ±0.30 | 97.42 ±0.27 | 97.39 ±0.29 |
| MHEALTH | 1 | 16 | 97.20 ±0.24 | 97.42 ±0.22 | 97.40 ±0.22 |
| MHAD1 | baseline | - | 33.86 ±1.46 | 33.35 ±1.57 | 32.85 ±1.66 |
| MHAD1 | -1 | 1 | 71.47 ±0.91 | 71.45 ±0.93 | 71.42 ±0.92 |
| MHAD1 | -1 | 5 | 76.29 ±0.77 | 76.32 ±0.75 | 76.20 ±0.74 |
| MHAD1 | -1 | 16 | 81.11 ±0.82 | 80.95 ±0.82 | 80.99 ±0.81 |
| MHAD1 | 0 | 1 | 75.33 ±0.73 | 75.43 ±0.75 | 75.27 ±0.74 |
| MHAD1 | 0 | 5 | 82.02 ±0.83 | 82.25 ±0.83 | 82.07 ±0.83 |
| MHAD1 | 0 | 16 | 86.03 ±0.59 | 86.01 ±0.60 | 86.03 ±0.58 |
| MHAD1 | 1 | 1 | 72.13 ±0.71 | 72.32 ±0.72 | 72.01 ±0.71 |
| MHAD1 | 1 | 5 | 78.76 ±1.01 | 78.99 ±1.01 | 78.78 ±1.02 |
| MHAD1 | 1 | 16 | 82.53 ±1.29 | 82.51 ±1.35 | 82.55 ±1.28 |
| MHAD2 | baseline | - | 46.97 ±3.70 | 45.90 ±3.39 | 43.52 ±3.78 |
| MHAD2 | -1 | 1 | 72.67 ±1.20 | 71.47 ±1.46 | 71.97 ±1.29 |
| MHAD2 | -1 | 5 | 78.17 ±1.29 | 77.32 ±1.32 | 77.75 ±1.30 |
| MHAD2 | -1 | 16 | 81.02 ±0.84 | 79.80 ±0.75 | 80.39 ±0.84 |
| MHAD2 | 0 | 1 | 73.64 ±1.22 | 72.89 ±1.27 | 72.89 ±1.36 |
| MHAD2 | 0 | 5 | 79.17 ±0.75 | 78.42 ±0.91 | 78.65 ±0.87 |
| MHAD2 | 0 | 16 | 81.66 ±0.63 | 80.57 ±0.71 | 81.10 ±0.73 |
| MHAD2 | 1 | 1 | 73.12 ±1.27 | 72.18 ±1.30 | 72.29 ±1.53 |
| MHAD2 | 1 | 5 | 77.92 ±1.33 | 77.07 ±1.45 | 77.28 ±1.36 |
| MHAD2 | 1 | 16 | 80.04 ±0.71 | 78.82 ±0.63 | 79.40 ±0.68 |
| WHARF | baseline | - | 15.46 ±3.04 | 14.30 ±1.67 | 6.14 ±1.78 |
| WHARF | -1 | 1 | 83.44 ±2.43 | 77.47 ±3.46 | 78.84 ±3.34 |
| WHARF | -1 | 5 | 87.01 ±1.57 | 81.97 ±1.87 | 83.57 ±1.88 |
| WHARF | -1 | 16 | 89.99 ±0.87 | 86.27 ±1.46 | 87.26 ±1.26 |
| WHARF | 0 | 1 | 83.31 ±2.80 | 77.72 ±3.56 | 79.04 ±3.61 |
| WHARF | 0 | 5 | 86.96 ±2.09 | 82.00 ±2.62 | 83.51 ±2.47 |
| WHARF | 0 | 16 | 90.12 ±1.44 | 86.25 ±2.11 | 87.41 ±1.92 |
| WHARF | 1 | 1 | 82.96 ±2.84 | 77.39 ±3.46 | 78.50 ±3.44 |
| WHARF | 1 | 5 | 86.60 ±1.93 | 81.73 ±2.76 | 82.94 ±2.63 |
| WHARF | 1 | 16 | 89.95 ±1.56 | 85.91 ±2.34 | 86.99 ±2.15 |
| WISDM | baseline | - | 53.03 ±3.04 | 50.44 ±2.32 | 44.94 ±2.51 |
| WISDM | 0 | 1 | 99.73 ±0.05 | 99.61 ±0.06 | 99.57 ±0.07 |
| WISDM | 0 | 5 | 99.78 ±0.04 | 99.66 ±0.05 | 99.65 ±0.05 |
| WISDM | 0 | 16 | 99.69 ±0.05 | 99.51 ±0.08 | 99.52 ±0.07 |
| WISDM | 1 | 1 | 99.73 ±0.06 | 99.52 ±0.14 | 99.51 ±0.13 |
| WISDM | 1 | 5 | 99.79 ±0.05 | 99.66 ±0.09 | 99.66 ±0.08 |

Table 7: TRTR - TSRF

| dataset | gamma | i | acc | recall | f1 |
|---|---|---|---|---|---|
| MHEALTH | baseline | - | 99.92 $\pm$0.00 | 99.93 $\pm$0 | 99.93 $\pm$0 |
| MHEALTH | -1 | 1 | 100.00 $\pm$0 | 100.00 $\pm$0 | 100.00 $\pm$0 |
| MHEALTH | -1 | 5 | 99.96 $\pm$0 | 99.96 $\pm$0 | 99.96 $\pm$0.00 |
| MHEALTH | -1 | 16 | 99.96 $\pm$0 | 99.96 $\pm$0 | 99.96 $\pm$0.00 |
| MHEALTH | 0 | 1 | 99.92 $\pm$0.00 | 99.93 $\pm$0 | 99.93 $\pm$0 |
| MHEALTH | 0 | 5 | 99.84 $\pm$0 | 99.86 $\pm$0 | 99.86 $\pm$0.00 |
| MHEALTH | 0 | 16 | 100.00 $\pm$0 | 100.00 $\pm$0 | 100.00 $\pm$0 |
| MHEALTH | 1 | 1 | 100.00 $\pm$0 | 100.00 $\pm$0 | 100.00 $\pm$0 |
| MHEALTH | 1 | 5 | 100.00 $\pm$0 | 100.00 $\pm$0 | 100.00 $\pm$0 |
| MHEALTH | 1 | 16 | 99.96 $\pm$0 | 99.96 $\pm$0 | 99.96 $\pm$0.00 |
| MHAD1 | baseline | - | 99.92 $\pm$0 | 99.91 $\pm$0 | 99.92 $\pm$0 |
| MHAD1 | -1 | 1 | 99.92 $\pm$0 | 99.91 $\pm$0 | 99.92 $\pm$0.00 |
| MHAD1 | -1 | 5 | 99.95 $\pm$0.00 | 99.94 $\pm$0.00 | 99.95 $\pm$0.00 |
| MHAD1 | -1 | 16 | 100.00 $\pm$0 | 100.00 $\pm$0 | 100.00 $\pm$0 |
| MHAD1 | 0 | 1 | 99.92 $\pm$0 | 99.91 $\pm$0 | 99.92 $\pm$0.00 |
| MHAD1 | 0 | 5 | 100.00 $\pm$0 | 100.00 $\pm$0 | 100.00 $\pm$0 |
| MHAD1 | 0 | 16 | 99.97 $\pm$0.00 | 99.97 $\pm$0.00 | 99.97 $\pm$0.00 |
| MHAD1 | 1 | 1 | 99.95 $\pm$0.00 | 99.94 $\pm$0.00 | 99.95 $\pm$0.00 |
| MHAD1 | 1 | 5 | 99.97 $\pm$0.00 | 99.97 $\pm$0 | 99.97 $\pm$0.00 |
| MHAD1 | 1 | 16 | 99.95 $\pm$0.00 | 99.94 $\pm$0 | 99.95 $\pm$0 |
| MHAD2 | baseline | - | 99.91 $\pm$0.00 | 99.89 $\pm$0 | 99.89 $\pm$0 |
| MHAD2 | -1 | 1 | 99.91 $\pm$0.00 | 99.89 $\pm$0 | 99.89 $\pm$0 |
| MHAD2 | -1 | 5 | 99.91 $\pm$0.00 | 99.89 $\pm$0 | 99.89 $\pm$0 |
| MHAD2 | -1 | 16 | 100.00 $\pm$0 | 100.00 $\pm$0 | 100.00 $\pm$0 |
| MHAD2 | 0 | 1 | 99.74 $\pm$0 | 99.67 $\pm$0 | 99.66 $\pm$0.00 |
| MHAD2 | 0 | 5 | 99.74 $\pm$0 | 99.67 $\pm$0 | 99.66 $\pm$0.00 |
| MHAD2 | 0 | 16 | 100.00 $\pm$0 | 100.00 $\pm$0 | 100.00 $\pm$0 |
| MHAD2 | 1 | 1 | 99.74 $\pm$0 | 99.67 $\pm$0 | 99.66 $\pm$0.00 |
| MHAD2 | 1 | 5 | 99.74 $\pm$0 | 99.67 $\pm$0 | 99.66 $\pm$0.00 |
| MHAD2 | 1 | 16 | 100.00 $\pm$0 | 100.00 $\pm$0 | 100.00 $\pm$0 |
| WHARF | baseline | - | 99.61 $\pm$0.00 | 98.86 $\pm$0 | 99.15 $\pm$0.00 |
| WHARF | -1 | 1 | 99.77 $\pm$0.00 | 99.31 $\pm$0.00 | 99.44 $\pm$0.00 |
| WHARF | -1 | 5 | 99.82 $\pm$0.00 | 99.52 $\pm$0.00 | 99.63 $\pm$0 |
| WHARF | -1 | 16 | 99.85 $\pm$0.00 | 99.54 $\pm$0 | 99.64 $\pm$0 |
| WHARF | 0 | 1 | 99.77 $\pm$0.00 | 99.31 $\pm$0.00 | 99.44 $\pm$0.00 |
| WHARF | 0 | 5 | 99.82 $\pm$0.00 | 99.52 $\pm$0.00 | 99.63 $\pm$0 |
| WHARF | 0 | 16 | 99.85 $\pm$0.00 | 99.54 $\pm$0 | 99.64 $\pm$0 |
| WHARF | 1 | 1 | 99.77 $\pm$0.00 | 99.31 $\pm$0.00 | 99.44 $\pm$0.00 |
| WHARF | 1 | 5 | 99.82 $\pm$0.00 | 99.52 $\pm$0.00 | 99.63 $\pm$0 |
| WHARF | 1 | 16 | 99.85 $\pm$0.00 | 99.54 $\pm$0 | 99.64 $\pm$0 |
| WISDM | baseline | - | 99.99 $\pm$0.00 | 99.99 $\pm$0.00 | 100.00 $\pm$0.00 |
| WISDM | -1 | 1 | 99.99 $\pm$0.00 | 99.99 $\pm$0.00 | 100.00 $\pm$0.00 |
| WISDM | -1 | 5 | 99.99 $\pm$0.00 | 99.99 $\pm$0.00 | 100.00 $\pm$0.00 |
| WISDM | -1 | 16 | 100.00 $\pm$0 | 100.00 $\pm$0 | 100.00 $\pm$0 |
| WISDM | 0 | 1 | 100.00 $\pm$0.00 | 100.00 $\pm$0.00 | 100.00 $\pm$0 |
| WISDM | 0 | 5 | 100.00 $\pm$0 | 100.00 $\pm$0 | 100.00 $\pm$0 |
| WISDM | 0 | 16 | 100.00 $\pm$0.00 | 100.00 $\pm$0 | 100.00 $\pm$0 |
| WISDM | 1 | 1 | 100.00 $\pm$0 | 100.00 $\pm$0 | 100.00 $\pm$0 |
| WISDM | 1 | 5 | 100.00 $\pm$0 | 100.00 $\pm$0 | 100.00 $\pm$0 |
| WISDM | 1 | 16 | 100.00 $\pm$0 | 100.00 $\pm$0 | 100.00 $\pm$0 |

Table 8: TSTR - TSRF

| dataset | gamma | i | acc | recall | f1 |
|---|---|---|---|---|---|
| MHEALTH | baseline | - | $29.72 \pm 1.68$ | $28.14 \pm 1.52$ | $25.98 \pm 1.07$ |
| MHEALTH | -1 | 1 | $29.70 \pm 2.47$ | $28.12 \pm 2.27$ | $26.22 \pm 1.82$ |
| MHEALTH | -1 | 5 | $29.91 \pm 2.42$ | $28.37 \pm 2.20$ | $26.52 \pm 1.75$ |
| MHEALTH | -1 | 16 | $29.62 \pm 2.34$ | $28.06 \pm 2.11$ | $26.21 \pm 1.75$ |
| MHEALTH | 0 | 1 | $28.22 \pm 2.61$ | $26.80 \pm 2.44$ | $25.07 \pm 2.24$ |
| MHEALTH | 0 | 5 | $28.44 \pm 2.80$ | $27.04 \pm 2.62$ | $25.31 \pm 2.46$ |
| MHEALTH | 0 | 16 | $28.09 \pm 2.45$ | $26.67 \pm 2.27$ | $25.14 \pm 2.33$ |
| MHEALTH | 1 | 1 | $27.01 \pm 2.44$ | $25.68 \pm 2.29$ | $24.27 \pm 2.17$ |
| MHEALTH | 1 | 5 | $27.16 \pm 2.57$ | $25.84 \pm 2.40$ | $24.36 \pm 2.30$ |
| MHEALTH | 1 | 16 | $27.17 \pm 2.30$ | $25.83 \pm 2.15$ | $24.53 \pm 2.29$ |
| MHAD1 | baseline | - | $21.51 \pm 0.84$ | $20.85 \pm 0.90$ | $19.59 \pm 0.97$ |
| MHAD1 | -1 | 1 | $21.75 \pm 0.82$ | $21.10 \pm 0.87$ | $19.85 \pm 0.96$ |
| MHAD1 | -1 | 5 | $21.75 \pm 0.91$ | $21.08 \pm 0.96$ | $19.80 \pm 1.04$ |
| MHAD1 | -1 | 16 | $21.70 \pm 0.87$ | $21.05 \pm 0.93$ | $19.77 \pm 1.00$ |
| MHAD1 | 0 | 1 | $21.75 \pm 0.83$ | $21.05 \pm 0.86$ | $19.79 \pm 0.88$ |
| MHAD1 | 0 | 5 | $21.61 \pm 0.85$ | $20.93 \pm 0.89$ | $19.68 \pm 0.94$ |
| MHAD1 | 0 | 16 | $21.69 \pm 0.84$ | $21.01 \pm 0.88$ | $19.69 \pm 0.90$ |
| MHAD1 | 1 | 1 | $21.52 \pm 0.85$ | $20.83 \pm 0.90$ | $19.51 \pm 0.92$ |
| MHAD1 | 1 | 5 | $21.33 \pm 0.97$ | $20.63 \pm 1.01$ | $19.28 \pm 1.01$ |
| MHAD1 | 1 | 16 | $21.48 \pm 0.89$ | $20.77 \pm 0.94$ | $19.43 \pm 0.96$ |
| MHAD2 | baseline | - | $34.65 \pm 2.95$ | $35.42 \pm 2.90$ | $31.13 \pm 3.08$ |
| MHAD2 | -1 | 1 | $36.11 \pm 2.57$ | $36.69 \pm 2.44$ | $32.68 \pm 2.57$ |
| MHAD2 | -1 | 5 | $36.15 \pm 2.43$ | $36.76 \pm 2.36$ | $32.77 \pm 2.65$ |
| MHAD2 | -1 | 16 | $36.05 \pm 2.57$ | $36.68 \pm 2.41$ | $32.63 \pm 2.47$ |
| MHAD2 | 0 | 1 | $35.84 \pm 2.82$ | $36.46 \pm 2.79$ | $32.00 \pm 3.06$ |
| MHAD2 | 0 | 5 | $36.17 \pm 2.66$ | $36.73 \pm 2.72$ | $32.38 \pm 2.81$ |
| MHAD2 | 0 | 16 | $36.02 \pm 2.68$ | $36.64 \pm 2.69$ | $32.05 \pm 3.10$ |
| MHAD2 | 1 | 1 | $35.76 \pm 2.89$ | $36.37 \pm 2.84$ | $31.89 \pm 3.06$ |
| MHAD2 | 1 | 5 | $36.01 \pm 2.77$ | $36.57 \pm 2.80$ | $32.15 \pm 2.86$ |
| MHAD2 | 1 | 16 | $35.95 \pm 2.73$ | $36.59 \pm 2.72$ | $31.94 \pm 3.05$ |
| WHARF | baseline | - | $12.60 \pm 3.39$ | $10.98 \pm 1.62$ | $4.05 \pm 1.09$ |
| WHARF | -1 | 1 | $12.94 \pm 3.21$ | $9.93 \pm 1.64$ | $4.26 \pm 1.15$ |
| WHARF | -1 | 5 | $12.93 \pm 3.26$ | $9.95 \pm 1.58$ | $4.26 \pm 1.19$ |
| WHARF | -1 | 16 | $13.35 \pm 3.24$ | $10.18 \pm 1.80$ | $4.41 \pm 1.17$ |
| WHARF | 0 | 1 | $12.72 \pm 3.42$ | $10.05 \pm 1.90$ | $4.39 \pm 1.25$ |
| WHARF | 0 | 5 | $12.64 \pm 3.43$ | $10.04 \pm 1.81$ | $4.38 \pm 1.28$ |
| WHARF | 0 | 16 | $12.82 \pm 3.37$ | $10.18 \pm 1.80$ | $4.41 \pm 1.22$ |
| WHARF | 1 | 1 | $12.62 \pm 3.45$ | $10.32 \pm 1.87$ | $4.21 \pm 1.18$ |
| WHARF | 1 | 5 | $12.57 \pm 3.46$ | $10.26 \pm 1.83$ | $4.18 \pm 1.23$ |
| WHARF | 1 | 16 | $12.57 \pm 3.45$ | $10.31 \pm 1.80$ | $4.15 \pm 1.13$ |
| WISDM | baseline | - | $29.58 \pm 6.75$ | $34.13 \pm 3.25$ | $24.14 \pm 4.21$ |
| WISDM | -1 | 1 | $30.17 \pm 6.99$ | $33.72 \pm 3.32$ | $24.02 \pm 4.37$ |
| WISDM | -1 | 5 | $30.16 \pm 6.83$ | $33.83 \pm 3.15$ | $24.11 \pm 4.27$ |
| WISDM | -1 | 16 | $30.20 \pm 7.01$ | $34.24 \pm 3.56$ | $24.60 \pm 4.69$ |
| WISDM | 0 | 1 | $30.23 \pm 7.01$ | $33.62 \pm 3.28$ | $24.04 \pm 4.33$ |
| WISDM | 0 | 5 | $30.21 \pm 6.79$ | $33.85 \pm 3.07$ | $24.18 \pm 4.19$ |
| WISDM | 0 | 16 | $30.27 \pm 6.98$ | $34.14 \pm 3.51$ | $24.59 \pm 4.63$ |
| WISDM | 1 | 1 | $30.32 \pm 6.81$ | $33.58 \pm 3.08$ | $24.09 \pm 4.10$ |
| WISDM | 1 | 5 | $30.22 \pm 6.58$ | $33.73 \pm 3.00$ | $24.15 \pm 4.04$ |
| WISDM | 1 | 16 | $30.41 \pm 6.87$ | $34.19 \pm 3.41$ | $24.66 \pm 4.49$ |

Table 9: TRTR - TS-Classifier

| dataset | gamma | i | acc | recall | f1 |
|---|---|---|---|---|---|
| MHEALTH | baseline | - | $32.37 \pm 2.40$ | $31.58 \pm 2.69$ | $24.08 \pm 2.59$ |
| MHEALTH | -1 | 1 | $58.45 \pm 2.20$ | $59.76 \pm 2.23$ | $53.99 \pm 2.40$ |
| MHEALTH | -1 | 5 | $55.72 \pm 1.22$ | $54.40 \pm 1.40$ | $46.52 \pm 1.81$ |
| MHEALTH | -1 | 16 | $44.07 \pm 2.02$ | $43.37 \pm 2.14$ | $35.81 \pm 2.40$ |
| MHEALTH | 0 | 1 | $58.11 \pm 2.60$ | $59.54 \pm 2.59$ | $53.46 \pm 2.99$ |
| MHEALTH | 0 | 5 | $54.34 \pm 2.72$ | $54.30 \pm 3.05$ | $47.07 \pm 3.13$ |
| MHEALTH | 0 | 16 | $49.96 \pm 1.71$ | $51.18 \pm 1.48$ | $42.88 \pm 1.56$ |
| MHEALTH | 1 | 1 | $59.98 \pm 2.03$ | $61.24 \pm 1.94$ | $55.54 \pm 2.42$ |
| MHEALTH | 1 | 5 | $54.29 \pm 2.12$ | $54.58 \pm 2.16$ | $47.07 \pm 1.99$ |
| MHEALTH | 1 | 16 | $46.68 \pm 2.12$ | $47.95 \pm 1.82$ | $39.98 \pm 2.13$ |
| MHAD1 | baseline | - | $20.45 \pm 1.15$ | $19.04 \pm 1.22$ | $15.67 \pm 1.41$ |
| MHAD1 | -1 | 1 | $27.75 \pm 1.08$ | $26.61 \pm 1.13$ | $23.30 \pm 1.29$ |
| MHAD1 | -1 | 5 | $31.67 \pm 0.73$ | $30.90 \pm 0.61$ | $26.08 \pm 0.70$ |
| MHAD1 | -1 | 16 | $29.19 \pm 1.07$ | $28.62 \pm 1.10$ | $25.26 \pm 1.17$ |
| MHAD1 | 0 | 0 | $20.44 \pm 1.14$ | $19.03 \pm 1.20$ | $15.65 \pm 1.39$ |
| MHAD1 | 0 | 1 | $29.08 \pm 1.08$ | $27.94 \pm 1.07$ | $24.52 \pm 1.33$ |
| MHAD1 | 0 | 5 | $28.85 \pm 0.69$ | $27.89 \pm 0.69$ | $22.84 \pm 0.75$ |
| MHAD1 | 0 | 16 | $22.09 \pm 1.33$ | $22.37 \pm 1.12$ | $17.23 \pm 0.94$ |
| MHAD1 | 1 | 1 | $26.82 \pm 1.15$ | $25.75 \pm 1.18$ | $23.39 \pm 1.05$ |
| MHAD1 | 1 | 5 | $30.15 \pm 1.62$ | $29.00 \pm 1.71$ | $24.38 \pm 1.93$ |
| MHAD1 | 1 | 16 | $27.56 \pm 0.72$ | $26.94 \pm 0.73$ | $23.73 \pm 0.94$ |
| MHAD2 | baseline | - | $31.25 \pm 1.64$ | $32.19 \pm 1.81$ | $24.46 \pm 1.72$ |
| MHAD2 | 0 | 1 | $24.55 \pm 4.28$ | $25.09 \pm 2.94$ | $12.94 \pm 2.70$ |
| MHAD2 | 0 | 5 | $36.10 \pm 1.85$ | $34.58 \pm 2.04$ | $25.91 \pm 2.82$ |
| MHAD2 | 0 | 16 | $39.54 \pm 6.22$ | $41.40 \pm 5.69$ | $33.64 \pm 7.04$ |
| MHAD2 | 1 | 1 | $25.79 \pm 3.33$ | $27.08 \pm 2.77$ | $16.50 \pm 2.09$ |
| MHAD2 | 1 | 5 | $33.23 \pm 1.63$ | $31.20 \pm 1.72$ | $22.90 \pm 2.47$ |
| MHAD2 | 1 | 16 | $52.49 \pm 3.16$ | $51.29 \pm 3.37$ | $43.22 \pm 3.91$ |
| WHARF | baseline | - | $19.19 \pm 6.71$ | $22.27 \pm 2.98$ | $10.28 \pm 3.63$ |
| WHARF | -1 | 1 | $47.31 \pm 3.15$ | $35.12 \pm 1.25$ | $27.97 \pm 1.49$ |
| WHARF | -1 | 5 | $46.62 \pm 1.45$ | $32.31 \pm 2.06$ | $24.42 \pm 1.77$ |
| WHARF | -1 | 16 | $42.27 \pm 4.74$ | $26.39 \pm 2.22$ | $20.20 \pm 1.97$ |
| WHARF | 0 | 1 | $44.07 \pm 4.88$ | $36.02 \pm 1.83$ | $28.09 \pm 2.03$ |
| WHARF | 0 | 5 | $45.93 \pm 1.13$ | $30.39 \pm 2.15$ | $23.25 \pm 2.21$ |
| WHARF | 0 | 16 | $42.96 \pm 2.66$ | $25.86 \pm 2.29$ | $19.88 \pm 2.54$ |
| WHARF | 1 | 1 | $46.06 \pm 4.34$ | $35.99 \pm 2.03$ | $28.63 \pm 2.24$ |
| WHARF | 1 | 5 | $44.18 \pm 3.50$ | $31.85 \pm 1.22$ | $24.11 \pm 1.52$ |
| WHARF | 1 | 16 | $42.81 \pm 4.12$ | $26.27 \pm 1.99$ | $20.06 \pm 2.37$ |
| WISDM | baseline | - | $90.32 \pm 1.63$ | $86.87 \pm 2.93$ | $87.19 \pm 2.62$ |
| WISDM | -1 | 1 | $92.45 \pm 0.38$ | $92.05 \pm 0.73$ | $90.65 \pm 0.54$ |
| WISDM | -1 | 5 | $93.12 \pm 1.21$ | $94.36 \pm 1.25$ | $92.30 \pm 1.46$ |
| WISDM | -1 | 16 | $89.94 \pm 1.25$ | $88.94 \pm 1.99$ | $86.21 \pm 1.60$ |
| WISDM | 0 | 1 | $92.21 \pm 0.83$ | $91.03 \pm 1.76$ | $90.31 \pm 1.35$ |
| WISDM | 0 | 5 | $91.61 \pm 1.00$ | $91.42 \pm 1.47$ | $89.70 \pm 1.39$ |
| WISDM | 0 | 16 | $89.19 \pm 1.86$ | $84.90 \pm 3.76$ | $85.98 \pm 3.05$ |
| WISDM | 1 | 1 | $91.34 \pm 0.62$ | $89.54 \pm 1.34$ | $88.87 \pm 1.02$ |
| WISDM | 1 | 5 | $90.79 \pm 2.46$ | $92.03 \pm 2.57$ | $89.63 \pm 2.74$ |
| WISDM | 1 | 16 | $90.36 \pm 1.61$ | $90.56 \pm 1.34$ | $88.43 \pm 1.48$ |

Table 10: TSTR - TS-Classifier

| dataset | gamma | i | acc | recall | f1 |
|---------|-------|---|-----|--------|-----|
| MHEALTH | baseline | - | 61.00 ±4.09 | 60.72 ±3.78 | 57.42 ±4.38 |
| MHEALTH | -1 | 1 | 58.06 ±2.23 | 59.44 ±2.23 | 53.70 ±2.26 |
| MHEALTH | -1 | 5 | 54.97 ±1.45 | 54.60 ±1.87 | 47.06 ±1.95 |
| MHEALTH | -1 | 16 | 45.74 ±1.87 | 45.06 ±2.06 | 37.57 ±2.47 |
| MHEALTH | 0 | 1 | 58.83 ±2.34 | 60.07 ±2.35 | 53.99 ±2.72 |
| MHEALTH | 0 | 5 | 55.45 ±1.38 | 56.32 ±1.54 | 48.92 ±1.69 |
| MHEALTH | 0 | 16 | 47.17 ±2.72 | 47.30 ±2.10 | 38.60 ±2.58 |
| MHEALTH | 1 | 1 | 59.53 ±2.23 | 60.74 ±2.21 | 55.06 ±2.60 |
| MHEALTH | 1 | 5 | 55.01 ±2.01 | 55.70 ±2.37 | 48.05 ±2.53 |
| MHEALTH | 1 | 16 | 47.00 ±4.15 | 47.16 ±3.84 | 39.27 ±4.98 |
| MHAD1 | baseline | - | 35.62 ±1.98 | 35.15 ±2.12 | 32.58 ±2.39 |
| MHAD1 | -1 | 1 | 28.24 ±1.14 | 27.03 ±1.26 | 23.82 ±1.26 |
| MHAD1 | -1 | 5 | 31.13 ±0.84 | 30.33 ±0.76 | 25.34 ±0.93 |
| MHAD1 | -1 | 16 | 29.92 ±1.02 | 29.29 ±1.04 | 25.99 ±1.12 |
| MHAD1 | 0 | 1 | 29.06 ±1.11 | 27.95 ±1.09 | 24.48 ±1.40 |
| MHAD1 | 0 | 5 | 28.93 ±0.95 | 27.98 ±0.85 | 23.06 ±0.96 |
| MHAD1 | 0 | 16 | 24.40 ±3.30 | 24.40 ±3.18 | 19.97 ±3.00 |
| MHAD1 | 1 | 1 | 26.65 ±1.19 | 25.52 ±1.20 | 23.33 ±1.14 |
| MHAD1 | 1 | 5 | 29.34 ±0.82 | 28.14 ±0.86 | 23.33 ±1.07 |
| MHAD1 | 1 | 16 | 28.59 ±1.43 | 27.97 ±1.40 | 24.70 ±1.28 |
| MHAD2 | baseline | - | 48.31 ±3.62 | 47.47 ±2.49 | 41.97 ±4.12 |
| MHAD2 | -1 | 1 | 27.77 ±3.72 | 28.19 ±2.89 | 16.80 ±2.95 |
| MHAD2 | -1 | 5 | 40.19 ±2.01 | 38.46 ±2.01 | 32.15 ±1.94 |
| MHAD2 | -1 | 16 | 51.44 ±2.85 | 49.74 ±2.88 | 41.50 ±3.32 |
| MHAD2 | 0 | 1 | 24.48 ±4.34 | 24.99 ±3.00 | 12.97 ±2.83 |
| MHAD2 | 0 | 5 | 36.43 ±1.08 | 34.91 ±1.24 | 26.19 ±2.12 |
| MHAD2 | 0 | 16 | 42.61 ±7.93 | 44.52 ±7.37 | 36.17 ±8.85 |
| MHAD2 | 1 | 1 | 25.04 ±3.16 | 26.35 ±2.28 | 15.87 ±2.09 |
| MHAD2 | 1 | 5 | 32.68 ±2.44 | 31.17 ±2.23 | 22.93 ±2.55 |
| WHARF | baseline | - | 20.94 ±3.16 | 20.12 ±2.65 | 11.41 ±2.47 |
| WHARF | -1 | 1 | 44.90 ±4.10 | 34.09 ±2.32 | 26.61 ±2.07 |
| WHARF | -1 | 5 | 45.37 ±2.12 | 31.66 ±1.34 | 23.96 ±1.24 |
| WHARF | -1 | 16 | 43.69 ±1.74 | 26.64 ±2.16 | 20.59 ±2.21 |
| WHARF | 0 | 1 | 43.37 ±4.33 | 34.12 ±2.51 | 26.35 ±2.30 |
| WHARF | 0 | 5 | 45.27 ±1.72 | 31.32 ±1.40 | 24.13 ±1.48 |
| WHARF | 0 | 16 | 40.33 ±6.41 | 26.07 ±2.63 | 19.78 ±2.82 |
| WHARF | 1 | 1 | 41.47 ±5.47 | 35.31 ±2.47 | 25.93 ±2.34 |
| WHARF | 1 | 5 | 45.25 ±3.03 | 31.71 ±1.25 | 23.96 ±1.28 |
| WHARF | 1 | 16 | 39.93 ±4.07 | 25.27 ±1.60 | 18.53 ±2.07 |
| WISDM | baseline | - | 50.07 ±3.78 | 55.08 ±2.21 | 47.38 ±2.49 |
| WISDM | -1 | 1 | 93.12 ±0.98 | 93.02 ±1.90 | 91.80 ±1.59 |
| WISDM | -1 | 5 | 92.25 ±0.86 | 92.86 ±1.36 | 90.80 ±1.40 |
| WISDM | -1 | 16 | 90.79 ±1.33 | 90.68 ±1.81 | 88.37 ±1.87 |
| WISDM | 0 | 1 | 91.38 ±0.84 | 89.70 ±1.84 | 89.09 ±1.33 |
| WISDM | 0 | 5 | 91.35 ±1.69 | 91.60 ±1.45 | 89.94 ±1.63 |
| WISDM | 0 | 16 | 88.57 ±1.40 | 84.39 ±2.87 | 84.85 ±2.58 |
| WISDM | 1 | 1 | 91.46 ±1.05 | 89.48 ±2.05 | 88.87 ±1.60 |
| WISDM | 1 | 5 | 93.14 ±1.50 | 94.55 ±1.60 | 92.42 ±1.76 |
| WISDM | 1 | 16 | 89.66 ±1.08 | 89.60 ±1.28 | 86.51 ±1.13 |

Table 11: TRTR - TSBF

| dataset | $\gamma$ | $i$ | Accuracy | Recall | F1 |
|---|---|---|---|---|---|
| MHEALTH | baseline | - | 99.92 ±0.00 | 99.93 ±0 | 99.93 ±0 |
| MHEALTH | -1 | 1 | 100.00 ±0 | 100.00 ±0 | 100.00 ±0 |
| MHEALTH | -1 | 5 | 99.96 ±0 | 99.96 ±0 | 99.96 ±0.00 |
| MHEALTH | -1 | 16 | 99.96 ±0 | 99.96 ±0 | 99.96 ±0.00 |
| MHEALTH | 0 | 1 | 99.92 ±0.00 | 99.93 ±0 | 99.93 ±0 |
| MHEALTH | 0 | 5 | 99.84 ±0 | 99.86 ±0 | 99.86 ±0.00 |
| MHEALTH | 0 | 16 | 100.00 ±0 | 100.00 ±0 | 100.00 ±0 |
| MHEALTH | 1 | 1 | 100.00 ±0 | 100.00 ±0 | 100.00 ±0 |
| MHEALTH | 1 | 5 | 100.00 ±0 | 100.00 ±0 | 100.00 ±0 |
| MHEALTH | 1 | 16 | 99.96 ±0 | 99.96 ±0 | 99.96 ±0.00 |
| MHAD1 | baseline | - | 99.92 ±0 | 99.91 ±0 | 99.92 ±0 |
| MHAD1 | -1 | 1 | 99.92 ±0 | 99.91 ±0 | 99.92 ±0.00 |
| MHAD1 | -1 | 5 | 99.95 ±0.00 | 99.94 ±0.00 | 99.95 ±0.00 |
| MHAD1 | -1 | 16 | 100.00 ±0 | 100.00 ±0 | 100.00 ±0 |
| MHAD1 | 0 | 1 | 99.92 ±0 | 99.91 ±0 | 99.92 ±0.00 |
| MHAD1 | 0 | 5 | 100.00 ±0 | 100.00 ±0 | 100.00 ±0 |
| MHAD1 | 0 | 16 | 99.97 ±0.00 | 99.97 ±0.00 | 99.97 ±0.00 |
| MHAD1 | 1 | 1 | 99.95 ±0.00 | 99.94 ±0.00 | 99.95 ±0.00 |
| MHAD1 | 1 | 5 | 99.97 ±0.00 | 99.97 ±0 | 99.97 ±0.00 |
| MHAD1 | 1 | 16 | 99.95 ±0.00 | 99.94 ±0 | 99.95 ±0 |
| MHAD2 | baseline | - | 99.91 ±0.00 | 99.89 ±0 | 99.89 ±0 |
| MHAD2 | -1 | 1 | 99.91 ±0.00 | 99.89 ±0 | 99.89 ±0 |
| MHAD2 | -1 | 5 | 99.91 ±0.00 | 99.89 ±0 | 99.89 ±0 |
| MHAD2 | -1 | 16 | 100.00 ±0 | 100.00 ±0 | 100.00 ±0 |
| MHAD2 | 0 | 1 | 99.74 ±0 | 99.67 ±0 | 99.66 ±0.00 |
| MHAD2 | 0 | 5 | 99.74 ±0 | 99.67 ±0 | 99.66 ±0.00 |
| MHAD2 | 0 | 16 | 100.00 ±0 | 100.00 ±0 | 100.00 ±0 |
| MHAD2 | 1 | 1 | 99.74 ±0 | 99.67 ±0 | 99.66 ±0.00 |
| MHAD2 | 1 | 5 | 99.74 ±0 | 99.67 ±0 | 99.66 ±0.00 |
| MHAD2 | 1 | 16 | 100.00 ±0 | 100.00 ±0 | 100.00 ±0 |
| WHARF | baseline | - | 99.61 ±0.00 | 98.86 ±0 | 99.15 ±0.00 |
| WHARF | -1 | 1 | 99.77 ±0.00 | 99.31 ±0.00 | 99.44 ±0.00 |
| WHARF | -1 | 5 | 99.82 ±0.00 | 99.52 ±0.00 | 99.63 ±0 |
| WHARF | -1 | 16 | 99.85 ±0.00 | 99.54 ±0 | 99.64 ±0 |
| WHARF | 0 | 1 | 99.77 ±0.00 | 99.31 ±0.00 | 99.44 ±0.00 |
| WHARF | 0 | 5 | 99.82 ±0.00 | 99.52 ±0.00 | 99.63 ±0 |
| WHARF | 0 | 16 | 99.85 ±0.00 | 99.54 ±0 | 99.64 ±0 |
| WHARF | 1 | 1 | 99.77 ±0.00 | 99.31 ±0.00 | 99.44 ±0.00 |
| WHARF | 1 | 5 | 99.82 ±0.00 | 99.52 ±0.00 | 99.63 ±0 |
| WHARF | 1 | 16 | 99.85 ±0.00 | 99.54 ±0 | 99.64 ±0 |
| WISDM | baseline | - | 99.99 ±0.00 | 99.99 ±0.00 | 100.00 ±0.00 |
| WISDM | -1 | 1 | 99.99 ±0.00 | 99.99 ±0.00 | 100.00 ±0.00 |
| WISDM | -1 | 5 | 99.99 ±0.00 | 99.99 ±0.00 | 100.00 ±0.00 |
| WISDM | -1 | 16 | 100.00 ±0 | 100.00 ±0 | 100.00 ±0 |
| WISDM | 0 | 1 | 100.00 ±0.00 | 100.00 ±0.00 | 100.00 ±0 |
| WISDM | 0 | 5 | 100.00 ±0 | 100.00 ±0 | 100.00 ±0 |
| WISDM | 0 | 16 | 100.00 ±0.00 | 100.00 ±0 | 100.00 ±0 |
| WISDM | 1 | 1 | 100.00 ±0 | 100.00 ±0 | 100.00 ±0 |
| WISDM | 1 | 5 | 100.00 ±0 | 100.00 ±0 | 100.00 ±0 |
| WISDM | 1 | 16 | 100.00 ±0 | 100.00 ±0 | 100.00 ±0 |

| dataset | gamma | i | acc | recall | f1 |
|---|---|---|---|---|---|
| MHEALTH | baseline | - | $61.00 \pm 4.09$ | $60.72 \pm 3.78$ | $57.42 \pm 4.38$ |
| MHEALTH | -1 | 1 | $58.06 \pm 2.23$ | $59.44 \pm 2.23$ | $53.70 \pm 2.26$ |
| MHEALTH | -1 | 5 | $54.97 \pm 1.45$ | $54.60 \pm 1.87$ | $47.06 \pm 1.95$ |
| MHEALTH | -1 | 16 | $45.74 \pm 1.87$ | $45.06 \pm 2.06$ | $37.57 \pm 2.47$ |
| MHEALTH | 0 | 1 | $58.83 \pm 2.34$ | $60.07 \pm 2.35$ | $53.99 \pm 2.72$ |
| MHEALTH | 0 | 5 | $55.45 \pm 1.38$ | $56.32 \pm 1.54$ | $48.92 \pm 1.69$ |
| MHEALTH | 0 | 16 | $47.17 \pm 2.72$ | $47.30 \pm 2.10$ | $38.60 \pm 2.58$ |
| MHEALTH | 1 | 1 | $59.53 \pm 2.23$ | $60.74 \pm 2.21$ | $55.06 \pm 2.60$ |
| MHEALTH | 1 | 5 | $55.01 \pm 2.01$ | $55.70 \pm 2.37$ | $48.05 \pm 2.53$ |
| MHEALTH | 1 | 16 | $47.00 \pm 4.15$ | $47.16 \pm 3.84$ | $39.27 \pm 4.98$ |
| MHAD1 | baseline | - | $35.62 \pm 1.98$ | $35.15 \pm 2.12$ | $32.58 \pm 2.39$ |
| MHAD1 | -1 | 1 | $28.24 \pm 1.14$ | $27.03 \pm 1.26$ | $23.82 \pm 1.26$ |
| MHAD1 | -1 | 5 | $31.13 \pm 0.84$ | $30.33 \pm 0.76$ | $25.34 \pm 0.93$ |
| MHAD1 | -1 | 16 | $29.92 \pm 1.02$ | $29.29 \pm 1.04$ | $25.99 \pm 1.12$ |
| MHAD1 | 0 | 1 | $29.06 \pm 1.11$ | $27.95 \pm 1.09$ | $24.48 \pm 1.40$ |
| MHAD1 | 0 | 5 | $28.93 \pm 0.95$ | $27.98 \pm 0.85$ | $23.06 \pm 0.96$ |
| MHAD1 | 0 | 16 | $24.40 \pm 3.30$ | $24.40 \pm 3.18$ | $19.97 \pm 3.00$ |
| MHAD1 | 1 | 1 | $26.65 \pm 1.19$ | $25.52 \pm 1.20$ | $23.33 \pm 1.14$ |
| MHAD1 | 1 | 5 | $29.34 \pm 0.82$ | $28.14 \pm 0.86$ | $23.33 \pm 1.07$ |
| MHAD1 | 1 | 16 | $28.59 \pm 1.43$ | $27.97 \pm 1.40$ | $24.70 \pm 1.28$ |
| MHAD2 | baseline | - | $48.31 \pm 3.62$ | $47.47 \pm 2.49$ | $41.97 \pm 4.12$ |
| MHAD2 | -1 | 1 | $27.77 \pm 3.72$ | $28.19 \pm 2.89$ | $16.80 \pm 2.95$ |
| MHAD2 | -1 | 5 | $40.19 \pm 2.01$ | $38.46 \pm 2.01$ | $32.15 \pm 1.94$ |
| MHAD2 | -1 | 16 | $51.44 \pm 2.85$ | $49.74 \pm 2.88$ | $41.50 \pm 3.32$ |
| MHAD2 | 0 | 1 | $24.48 \pm 4.34$ | $24.99 \pm 3.00$ | $12.97 \pm 2.83$ |
| MHAD2 | 0 | 5 | $36.43 \pm 1.08$ | $34.91 \pm 1.24$ | $26.19 \pm 2.12$ |
| MHAD2 | 0 | 16 | $42.61 \pm 7.93$ | $44.52 \pm 7.37$ | $36.17 \pm 8.85$ |
| MHAD2 | 1 | 1 | $25.04 \pm 3.16$ | $26.35 \pm 2.28$ | $15.87 \pm 2.09$ |
| MHAD2 | 1 | 5 | $32.68 \pm 2.44$ | $31.17 \pm 2.23$ | $22.93 \pm 2.55$ |
| WHARF | baseline | - | $20.94 \pm 3.16$ | $20.12 \pm 2.65$ | $11.41 \pm 2.47$ |
| WHARF | -1 | 1 | $44.90 \pm 4.10$ | $34.09 \pm 2.32$ | $26.61 \pm 2.07$ |
| WHARF | -1 | 5 | $45.37 \pm 2.12$ | $31.66 \pm 1.34$ | $23.96 \pm 1.24$ |
| WHARF | -1 | 16 | $43.69 \pm 1.74$ | $26.64 \pm 2.16$ | $20.59 \pm 2.21$ |
| WHARF | 0 | 1 | $43.37 \pm 4.33$ | $34.12 \pm 2.51$ | $26.35 \pm 2.30$ |
| WHARF | 0 | 5 | $45.27 \pm 1.72$ | $31.32 \pm 1.40$ | $24.13 \pm 1.48$ |
| WHARF | 0 | 16 | $40.33 \pm 6.41$ | $26.07 \pm 2.63$ | $19.78 \pm 2.82$ |
| WHARF | 1 | 1 | $41.47 \pm 5.47$ | $35.31 \pm 2.47$ | $25.93 \pm 2.34$ |
| WHARF | 1 | 5 | $45.25 \pm 3.03$ | $31.71 \pm 1.25$ | $23.96 \pm 1.28$ |
| WHARF | 1 | 16 | $39.93 \pm 4.07$ | $25.27 \pm 1.60$ | $18.53 \pm 2.07$ |
| WISDM | baseline | - | $50.07 \pm 3.78$ | $55.08 \pm 2.21$ | $47.38 \pm 2.49$ |
| WISDM | -1 | 1 | $93.12 \pm 0.98$ | $93.02 \pm 1.90$ | $91.80 \pm 1.59$ |
| WISDM | -1 | 5 | $92.25 \pm 0.86$ | $92.86 \pm 1.36$ | $90.80 \pm 1.40$ |
| WISDM | -1 | 16 | $90.79 \pm 1.33$ | $90.68 \pm 1.81$ | $88.37 \pm 1.87$ |
| WISDM | 0 | 1 | $91.38 \pm 0.84$ | $89.70 \pm 1.84$ | $89.09 \pm 1.33$ |
| WISDM | 0 | 5 | $91.35 \pm 1.69$ | $91.60 \pm 1.45$ | $89.94 \pm 1.63$ |
| WISDM | 0 | 16 | $88.57 \pm 1.40$ | $84.39 \pm 2.87$ | $84.85 \pm 2.58$ |
| WISDM | 1 | 1 | $91.46 \pm 1.05$ | $89.48 \pm 2.05$ | $88.87 \pm 1.60$ |
| WISDM | 1 | 5 | $93.14 \pm 1.50$ | $94.55 \pm 1.60$ | $92.42 \pm 1.76$ |
| WISDM | 1 | 16 | $89.66 \pm 1.08$ | $89.60 \pm 1.28$ | $86.51 \pm 1.13$ |

Table 12: TSTR - TSBF

| dataset | gamma | i | acc | recall | f1 |
|---|---|---|---|---|---|
| MHEALTH | baseline | - | $55.79 \pm 2.44$ | $55.07 \pm 2.48$ | $52.56 \pm 2.33$ |
| MHEALTH | 5 | 1 | $59.27 \pm 1.80$ | $58.00 \pm 2.19$ | $55.69 \pm 2.32$ |
| MHEALTH | 5 | 5 | $61.95 \pm 2.13$ | $60.92 \pm 2.26$ | $59.17 \pm 2.32$ |
| MHEALTH | 5 | 16 | $62.54 \pm 2.21$ | $62.39 \pm 1.98$ | $60.00 \pm 1.77$ |
| MHAD1 | baseline | - | $33.86 \pm 1.46$ | $33.35 \pm 1.57$ | $32.85 \pm 1.66$ |
| MHAD2 | baseline | - | $46.97 \pm 3.70$ | $45.90 \pm 3.39$ | $43.52 \pm 3.78$ |
| MHAD2 | 5 | 1 | $46.81 \pm 3.70$ | $46.29 \pm 3.42$ | $43.84 \pm 3.75$ |
| MHAD2 | 5 | 5 | $49.19 \pm 3.46$ | $48.71 \pm 3.14$ | $46.82 \pm 3.63$ |
| MHAD2 | 5 | 16 | $53.25 \pm 2.18$ | $52.88 \pm 1.74$ | $50.68 \pm 2.53$ |
| WHARF | baseline | - | $15.46 \pm 3.04$ | $14.30 \pm 1.67$ | $6.14 \pm 1.78$ |
| WHARF | 5 | 1 | $15.49 \pm 2.84$ | $13.68 \pm 1.41$ | $5.84 \pm 1.16$ |
| WHARF | 5 | 5 | $17.21 \pm 3.60$ | $13.16 \pm 2.20$ | $5.76 \pm 1.43$ |
| WHARF | 5 | 16 | $16.32 \pm 3.57$ | $12.86 \pm 1.70$ | $6.32 \pm 1.27$ |
| WISDM | baseline | - | $53.03 \pm 3.04$ | $50.44 \pm 2.32$ | $44.94 \pm 2.51$ |
| WISDM | 5 | 1 | $50.65 \pm 4.22$ | $47.15 \pm 2.37$ | $41.49 \pm 2.28$ |
| WISDM | 5 | 5 | $48.09 \pm 4.79$ | $45.71 \pm 2.63$ | $39.48 \pm 2.94$ |
| WISDM | 5 | 16 | $47.96 \pm 5.75$ | $45.27 \pm 2.78$ | $39.01 \pm 3.20$ |

Table 13: TSTR-DClassifier- $\gamma$=5

| dataset | gamma | i | acc | recall | f1 |
|---|---|---|---|---|---|
| MHEALTH | baseline | - | $96.08 \pm 0.79$ | $96.38 \pm 0.74$ | $96.37 \pm 0.75$ |
| MHEALTH | 5 | 1 | $96.64 \pm 0.50$ | $96.91 \pm 0.46$ | $96.89 \pm 0.47$ |
| MHEALTH | 5 | 5 | $97.20 \pm 0.25$ | $97.43 \pm 0.23$ | $97.40 \pm 0.24$ |
| MHEALTH | 5 | 16 | $97.33 \pm 0.37$ | $97.55 \pm 0.34$ | $97.53 \pm 0.34$ |
| MHAD1 | baseline | - | $67.08 \pm 1.18$ | $67.15 \pm 1.17$ | $67.13 \pm 1.14$ |
| MHAD2 | baseline | - | $68.32 \pm 1.05$ | $67.55 \pm 1.02$ | $67.58 \pm 1.31$ |
| MHAD2 | 5 | 1 | $71.66 \pm 1.45$ | $70.50 \pm 1.60$ | $71.18 \pm 1.52$ |
| MHAD2 | 5 | 5 | $78.02 \pm 0.98$ | $77.03 \pm 1.00$ | $77.46 \pm 0.92$ |
| MHAD2 | 5 | 16 | $81.78 \pm 1.41$ | $80.58 \pm 1.35$ | $81.19 \pm 1.43$ |
| WHARF | baseline | - | $83.11 \pm 1.85$ | $77.63 \pm 2.34$ | $78.62 \pm 2.08$ |
| WHARF | 5 | 1 | $81.97 \pm 3.34$ | $75.61 \pm 4.48$ | $76.72 \pm 4.62$ |
| WHARF | 5 | 5 | $85.34 \pm 2.85$ | $80.41 \pm 3.66$ | $81.74 \pm 3.50$ |
| WHARF | 5 | 16 | $88.21 \pm 2.66$ | $84.02 \pm 3.84$ | $84.65 \pm 3.60$ |
| WISDM | baseline | - | $99.47 \pm 0.09$ | $99.20 \pm 0.15$ | $99.21 \pm 0.12$ |
| WISDM | 5 | 1 | $99.53 \pm 0.06$ | $99.18 \pm 0.12$ | $99.25 \pm 0.08$ |
| WISDM | 5 | 5 | $99.59 \pm 0.05$ | $99.21 \pm 0.11$ | $99.36 \pm 0.08$ |
| WISDM | 5 | 16 | $99.39 \pm 0.03$ | $98.73 \pm 0.07$ | $99.01 \pm 0.04$ |

Table 14: DClassifier - TRTR - $\gamma = 5$

| dataset | gamma | i | acc | recall | f1 |
|---|---|---|---|---|---|
| MHEALTH | 5 | 1 | $100.00 \pm 0.0$ | $100.00 \pm 0.0$ | $100.00 \pm 0.0$ |
| MHEALTH | 5 | 5 | $100.00 \pm 0.0$ | $100.00 \pm 0.0$ | $100.00 \pm 0.0$ |
| MHEALTH | 5 | 5 | $100.00 \pm 0.0$ | $100.00 \pm 0.0$ | $100.00 \pm 0.0$ |
| MHEALTH | 5 | 5 | $99.96 \pm 0.0$ | $99.96 \pm 0.0$ | $99.96 \pm 0.00$ |
| MHEALTH | 5 | 16 | $100.00 \pm 0.0$ | $100.00 \pm 0.0$ | $100.00 \pm 0.0$ |
| MHEALTH | 5 | 16 | $99.96 \pm 0.0$ | $99.96 \pm 0.0$ | $99.96 \pm 0.00$ |
| MHAD1 | 5 | 1 | $100.00 \pm 0.0$ | $100.00 \pm 0.0$ | $100.00 \pm 0.0$ |
| MHAD1 | 5 | 5 | $100.00 \pm 0.0$ | $100.00 \pm 0.0$ | $100.00 \pm 0.0$ |
| MHAD1 | 5 | 5 | $99.95 \pm 0.00$ | $99.94 \pm 0.0$ | $99.94 \pm 0.00$ |
| MHAD1 | 5 | 5 | $99.92 \pm 0.0$ | $99.91 \pm 0.0$ | $99.92 \pm 0.00$ |
| MHAD1 | 5 | 16 | $100.00 \pm 0.0$ | $100.00 \pm 0.0$ | $100.00 \pm 0.0$ |
| MHAD1 | 5 | 16 | $99.95 \pm 0.00$ | $99.94 \pm 0.0$ | $99.94 \pm 0.00$ |
| MHAD2 | 5 | 1 | $100.00 \pm 0.0$ | $100.00 \pm 0.0$ | $100.00 \pm 0.0$ |
| MHAD2 | 5 | 5 | $100.00 \pm 0.0$ | $100.00 \pm 0.0$ | $100.00 \pm 0.0$ |
| MHAD2 | 5 | 5 | $100.00 \pm 0.0$ | $100.00 \pm 0.0$ | $100.00 \pm 0.0$ |
| MHAD2 | 5 | 5 | $99.74 \pm 0.0$ | $99.67 \pm 0.0$ | $99.66 \pm 0.00$ |
| MHAD2 | 5 | 16 | $100.00 \pm 0.0$ | $100.00 \pm 0.0$ | $100.00 \pm 0.0$ |
| MHAD2 | 5 | 16 | $99.74 \pm 0.0$ | $99.67 \pm 0.0$ | $99.66 \pm 0.00$ |
| WHARF | 5 | 1 | $100.00 \pm 0.0$ | $100.00 \pm 0.0$ | $100.00 \pm 0.0$ |
| WHARF | 5 | 5 | $100.00 \pm 0.0$ | $100.00 \pm 0.0$ | $100.00 \pm 0.0$ |
| WHARF | 5 | 5 | $99.85 \pm 0.00$ | $99.54 \pm 0.0$ | $99.64 \pm 0.0$ |
| WHARF | 5 | 5 | $99.82 \pm 0.00$ | $99.52 \pm 0.00$ | $99.63 \pm 0.0$ |
| WHARF | 5 | 16 | $100.00 \pm 0.0$ | $100.00 \pm 0.0$ | $100.00 \pm 0.0$ |
| WHARF | 5 | 16 | $99.77 \pm 0.00$ | $99.31 \pm 0.00$ | $99.44 \pm 0.00$ |
| WISDM | 5 | 1 | $100.00 \pm 0.0$ | $100.00 \pm 0.0$ | $100.00 \pm 0.0$ |
| WISDM | 5 | 5 | $100.00 \pm 0.0$ | $100.00 \pm 0.0$ | $100.00 \pm 0.0$ |
| WISDM | 5 | 5 | $100.00 \pm 0.0$ | $100.00 \pm 0.0$ | $100.00 \pm 0.0$ |
| WISDM | 5 | 5 | $99.99 \pm 0.0$ | $99.99 \pm 0.0$ | $99.99 \pm 0.00$ |
| WISDM | 5 | 16 | $100.00 \pm 0.0$ | $100.00 \pm 0.0$ | $100.00 \pm 0.0$ |
| WISDM | 5 | 16 | $100.00 \pm 0.0$ | $100.00 \pm 0.0$ | $100.00 \pm 0.0$ |

Table 15: TRTR - TSRF - $\gamma = 5$

| dataset | gamma | i | acc | recall | f1 |
|---|---|---|---|---|---|
| MHAD2 | 5 | 5 | $38.06 \pm 1.61$ | $38.55 \pm 1.64$ | $33.79 \pm 2.02$ |
| MHAD2 | 5 | 5 | $38.04 \pm 1.65$ | $38.36 \pm 1.61$ | $33.67 \pm 2.05$ |
| MHAD2 | 5 | 16 | $37.77 \pm 1.75$ | $38.27 \pm 1.59$ | $33.49 \pm 2.10$ |
| MHAD2 | 5 | 5 | $36.01 \pm 2.77$ | $36.57 \pm 2.80$ | $32.15 \pm 2.86$ |
| MHAD2 | 5 | 16 | $35.95 \pm 2.73$ | $36.59 \pm 2.72$ | $31.94 \pm 3.05$ |
| MHAD2 | 5 | 1 | $35.76 \pm 2.89$ | $36.37 \pm 2.84$ | $31.89 \pm 3.06$ |
| WISDM | 5 | 5 | $38.00 \pm 7.01$ | $34.62 \pm 2.46$ | $29.36 \pm 3.59$ |
| WISDM | 5 | 16 | $37.91 \pm 5.62$ | $34.42 \pm 2.62$ | $29.31 \pm 3.56$ |
| WISDM | 5 | 5 | $37.27 \pm 5.47$ | $34.67 \pm 2.18$ | $28.57 \pm 3.48$ |
| MHEALTH | 5 | 16 | $31.46 \pm 2.21$ | $30.10 \pm 1.78$ | $26.76 \pm 2.24$ |
| MHEALTH | 5 | 5 | $31.22 \pm 2.81$ | $29.86 \pm 2.18$ | $26.57 \pm 2.61$ |
| MHEALTH | 5 | 5 | $30.85 \pm 2.57$ | $29.46 \pm 1.99$ | $26.18 \pm 2.80$ |
| WISDM | 5 | 16 | $30.71 \pm 6.96$ | $34.58 \pm 3.46$ | $25.30 \pm 4.57$ |
| WISDM | 5 | 1 | $30.62 \pm 6.72$ | $34.18 \pm 3.25$ | $25.17 \pm 4.25$ |
| WISDM | 5 | 5 | $30.48 \pm 6.84$ | $34.15 \pm 3.18$ | $24.74 \pm 4.39$ |
| MHEALTH | 5 | 1 | $24.11 \pm 2.98$ | $23.03 \pm 2.73$ | $21.03 \pm 2.74$ |
| MHEALTH | 5 | 16 | $23.65 \pm 3.29$ | $22.60 \pm 3.02$ | $20.37 \pm 3.00$ |
| MHEALTH | 5 | 5 | $23.50 \pm 3.19$ | $22.51 \pm 2.93$ | $20.24 \pm 3.00$ |
| MHAD1 | 5 | 1 | $21.59 \pm 0.85$ | $20.89 \pm 0.89$ | $19.56 \pm 0.93$ |
| MHAD1 | 5 | 16 | $21.43 \pm 0.91$ | $20.74 \pm 0.96$ | $19.43 \pm 0.97$ |
| MHAD1 | 5 | 5 | $21.37 \pm 0.95$ | $20.67 \pm 0.97$ | $19.30 \pm 0.99$ |
| MHAD1 | 5 | 5 | $20.32 \pm 0.77$ | $19.73 \pm 0.78$ | $18.78 \pm 0.73$ |
| MHAD1 | 5 | 5 | $20.13 \pm 0.86$ | $19.54 \pm 0.87$ | $18.63 \pm 0.82$ |
| MHAD1 | 5 | 16 | $20.09 \pm 0.84$ | $19.49 \pm 0.84$ | $18.59 \pm 0.79$ |
| WHARF | 5 | 5 | $15.37 \pm 3.69$ | $13.84 \pm 1.64$ | $5.45 \pm 1.69$ |
| WHARF | 5 | 5 | $15.03 \pm 3.72$ | $13.53 \pm 1.59$ | $5.19 \pm 1.57$ |
| WHARF | 5 | 16 | $15.01 \pm 3.55$ | $13.55 \pm 1.56$ | $5.04 \pm 1.37$ |
| WHARF | 5 | 5 | $12.44 \pm 3.52$ | $10.40 \pm 1.81$ | $4.03 \pm 1.24$ |
| WHARF | 5 | 16 | $12.40 \pm 3.52$ | $10.37 \pm 1.78$ | $3.99 \pm 1.15$ |
| WHARF | 5 | 1 | $12.45 \pm 3.51$ | $10.43 \pm 1.83$ | $3.98 \pm 1.19$ |

Table 16: TSTR - TSRF - $\gamma = 5$

| dataset | gamma | i | acc | recall | f1 |
|---------|-------|---|-----|--------|-----|
| MHEALTH | 5 | 1 | $31.46 \pm 2.21$ | $30.10 \pm 1.78$ | $26.76 \pm 2.24$ |
| MHEALTH | 5 | 1 | $23.50 \pm 3.19$ | $22.51 \pm 2.93$ | $20.24 \pm 3.00$ |
| MHEALTH | 5 | 5 | $31.22 \pm 2.81$ | $29.86 \pm 2.18$ | $26.57 \pm 2.61$ |
| MHEALTH | 5 | 16 | $30.85 \pm 2.57$ | $29.46 \pm 1.99$ | $26.18 \pm 2.80$ |
| MHEALTH | 5 | 16 | $24.11 \pm 2.98$ | $23.03 \pm 2.73$ | $21.03 \pm 2.74$ |
| MHEALTH | 5 | 16 | $23.65 \pm 3.29$ | $22.60 \pm 3.02$ | $20.37 \pm 3.00$ |
| MHAD1 | 5 | 1 | $21.37 \pm 0.95$ | $20.67 \pm 0.97$ | $19.30 \pm 0.99$ |
| MHAD1 | 5 | 1 | $20.09 \pm 0.84$ | $19.49 \pm 0.84$ | $18.59 \pm 0.79$ |
| MHAD1 | 5 | 5 | $20.32 \pm 0.77$ | $19.73 \pm 0.78$ | $18.78 \pm 0.73$ |
| MHAD1 | 5 | 16 | $21.59 \pm 0.85$ | $20.89 \pm 0.89$ | $19.56 \pm 0.93$ |
| MHAD1 | 5 | 16 | $21.43 \pm 0.91$ | $20.74 \pm 0.96$ | $19.43 \pm 0.97$ |
| MHAD1 | 5 | 16 | $20.13 \pm 0.86$ | $19.54 \pm 0.87$ | $18.63 \pm 0.82$ |
| MHAD2 | 5 | 1 | $37.77 \pm 1.75$ | $38.27 \pm 1.59$ | $33.49 \pm 2.10$ |
| MHAD2 | 5 | 1 | $36.01 \pm 2.77$ | $36.57 \pm 2.80$ | $32.15 \pm 2.86$ |
| MHAD2 | 5 | 5 | $38.06 \pm 1.61$ | $38.55 \pm 1.64$ | $33.79 \pm 2.02$ |
| MHAD2 | 5 | 16 | $38.04 \pm 1.65$ | $38.36 \pm 1.61$ | $33.67 \pm 2.05$ |
| MHAD2 | 5 | 16 | $35.95 \pm 2.73$ | $36.59 \pm 2.72$ | $31.94 \pm 3.05$ |
| MHAD2 | 5 | 16 | $35.76 \pm 2.89$ | $36.37 \pm 2.84$ | $31.89 \pm 3.06$ |
| WHARF | 5 | 1 | $15.01 \pm 3.55$ | $13.55 \pm 1.56$ | $5.04 \pm 1.37$ |
| WHARF | 5 | 1 | $12.44 \pm 3.52$ | $10.40 \pm 1.81$ | $4.03 \pm 1.24$ |
| WHARF | 5 | 5 | $15.37 \pm 3.69$ | $13.84 \pm 1.64$ | $5.45 \pm 1.69$ |
| WHARF | 5 | 16 | $15.03 \pm 3.72$ | $13.53 \pm 1.59$ | $5.19 \pm 1.57$ |
| WHARF | 5 | 16 | $12.40 \pm 3.52$ | $10.37 \pm 1.78$ | $3.99 \pm 1.15$ |
| WHARF | 5 | 16 | $12.45 \pm 3.51$ | $10.43 \pm 1.83$ | $3.98 \pm 1.19$ |
| WISDM | 5 | 1 | $37.91 \pm 5.62$ | $34.42 \pm 2.62$ | $29.31 \pm 3.56$ |
| WISDM | 5 | 1 | $30.48 \pm 6.84$ | $34.15 \pm 3.18$ | $24.74 \pm 4.39$ |
| WISDM | 5 | 5 | $37.27 \pm 5.47$ | $34.67 \pm 2.18$ | $28.57 \pm 3.48$ |
| WISDM | 5 | 16 | $38.00 \pm 7.01$ | $34.62 \pm 2.46$ | $29.36 \pm 3.59$ |
| WISDM | 5 | 16 | $30.71 \pm 6.96$ | $34.58 \pm 3.46$ | $25.30 \pm 4.57$ |
| WISDM | 5 | 16 | $30.62 \pm 6.72$ | $34.18 \pm 3.25$ | $25.17 \pm 4.25$ |

Table 17: TSTR - TSBF - $\gamma = 5$

| dataset | gamma | i | acc | recall | f1 |
|---|---|---|---|---|---|
| MHEALTH | 5 | 1 | $100.00 \pm 0.0$ | $100.00 \pm 0.0$ | $100.00 \pm 0.0$ |
| MHEALTH | 5 | 1 | $99.96 \pm 0.0$ | $99.96 \pm 0.0$ | $99.96 \pm 0.00$ |
| MHEALTH | 5 | 5 | $100.00 \pm 0.0$ | $100.00 \pm 0.0$ | $100.00 \pm 0.0$ |
| MHEALTH | 5 | 16 | $100.00 \pm 0.0$ | $100.00 \pm 0.0$ | $100.00 \pm 0.0$ |
| MHEALTH | 5 | 16 | $100.00 \pm 0.0$ | $100.00 \pm 0.0$ | $100.00 \pm 0.0$ |
| MHEALTH | 5 | 16 | $99.96 \pm 0.0$ | $99.96 \pm 0.0$ | $99.96 \pm 0.00$ |
| MHAD1 | 5 | 1 | $100.00 \pm 0.0$ | $100.00 \pm 0.0$ | $100.00 \pm 0.0$ |
| MHAD1 | 5 | 1 | $99.95 \pm 0.00$ | $99.94 \pm 0.0$ | $99.94 \pm 0.00$ |
| MHAD1 | 5 | 5 | $99.95 \pm 0.00$ | $99.94 \pm 0.0$ | $99.94 \pm 0.00$ |
| MHAD1 | 5 | 16 | $100.00 \pm 0.0$ | $100.00 \pm 0.0$ | $100.00 \pm 0.0$ |
| MHAD1 | 5 | 16 | $100.00 \pm 0.0$ | $100.00 \pm 0.0$ | $100.00 \pm 0.0$ |
| MHAD1 | 5 | 16 | $99.92 \pm 0.0$ | $99.91 \pm 0.0$ | $99.92 \pm 0.00$ |
| MHAD2 | 5 | 1 | $100.00 \pm 0.0$ | $100.00 \pm 0.0$ | $100.00 \pm 0.0$ |
| MHAD2 | 5 | 1 | $99.74 \pm 0.0$ | $99.67 \pm 0.0$ | $99.66 \pm 0.00$ |
| MHAD2 | 5 | 5 | $99.74 \pm 0.0$ | $99.67 \pm 0.0$ | $99.66 \pm 0.00$ |
| MHAD2 | 5 | 16 | $100.00 \pm 0.0$ | $100.00 \pm 0.0$ | $100.00 \pm 0.0$ |
| MHAD2 | 5 | 16 | $100.00 \pm 0.0$ | $100.00 \pm 0.0$ | $100.00 \pm 0.0$ |
| MHAD2 | 5 | 16 | $100.00 \pm 0.0$ | $100.00 \pm 0.0$ | $100.00 \pm 0.0$ |
| WHARF | 5 | 1 | $100.00 \pm 0.0$ | $100.00 \pm 0.0$ | $100.00 \pm 0.0$ |
| WHARF | 5 | 1 | $99.77 \pm 0.00$ | $99.31 \pm 0.00$ | $99.44 \pm 0.00$ |
| WHARF | 5 | 5 | $99.82 \pm 0.00$ | $99.52 \pm 0.00$ | $99.63 \pm 0.0$ |
| WHARF | 5 | 16 | $100.00 \pm 0.0$ | $100.00 \pm 0.0$ | $100.00 \pm 0.0$ |
| WHARF | 5 | 16 | $100.00 \pm 0.0$ | $100.00 \pm 0.0$ | $100.00 \pm 0.0$ |
| WHARF | 5 | 16 | $99.85 \pm 0.00$ | $99.54 \pm 0.0$ | $99.64 \pm 0.0$ |
| WISDM | 5 | 1 | $100.00 \pm 0.0$ | $100.00 \pm 0.0$ | $100.00 \pm 0.0$ |
| WISDM | 5 | 1 | $100.00 \pm 0.0$ | $100.00 \pm 0.0$ | $100.00 \pm 0.0$ |
| WISDM | 5 | 5 | $100.00 \pm 0.0$ | $100.00 \pm 0.0$ | $100.00 \pm 0.0$ |
| WISDM | 5 | 16 | $100.00 \pm 0.0$ | $100.00 \pm 0.0$ | $100.00 \pm 0.0$ |
| WISDM | 5 | 16 | $100.00 \pm 0.0$ | $100.00 \pm 0.0$ | $100.00 \pm 0.0$ |
| WISDM | 5 | 16 | $99.99 \pm 0.0$ | $99.99 \pm 0.0$ | $99.99 \pm 0.00$ |

Table 18: TRTR - TSBF - $\gamma = 5$

## D.2 TRANSFORMER-BASED MODELS

Considering the need to investigate whether RIP maintains its effectiveness under more recent architectures, we selected the MHAD2 dataset. Although relatively small, it provides an appropriate benchmark for assessing RIP's behavior in this setting.

| Dataset | Classifier | Approach | F1 |
|---------|-----------|----------|-----|
| MHAD2 | TimeTransformer | Baseline | $08.12 \pm 4.0$ |
| MHAD2 | TimeTransformer | RIP ($\gamma = 0$, $i = 16$) | $\mathbf{12.49 \pm 2.0}$ |
| Synthetic MHAD2 | TimeTransformer | Baseline | $23.68 \pm 4.0$ |
| Synthetic MHAD2 | TimeTransformer | RIP ($\gamma = 5$, $i = 16$) | $\mathbf{33.09 \pm 3.0}$ |
| MHAD2 | RevTransformer | Baseline | $69.03 \pm 1.0$ |
| MHAD2 | RevTransformer | RIP ($\gamma = 1$, $i = 5$) | $57.47 \pm 2.0$ |
| Synthetic MHAD2 | RevTransformer | Baseline | $44.29 \pm 3.0$ |
| Synthetic MHAD2 | RevTransformer | RIP ($\gamma = 1$, $i = 16$) | $\mathbf{44.98 \pm 3.0}$ |

These results show that RIP consistently improves performance on the lighter TimeTransformer model. In contrast, its effect is more nuanced on the RevTransformer, which already relies on temporal attention mechanisms that overlap with those enforced by RIP. We added a theoretical discussion in the revised paper explaining how RIP interacts with transformer-style temporal aggregation and why specific architectures benefit more than others.

## D.3 LEAVE-ONE-SUBJECT-OUT RESULTS

Domain generalization and domain randomization are related but distinct strategies for making models robust to domain shifts. Domain generalization aims to improve robustness to unseen target domains by training on multiple real source domains and learning features that remain stable across them. In contrast, domain randomization seeks robustness by exposing the model to a wide range of synthetic variations, so that the real-world domain becomes just another variation within that space. In other words, domain generalization focuses on learning invariances across multiple real datasets, whereas domain randomization relies on heavy variation from a single real source domain augmented through randomized transformations.

So far, the experiments have mainly focused on domain randomization. However, we performed an experiment using a Leave-One-Subject-Out (LOSO) domain generalization protocol to evaluate RIP's performance in this setting. The dataset used in this experiment is a publicly available variant of UTD-MHAD2 obtained from an online repository https://personal.utdallas.edu/~kehtar/UTD-MHAD.html . This version is organized per subject, making the LOSO protocol the standard choice for evaluating generalization to unseen individuals.

Since the dataset contains eight subjects, LOSO naturally yields eight folds. Table 19 presents the complete per-subject results as well as the global mean $\pm$ standard deviation for Accuracy, Recall, and F1-score for the TRTR setup without applying the RIP method.

Table 19: Per-subject performance metrics in TRTR setup without RIP. The AVG row shows the average for each metric.

| Subject | Accuracy | F1 | Recall |
|---------|----------|-----|--------|
| 1 | 0.9063 | 0.7446 | 0.7238 |
| 2 | 0.8594 | 0.8052 | 0.8310 |
| 3 | 0.7969 | 0.7946 | 0.7954 |
| 4 | 0.7969 | 0.7867 | 0.8092 |
| 5 | 0.7656 | 0.7104 | 0.7188 |
| 6 | 0.6406 | 0.4971 | 0.4875 |
| 7 | 0.7656 | 0.8120 | 0.8126 |
| 8 | 0.7656 | 0.6213 | 0.6400 |
| AVG | $0.7879 \pm 0.0838$ | $0.7461 \pm 0.1126$ | $0.7275 \pm 0.1211$ |

**Synthetic-data results.** Synthetic data were generated using the TLCGAN (learning one subject for testing out) to preserve temporal structure. The per-subject results for the synthetic LOSO experiment are presented in Table 20:

Table 20: Per-subject (synthetic) performance metrics with TSTR setup without RIP. The AVG row shows the average for each metric.

| Subject | Accuracy | F1 | Recall |
|---|---|---|---|
| 1 | 0.6094 | 0.4634 | 0.4739 |
| 2 | 0.4063 | 0.4048 | 0.3821 |
| 3 | 0.4688 | 0.3570 | 0.3953 |
| 4 | 0.6406 | 0.6289 | 0.6360 |
| 5 | 0.8750 | 0.8065 | 0.8057 |
| 6 | 0.4375 | 0.3761 | 0.3621 |
| 7 | 0.6250 | 0.5058 | 0.5515 |
| 8 | 0.5781 | 0.4756 | 0.4951 |
| AVG | $0.5801 \pm 0.1398$ | $0.5023 \pm 0.1403$ | $0.5127 \pm 0.1405$ |

**Final comparison.** Table 21 summarises the key comparison between baseline methods and RIP for both real and synthetic LOSO evaluation.

Table 21: Summary of LOSO experiments with and without RIP.

| Dataset | Approach | F1-score |
|---|---|---|
| LOSO Real | Baseline | 0.7461 |
| LOSO Real | RIP | **0.8552** |
| LOSO Synthetic | Baseline | 0.50 |
| LOSO Synthetic | RIP | **0.5527** |

# E  DISCUSSIONS

We performed a layer-wise analysis of the DClassifier's weight distributions to investigate the effects of RIP as a regularization technique. We hypothesize that RIP promotes weight uniformity—a property often associated with effective regularization Zhang et al. (2019). To test this, we compare the weight distributions of a model trained with RIP against a baseline model (trained without RIP) and a uniform distribution. We use the Kolmogorov–Smirnov (KS) test Cong et al. (2021) and the Wasserstein distance Panaretos & Zemel (2019) as our primary metrics. We denote our KS test setting as:

$$S_{\text{RIP}}(L) = KS(L_{\text{RIP}(\sigma)}, L_{\text{Uniform}}) \quad \text{and} \quad S_{\text{baseline}}(L) = KS(L_{\text{baseline}}, L_{\text{Uniform}})$$

Analogously, our Wasserstein settings:

$$W_{\text{RIP}}(L) = W(L_{\text{RIP}(\sigma)}, L_{\text{Uniform}}) \quad \text{and} \quad W_{\text{baseline}}(L) = W(L_{\text{baseline}}, L_{\text{Uniform}})$$

where:

1. $L_{\text{RIP}(\sigma)}$ represents the weight values of layer $L$ trained with RIP under setting $\sigma = (\gamma, i, \text{MHAD2}, \text{DClassifier}); \gamma \in \{0, 1, 5\}$ and $i$ is the best performer in the dataset,

2. $L_{\text{baseline}}$ corresponds to the same layer trained without RIP,

3. $L_{\text{Uniform}}$ is a reference uniform distribution with the same shape as the weights in $L$ (i.e., $L_{\text{Uniform}} \sim$ `scipy.uniform(shape = L.shape)`).

The same $L_{\text{Uniform}}$ is used when comparing RIP and baseline weights to ensure fairness.

We selected the MHAD2 dataset, which has the fewest classes and allows faster training while preserving model validity. The DClassifier was chosen as it serves as a common baseline across

datasets. Since the model was trained using 10-fold cross-validation, we selected the fold with the best validation score, assuming that its weights represent the model's generalization capability [Grimm et al. (2017)]. This evaluation is carried out for both the real and synthetic versions of MHAD2, enabling us to assess whether RIP's regularization effect holds across data modalities. In the following sections, we discuss some aspects of the outcomes in this investigation. Lower values of the KS statistic ($S$) and the Wasserstein distance ($W$) indicate greater similarity to the uniform distribution, suggesting more effective regularization. Full details on the experimental setup, including the selection of the MHAD2 dataset and the specific model configuration, are provided in Appendix E.

**Effects on Synthetic Data.** *Weight Uniformity, Layer-wise Effects, and the Role of $\gamma$:* Figures 3 show that RIP, particularly with $\gamma = 1$ and $\gamma = 5$, promotes more uniform weight distributions across layers, as reflected by reduced $S_{\text{RIP}}(L)$ values compared to the baseline. This effect is most pronounced in contextual layers such as Self-Attention, LSTM, and RNN, indicating that RIP is well-suited to sequential architectures. The method's effectiveness stems from introducing invariant patterns — $\gamma_{i,j}, x_{i,j}, \gamma_{i,j}$ — which reinforce contextual information and reduce overfitting. However, the impact is not uniform: $\gamma = 1$ provides stable regularization across layers, while $\gamma = 5$ induces stronger but less stable shifts in deeper layers, and $\gamma = 0$ shows minimal effect. Overall, $\gamma = 1$ emerges as the most reliable setting. *Statistical Proximity vs. Distributional Cost:* KS statistics capture local deviations from uniformity, while Wasserstein distance reflects global shifts. RIP reduces KS values—improving local uniformity—but often increases Wasserstein distances, especially in Conv2D and Dense layers, suggesting broader structural changes. Among variants, $\gamma = 1$ yields the smoothest trade-off, whereas $\gamma = 0$ is unstable and $\gamma = 5$ introduces more dispersion. This reveals a key trade-off: RIP enforces local regularization but may distort global distributions, making careful tuning of $\gamma$ essential. *Output and Model Confidence:* RIP also affects output distributions. KS tests over logits (Figure 4) show that baseline models are overly uniform—reflecting uncertainty—while RIP, especially with $\gamma = 1$, produces less uniform logits, i.e., more confident predictions. This aligns with higher F1-scores, indicating that RIP improves both decisiveness and accuracy.

**Effects on Real Data.** *Layer-wise Propagation, Sensitivity, and Contextual Suitability:* On real datasets, RIP propagates nonlinearly and asymmetrically through the network. Early Conv2D layers show minor sensitivity to $\gamma$, while intermediate layers (LSTM3, SelfAttention1, Lambda) respond strongly, with divergence metrics decreasing as $\gamma$ increases. Deeper, denser layers show limited change, likely due to accumulated nonlinearity. This highlights that RIP is most effective in context-aware layers, with diminishing influence in final representations. *Predictive Confidence, Trustworthiness, and Reliability:* RIP regularizes output confidence differently on real data. It reduces overconfidence, producing more calibrated predictions (Figure 4). Unlike the baseline, which is often overconfident and inaccurate, RIP with $\gamma = 1$ yields the best performance and the most reliable calibration. The trade-off between variance control and accuracy reappears: $\gamma = 5$ enforces strong uniformity but reduces accuracy, while $\gamma = 1$ balances regularization and predictive power. In practice, moderate RIP enhances generalization by mitigating overfitting without overly constraining representational capacity.

**Is RIP a Form of Data-centric Regularization?** Our analysis shows that RIP has consistent architectural effects across datasets: contextual layers (e.g., LSTM, SelfAttention) are sensitive, while Dense layers are not. However, its influence on output confidence differs by domain. In synthetic data, RIP increases decisiveness, while in real data, it reduces overconfidence, leading to better-calibrated predictions in both cases. The performance–regularization trade-off is also domain-dependent. Our findings suggest that RIP is a data-centric regularization technique implemented via a novel form of structured data augmentation that shapes weight dynamics through the input structure, without modifying the architecture or loss function. Unlike traditional $\ell^2$ regularization, which imposes global penalties, RIP steers optimization toward more uniform weight distributions by leveraging properties of the data itself. Its effect is not monotonic with $\gamma$, varying across layer types and depths—highlighting the need for careful tuning. By embedding a structural prior directly into training, RIP emerges as a compelling alternative to classical methods: when properly tuned, it encourages more uniform weights, yields better-calibrated predictions, and enhances generalization without compromising performance. By reducing overconfidence and overfitting, RIP proves especially useful for models prone to excessive certainty, such as LSTM-based architectures.

**Why RIP is more noticeable for Synthetic Data?** To further analyze the impact of RIP at the

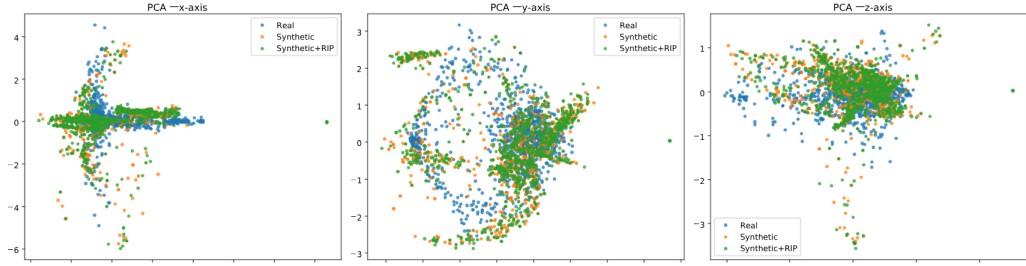

Figure 7: Comparing RIP under axis

channel level, Fig. 7 presents PCA projections computed independently for the $x$-, $y$-, and $z$-axes. The three plots reveal distinct behaviors across channels. For the $x$-axis, the synthetic data already lies close to the real distribution, which is expected, given the lower intrinsic variability typically associated with this axis; RIP, therefore, yields only modest refinement. In contrast, the $y$- and $z$-axes exhibit substantially greater structural complexity and synthetic–real mismatch. The raw synthetic samples display dispersion patterns and outliers that do not appear in real data, indicating generator-induced artifacts. RIP consistently suppresses these deviations, contracting the synthetic manifold and improving its alignment with the real distribution. This effect is particularly clear in the $y$-axis, where Synthetic+RIP closely follows the geometry of the real data cluster. These per-channel analyses reinforce that RIP enhances distributional fidelity most strongly in axes where the generator struggles to reproduce temporal consistency.

## F   SKETCHED FORMAL ANALYSIS

The model analyzed has six layers: Rnn inputs, Conv2d, Lambda, LSTM, Self-Attention, and Dense. Two are RNN-based. We will focus on RNN inputs and LSTMs because we covered them in the previous section. The LSTM in this TensorFlow implementation returns only three weights (called for 'whole sequence output,' 'final memory state,' and' final carry state'; for more details, see TensorFlow API Docs tfl).

As discussed in the main paper, the discussions showed that RNN-based layers are more affected by RIP. Considering that, we decided to analyze the impact of the RNN-based layers mathematically. We consider an RNN with bias parameters $b \in \mathbb{R}^h$ and identity activation function $\varphi(x) = x$ in the hidden layers. At time step $t$, the input is $x_t \in \mathbb{R}^d$, and the corresponding target is $y_t$. The hidden state $h_t \in \mathbb{R}^h$ and the output $o_t \in \mathbb{R}^q$ are computed as follows:

$$
\begin{aligned}
h_t &= W x_t + U h_{t-1} + b, \\
o_t &= V h_t + C, \\
\hat{y}_t &= \text{softmax}(o_t),
\end{aligned}
$$

where $W \in \mathbb{R}^{d \times h}$, $U \in \mathbb{R}^{h \times h}$, $V \in \mathbb{R}^{h \times q}$, and $C \in \mathbb{R}^q$. The output vector $o_t = (o_{t1}, \ldots, o_{tq})$ represents the logits for a $q$-class classification task. The softmax function is applied element-wise as $\text{softmax}(o_{ti}) = \frac{e^{o_{ti}}}{\sum_{j=1}^{q} e^{o_{tj}}}$, producing a probability vector $\hat{y}_t \in \mathbb{R}^q$.

For a multi-class classification setting, we define the loss as the average cross-entropy over a mini-batch:

$$
\ell = -\frac{1}{n} \sum_{i=1}^{n} \sum_{j=1}^{q} y_{ij} \log(\hat{y}_{ij}). \tag{4}
$$

Here, $h_t = h_t(W, x_t, h_{t-1}, b)$ and the loss depends on $V$, $h_t$, and $C$, i.e., $\ell = \ell(V, h_t, C)$. Therefore,

$$
\begin{aligned}
h_1 &= W x_1 + U h_0 + b = W \gamma + b \\
h_2 &= W x_2 + U h_1 + b = W \gamma + U [W \gamma + b] + b
\end{aligned}
$$

Then,
$$h_3 = Wx_3 + Uh_2 + b$$
$$= Wx_3 + U[U[W\gamma + b] + b] + b$$
$$= Wx_3 + U^2[W\gamma + b] + Ub + b$$

It implies in
$$h_4 = Wx_4 + Uh_3 + b$$
$$= W\gamma + U[Wx_3 + U^2[W\gamma + b] + Ub + b] + b$$
$$= W\gamma + UWx_3 + U^3[W\gamma + b] + U^2b + Ub + b$$

Finally,

$$h_5 = Wx_5 + Uh_4 + b$$
$$= W\gamma + U\{W\gamma + UWx_3 + U^3[W\gamma + b] + U^2b + Ub + b\} + b$$
$$= [U^4 + U^3 + U + 1]W\gamma + U^2Wx_3 + \sum_4^i U^i b$$

However, without CAR,

$$h_2 = Wx_2 + Uh_1 + b$$
$$= Wx_2 + U[Wx_1 + b] + b$$
$$= Wx_2 + UWx_1 + Ub + b$$

Then,
$$h_3 = Wx_3 + Uh_2 + b$$
$$= Wx_3 + U[Wx_2 + UWx_1 + Ub + b] + b$$
$$= Wx_3 + UWx_2 + U^2Wx_1 + U^2b + Ub + b$$

Consequently.
$$h_4 = Wx_4 + Uh_3 + b$$
$$= Wx_4 + U[Wx_3 + UWx_2 + U^2Wx_1 + U^2b + Ub + b] + b$$
$$= Wx_4 + UWx_3 + U^2Wx_2 + U^3Wx_1 + U^3b + U^2b + Ub + b$$

Finally,
$$h_5 = Wx_5 + Uh_4 + b$$
$$= Wx_5 + U[Wx_4 + UWx_3 + U^2Wx_2 + U^3Wx_1 + U^3b + U^2b + Ub + b] + b$$
$$= Wx_5 + UWx_4 + U^2Wx_3 + U^3Wx_2 + U^4x_1 + U^4b + U^3b + U^2b + +Ub + b$$
$$= \sum_{i=0}^{4} (U^i x_{5-i} + U^i b)$$

To investigate the effect of repeated input sequences, we simulate a training scenario with a mini-batch of 5 steps and initialize $h_0 = 0$. We assume that the constant samples $\gamma$ is present at steps $t = 1, 2, 4, 5$.

**Under RIP** As derived in previous equations, the hidden state at $t = 5$ can be written as:

$$h_5 = \left(U^4 + U^3 + U + I\right) W\gamma + U^2Wx_3 + \sum_{i=0}^{4} U^i b, \tag{5}$$

where $x_1 = x_2 = x_4 = x_5 = \gamma$ are the inputs, following the RIP procedure.

**Without RIP** In contrast, when all inputs are distinct (i.e., no constant samples added), the hidden state at $t = 5$ becomes:

$$h_5 = \sum_{i=0}^{4} \left( U^i b + U^i W x_{5-i} \right), \tag{6}$$

where $x_1, \ldots, x_5$ are assumed to be distinct inputs. This leads to a richer combination of temporal features, diversifying the contribution of $W$ and $b$ over time.

Equations 4 and 6 clearly show that the resulting $h_5$ is highly constrained by the repeated appearance of $\gamma$, and the majority of the dynamics come from powers of $U$ acting on a fixed term $W\gamma$. Thus, the variability of $h_t$ across time is dampened. Using RIP introduces a structural bias toward low-variance representations, much as traditional regularization techniques such as weight decay or dropout do. Indeed, comparing the variance of the hidden state $h_5$ in the two previously explored scenarios, and assuming that each input $x_t \in \mathbb{R}^h$ is drawn independently from a distribution with mean $\mu$ and covariance $\Sigma$, i.e.,

$$\mathbb{E}[x_t] = \mu, \quad \text{Cov}(x_t) = \Sigma.$$

**Without RIP** All inputs are distinct: $x_1, x_2, x_3, x_4, x_5$ are i.i.d. Then the hidden state at $t = 5$ as in Eq. **??**, the expectation and variance:

$$\mathbb{E}[h_5^{(\text{noRIP})}] = \sum_{i=0}^{4} U^i W \mu + \sum_{i=0}^{4} U^i b,$$

$$\text{Var}(h_5^{(\text{noRIP})}) = \sum_{i=0}^{4} U^i W \Sigma W^\top (U^i)^\top. \tag{7}$$

**With RIP** Assume inputs $x_1 = x_2 = x_4 = x_5 = \gamma$ are fixed and only $x_3$ is random. Then, the expected value and variance become:

$$\mathbb{E}[h_5^{(\text{RIP})}] = (U^4 + U^3 + U + I)W\gamma + U^2 W \mu + \sum_{i=0}^{4} U^i b,$$

$$\text{Var}(h_5^{(\text{RIP})}) = U^2 W \Sigma W^\top (U^2)^\top. \tag{8}$$

Although each input $x_t$ has the same covariance $\Sigma$, in the Eq. 7 the variance of $h_5$ accumulates over five independent sources $x_i; \forall i = 1, \cdots, 4$, while in the Eq. 8, it depends solely on one term, $x_3$.

This shows that RIP leads to reduced variance in the hidden representation. Repeatedly using the fixed input $\gamma$ reduces temporal diversity and constrains the representation to a smaller subspace. This supports the interpretation of RIP as an implicit regularizer, promoting smoother gradient flow and reducing overfitting—at the cost of reduced representational capacity when overused. The hidden states in RIP become combinations of a limited set of patterns, which reduces the RNN's ability to overfit to the training data. It results in more predictable, homogeneous hidden states, leading to lower gradient variance and serving as an implicit regularizer. This confirms the empirical observation that RIP can improve generalization — mainly when used with moderation — and explains why excessive repetition (e.g., high $i$ values) leads to stagnated performance.

Consequently, due to the chain rule, the weight updates are affected by the RIP, which reduces their variability and reinforces the effects of $\gamma$ and the regularization. Then, by the chain rule, for all weight $\Omega \in \{W, V, b, U\}$,

$$\Omega = \Omega_{\text{current}} - \alpha \frac{\partial \ell}{\partial P_k} \frac{\partial o_5}{\partial P_k} \frac{\partial P_k}{\partial \Omega}$$

where $\partial \Omega$ relies on $h_5$, implies a lower variability in the RIP scenario, as previously discussed.

**Nonlinear activations.** If the activation function $\varphi$ is Lipschitz-continuous with constant $L$ (e.g., ReLU, tanh, GELU), then:

$$\mathrm{Var}(h_t) = \mathrm{Var}(\varphi(z_t)) \leq L^2 \mathrm{Var}(z_t), \qquad z_t = W x_t + U h_{t-1} + b.$$

This yields the recursive bound:

$$\mathrm{Var}(h_t) \leq L^{2t} \sum_{i=0}^{t-1} U^i W \, \mathrm{Var}(x_{t-i}) \, W^\top (U^i)^\top.$$

Under RIP, only one input window contributes stochasticity, leading to:

$$\mathrm{Var}(h_t^{(\mathrm{RIP})}) \leq L^{2t} U^k W \Sigma W^\top (U^k)^\top,$$

whereas without RIP:

$$\mathrm{Var}(h_t^{(\mathrm{noRIP})}) \leq L^{2t} \sum_{i=0}^{t-1} U^i W \Sigma W^\top (U^i)^\top.$$

Thus, nonlinearities introduce a global factor $L^{2t}$ but preserve the core effect: RIP suppresses variance by limiting the number of independent stochastic inputs.

To make the connection explicit for Transformer-based models, we analyze the variance propagation inside a multi-head self-attention layer. Let

$$Q = X W_Q, \quad K = X W_K, \quad V = X W_V,$$

and consider a token sequence $X = [x_1, \ldots, x_T]$. The attention output for a query position $t$ is

$$\mathrm{Attn}_t = \sum_{j=1}^{T} \alpha_{tj} V_j, \qquad \alpha_{tj} = \frac{\exp\left(q_t k_j^\top / \sqrt{d}\right)}{\sum_{\ell=1}^{T} \exp\left(q_t k_\ell^\top / \sqrt{d}\right)}.$$

**Effect of RIP on variance.** Under RIP, all constant windows satisfy

$$\mathrm{Var}(x_j^{(\mathrm{const})}) = 0,$$

which implies

$$\mathrm{Var}(K_j^{(\mathrm{const})}) = 0, \qquad \mathrm{Var}(Q_j^{(\mathrm{const})}) = 0, \qquad \mathrm{Var}(V_j^{(\mathrm{const})}) = 0.$$

Thus, for any attention score involving a constant window,

$$\mathrm{Var}(q_t k_j^\top) = 0.$$

This has two direct consequences:

1. Attention logits become less stochastic. The logits corresponding to constant windows are deterministic, so the overall variance of the softmax distribution decreases:

$$\mathrm{Var}(\alpha_{tj}^{(\mathrm{RIP})}) < \mathrm{Var}(\alpha_{tj}^{(\mathrm{noRIP})}).$$

2. The number of stochastic value vectors decreases. Only the real window contributes non-zero variance in $V_j$.

**Variance bound.** For the attention output, we have

$$\mathrm{Var}(\mathrm{Attn}_t) = \mathrm{Var}\left(\sum_{j=1}^{T} \alpha_{tj} V_j\right) \leq \sum_{j=1}^{T} \mathbb{E}[\alpha_{tj}^2] \, \mathrm{Var}(V_j),$$

where cross-terms vanish when constant windows are deterministic.

Under RIP, only one index $j = r$ (the real window) has non-zero variance, so:

$$\mathrm{Var}(\mathrm{Attn}_t^{(\mathrm{RIP})}) \leq \mathbb{E}[\alpha_{tr}^2] \, \mathrm{Var}(V_r).$$

Without RIP, all windows have non-zero variance:

$$\mathrm{Var}(\mathrm{Attn}_t^{(\mathrm{noRIP})}) \leq \sum_{j=1}^{T} \mathbb{E}[\alpha_{tj}^2]\,\mathrm{Var}(V_j).$$

Thus, even in a Transformer, RIP reduces the variance of $Q$, $K$, and $V$ for constant windows; decreases the variance of the attention distribution; and reduces the number of stochastic value vectors contributing to the output.

Therefore, the qualitative conclusion from the RNN analysis remains valid: RIP suppresses variance in deep models by limiting the number of independent stochastic contributions, and in Transformers, this occurs through the attention mechanism rather than recurrent operators.

The variance-reduction effect of RIP does not depend on a specific duplication factor. To verify this, we repeated the derivations for $i = 2$, where the transformed sequence becomes:

$$x_1 = x_2 = x_3 = x_4\gamma \qquad , \qquad x_6 = x_7 = x_8 = x_9\gamma.$$

In this case, the final element is:

$$h_9 = \left( \sum_{j \in \{0,1,2,3,5,6,7,8\}} U^j \right) W\gamma \;+\; U^4 W x_5 \;+\; \sum_{j=0}^{8} U^j b.$$

Compared with the $i = 1$ case, this expression simply shifts the location of the single non-duplicated element. In all configurations, the variability of the hidden representation depends on *only one varying input*, while the duplicated components contribute deterministic terms.

More generally, for sequence length $T$:

$$H_t^{(\mathrm{RIP})} = \sum_{k=0}^{T-1} U^k W\gamma \;-\; U^{2k} W\gamma \;+\; U^{2k} W x_{2k+1} \;+\; \sum_{k=0}^{T-1} U^k b,$$

so the variation arises exclusively from the non-duplicated values. Without RIP, however, the hidden state depends on all $T$ variable inputs.

Thus, the variance-reduction argument holds for *any* duplication factor $i$, and the empirical choice of $\gamma$ does not affect the theoretical conclusion. We will include this clarification in the revised manuscript.

## G  ADDITIONAL EXPERIMENTS

To enable a more in-depth investigation, we performed supplementary experiments designed to address specific questions and validate our results.

### G.1  $\ell_1$ AND $\ell_2$

Since we are introducing a novel regularization strategy for Human Activity Recognition (HAR) classifiers, comparing our approach with similar techniques is essential. However, we could not identify existing regularization methods tailored to HAR scenarios. Therefore, we compared our method with the most traditional and widely used regularization techniques: $\ell_1$ and $\ell_2$ regularization.

Given the structure of the models, we applied these regularization techniques to the TS-Classifier and DClassifier models previously used in this study. The experiments were conducted using the same real and synthetic datasets employed throughout the paper. We evaluated four values for the regularization parameter: 0.1, 0.01, 0.001, and 1.

To provide a clear overview, we summarize the results for the real and synthetic datasets separately. Tables 25 and 23 present the best results obtained for each regularization technique in real and synthetic datasets, respectively. The complete experimental results are provided in tables 24 and 23.

Table 22: Performance analysis of the proposed RIP method versus a baseline approach and the $\ell_1$ and $\ell_2$ regularization methods. Experiments were conducted on five public, real-world datasets. For each metric, the highest-performing result is highlighted in bold. The reported RIP performance is based on the best hyperparameter configuration found.

| Model | Dataset | Method | Accuracy (%) | F1-score (%) |
|---|---|---|---|---|
| DClassifier | MHEALTH | Baseline | 91.21±1.61 | 90.46±2.84 |
| | | RIP ($\gamma = 0, i = 16$) | **97.84±0.21** | **98.01±0.20** |
| | | $\ell_1$ ($\epsilon = 0.001$) | 47.13±6.13 | 40.94±6.24 |
| | | $\ell_2$ ($\epsilon = 0.001$) | 44.65±6.04 | 37.84±6.03 |
| | MHAD1 | Baseline | 58.25±2.06 | 67.13±1.14 |
| | | RIP ($\gamma = 0, i = 16$) | **86.12±0.55** | **86.13±0.53** |
| | | $\ell_1$ ($\epsilon = 0.1$) | 29.36±2.29 | 26.63±2.12 |
| | | $\ell_2$ ($\epsilon = 0.001$) | 16.09±1.23 | 9.28±1.39 |
| | MHAD2 | Baseline | 68.32±1.05 | 67.58±1.31 |
| | | RIP ($\gamma = 0, i = 16$) | **82.12±0.72** | **81.68±0.74** |
| | | $\ell_1$ ($\epsilon = 0.001$) | 52.92±4.66 | 46.58±5.44 |
| | | $\ell_2$ ($\epsilon = 0.001$) | 41.56±6.21 | 30.05±8.18 |
| | WHARF | Baseline | 83.11±1.85 | 78.62±2.08 |
| | | RIP ($\gamma = 1, i = 16$) | **90.25±1.19** | **87.48±1.70** |
| | | $\ell_1$ ($\epsilon = 0.001$) | 19.51±2.09 | 10.92±1.98 |
| | | $\ell_2$ ($\epsilon = 0.001$) | 20.04±3.16 | 10.90±2.73 |
| | WISDM | Baseline | 99.47±0.09 | 99.21±0.12 |
| | | RIP ($\gamma = 0, i = 5$) | **99.77±0.03** | **99.46±0.05** |
| | | $\ell_1$ ($\epsilon = 1.0$) | 51.79±4.29 | 43.75±2.95 |
| | | $\ell_2$ ($\epsilon = 0.1$) | 48.65±5.66 | 42.77±3.40 |
| TS-Classifier | MHEALTH | Baseline | 32.37±2.40 | 24.08±2.59 |
| | | RIP ($\gamma = 1, i = 1$) | **59.98±2.03** | **55.54±2.42** |
| | | $\ell_1$ ($\epsilon = 1.0$) | 57.71±2.60 | 51.67±3.56 |
| | | $\ell_2$ ($\epsilon = 0.1$) | 58.25±3.37 | 51.61±4.45 |
| | MHAD1 | Baseline | 20.45±1.15 | 15.67±1.41 |
| | | RIP ($\gamma = -1, i = 5$) | 31.67±0.73 | 26.08±0.70 |
| | | $\ell_1$ ($\epsilon = 0.001$) | 31.49±1.95 | 27.62±2.00 |
| | | $\ell_2$ ($\epsilon = 0.001$) | **33.42±1.95** | **29.70±2.29** |
| | MHAD2 | Baseline | 31.25±1.64 | 24.46±1.72 |
| | | RIP ($\gamma = 1, i = 16$) | **52.49±3.16** | **43.22±3.91** |
| | | $\ell_1$ ($\epsilon = 0.001$) | 47.13±6.13 | 40.94±6.24 |
| | | $\ell_2$ ($\epsilon = 0.001$) | 44.65±6.04 | 37.84±6.03 |
| | WHARF | Baseline | 19.19±6.71 | 10.28±3.63 |
| | | RIP ($\gamma = 1, i = 1$) | **46.06±4.34** | **28.63±2.24** |
| | | $\ell_1$ ($\epsilon = 0.001$) | 19.51±2.09 | 10.92±1.98 |
| | | $\ell_2$ ($\epsilon = 0.001$) | 20.04±3.16 | 10.90±2.73 |
| | WISDM | Baseline | 90.32±1.63 | 87.19±2.62 |
| | | RIP ($\gamma = -1, i = 5$) | **93.12±1.21** | **92.30±1.46** |
| | | $\ell_1$ ($\epsilon = 0.001$) | 43.68±5.46 | 43.94±4.06 |
| | | $\ell_2$ ($\epsilon = 0.001$) | 47.88±6.81 | 46.12±3.21 |

As in previous sections, the baseline for each dataset is the classifier trained on the respective data type without any approach applied.

The following discussion analyzes the outcomes reported in Table 23. Due to the breadth of results and potential analyses, our discussion emphasizes the key patterns and takeaways.

Table 23: Regularization in Synthetic Datasets.

| Model | Dataset | Protocol | Rate | $(\gamma, i)$ | Acc | F1 |
|---|---|---|---|---|---|---|
| TS-Classifier | WISDM | $\ell_1$ | 0.001 | - | $43.68 \pm 5.46$ | $43.94 \pm 4.06$ |
| | | | 0.01 | - | $43.68 \pm 5.46$ | $43.94 \pm 4.06$ |
| | | | 0.1 | - | $38.06 \pm 6.39$ | $38.36 \pm 3.41$ |
| | | | 1.0 | | $32.41 \pm 9.07$ | $29.44 \pm 7.68$ |
| | | $\ell_2$ | 0.001 | - | $47.88 \pm 6.81$ | $46.12 \pm 3.21$ |
| | | | 0.01 | | $47.88 \pm 6.81$ | $46.12 \pm 3.21$ |
| | | | 0.1 | | $40.57 \pm 5.65$ | $37.77 \pm 3.71$ |
| | | | 1.0 | | $40.34 \pm 8.84$ | $38.43 \pm 4.64$ |
| | WHARF | $\ell_1$ | 0.001 | - | $19.51 \pm 2.09$ | $10.92 \pm 1.98$ |
| | | | 0.01 | - | $20.04 \pm 3.16$ | $10.90 \pm 2.73$ |
| | | | 0.1 | | $16.57 \pm 4.33$ | $10.90 \pm 3.32$ |
| | | | 1.0 | | $18.03 \pm 3.94$ | $8.19 \pm 2.58$ |
| | | $\ell_2$ | 0.001 | - | $20.04 \pm 3.16$ | $10.90 \pm 2.73$ |
| | | | 0.01 | | $20.04 \pm 3.16$ | $10.90 \pm 2.73$ |
| | | | 0.1 | | $16.57 \pm 4.33$ | $9.72 \pm 3.32$ |
| | | | 1.0 | | $18.03 \pm 3.94$ | $8.19 \pm 2.58$ |
| | MHEALTH | $\ell_1$ | 0.01 | - | $56.44 \pm 3.35$ | $50.95 \pm 3.05$ |
| | | | 0.001 | - | $56.44 \pm 3.35$ | $50.95 \pm 3.05$ |
| | | | 0.1 | | $48.93 \pm 3.26$ | $44.13 \pm 3.67$ |
| | | | 1.0 | | $31.07 \pm 5.40$ | $21.14 \pm 4.92$ |
| | | $\ell_2$ | 0.001 | - | $59.80 \pm 4.52$ | $59.05 \pm 3.89$ |
| | | | 0.01 | | $59.80 \pm 4.52$ | $59.05 \pm 3.89$ |
| | | | 0.1 | | $54.87 \pm 4.33$ | $50.14 \pm 3.42$ |
| | | | 1.0 | | $48.22 \pm 2.61$ | $43.18 \pm 2.90$ |
| | MHAD2 | $\ell_2$ | 0.001 | - | $47.13 \pm 6.13$ | $40.94 \pm 6.24$ |
| | | | 0.01 | | $47.13 \pm 6.13$ | $40.94 \pm 6.24$ |
| | | | 0.1 | | $38.98 \pm 3.72$ | $33.44 \pm 3.59$ |
| | | | 1.0 | | $39.74 \pm 3.45$ | $34.13 \pm 3.84$ |
| | | $\ell_1$ | 0.001 | - | $44.65 \pm 6.04$ | $37.84 \pm 6.03$ |
| | | | 0.01 | | $44.65 \pm 6.04$ | $37.84 \pm 6.03$ |
| | | | 0.1 | | $39.83 \pm 4.48$ | $33.64 \pm 4.03$ |
| | | | 1.0 | | $20.49 \pm 0.74$ | $5.67 \pm 0.17$ |
| | MHAD1 | $\ell_1$ | 0.001 | - | $31.49 \pm 1.95$ | $27.62 \pm 2.00$ |
| | | | 0.01 | | $31.49 \pm 1.95$ | $27.62 \pm 2.00$ |
| | | | 0.1 | | $26.13 \pm 1.92$ | $23.92 \pm 1.88$ |
| | | | 1.0 | | $6.41 \pm 0.23$ | $0.57 \pm 0.02$ |
| | | $\ell_2$ | 0.001 | - | $33.42 \pm 1.95$ | $29.70 \pm 2.29$ |
| | | | 0.01 | | $33.42 \pm 1.95$ | $29.70 \pm 2.29$ |
| | | | 0.1 | | $28.10 \pm 2.95$ | $24.93 \pm 2.68$ |
| | | | 1.0 | | $26.40 \pm 2.45$ | $22.73 \pm 2.72$ |

**Performance in High-Baseline Scenarios.** One of its key advantages is its ability to enhance performance in datasets with high baseline metrics. For instance, in the MHEALTH dataset with TS-Classifier, where the baseline accuracy is close to 99%, traditional regularization methods led to performance drops of up to 40 percentage points. In contrast, configurations maintained the baseline and, in some cases, reached 100% accuracy and F1. Similarly, in the WISDM dataset, which also showed strong baseline results, improved accuracy and F1 by one percentage point, whereas $\ell_1$ and $\ell_2$ regularization resulted in significant degradation. These findings suggest that it does not compromise performance in already high-performing models, an essential feature for sensitive or high-stakes applications.

**Robustness Across Diverse Data Distributions** Consistently outperformed traditional regularization methods across all five datasets tested with TS-Classifier and three out of five with DClassifier. Notably, neither $\ell_1$ nor $\ell_2$ regularization surpassed the baseline in any of the evaluated scenarios,

Table 24: Regularization in Synthetic Datasets.

| Model | Dataset | Protocol | Rate | $(\gamma, i)$ | Acc | F1 |
|---|---|---|---|---|---|---|
| | | $\ell_2$ | 1.0 | - | $51.79 \pm 4.29$ | $43.75 \pm 2.95$ |
| | | | 0.001 | | $51.69 \pm 5.22$ | $45.40 \pm 4.09$ |
| | | | 0.01 | | $51.69 \pm 5.22$ | $45.40 \pm 4.09$ |
| | | | 0.1 | | $46.96 \pm 6.40$ | $42.36 \pm 4.90$ |
| | WISDM | $\ell_1$ | 1.0 | - | $48.65 \pm 5.66$ | $42.77 \pm 3.40$ |
| | | | 0.001 | | $48.89 \pm 6.30$ | $43.58 \pm 3.06$ |
| | | | 0.01 | | $48.89 \pm 6.30$ | $43.58 \pm 3.06$ |
| | | | 0.1 | | $50.74 \pm 5.55$ | $45.59 \pm 3.53$ |
| | | | 1.0 | | $48.65 \pm 5.66$ | $42.77 \pm 3.40$ |
| | | $\ell_2$ | 0.1 | - | $18.36 \pm 4.51$ | $9.62 \pm 3.42$ |
| | | | 0.001 | | $17.36 \pm 3.85$ | $7.38 \pm 2.53$ |
| | | | 0.01 | | $17.36 \pm 3.85$ | $7.38 \pm 2.53$ |
| | | | 1.0 | | $16.62 \pm 4.60$ | $7.29 \pm 2.73$ |
| DClassifier | WHARF | $\ell_1$ | 0.001 | - | $19.45 \pm 3.92$ | $\mathbf{9.16 \pm 2.53}$ |
| | | | 0.01 | | $19.45 \pm 3.92$ | $9.16 \pm 2.53$ |
| | | | 0.1 | | $17.23 \pm 3.43$ | $9.11 \pm 1.15$ |
| | | | 1.0 | | $10.87 \pm 3.72$ | $7.88 \pm 2.31$ |
| | | $\ell_2$ | 0.1 | - | $15.06 \pm 1.64$ | $3.85 \pm 3.27$ |
| | | | 0.001 | | $14.90 \pm 1.34$ | $3.65 \pm 2.90$ |
| | | | 0.01 | | $14.90 \pm 1.34$ | $3.65 \pm 2.90$ |
| | | | 1.0 | | $15.85 \pm 3.09$ | $4.10 \pm 3.73$ |
| | | $\ell_2$ | 1.0 | - | $57.71 \pm 2.60$ | $51.67 \pm 3.56$ |
| | | | 0.1 | | $55.83 \pm 2.65$ | $50.24 \pm 3.54$ |
| | | | 0.01 | | $54.84 \pm 4.09$ | $47.91 \pm 3.91$ |
| | | | 0.001 | | $54.84 \pm 4.09$ | $47.91 \pm 3.91$ |
| | MHEALTH | $\ell_1$ | 0.1 | - | $58.25 \pm 3.37$ | $51.61 \pm 4.45$ |
| | | | 0.001 | | $56.64 \pm 2.79$ | $49.45 \pm 2.98$ |
| | | | 0.01 | | $56.64 \pm 2.79$ | $49.45 \pm 2.98$ |
| | | | 1.0 | | $53.84 \pm 2.27$ | $48.40 \pm 3.30$ |
| | | $\ell_2$ | 0.001 | - | $52.92 \pm 4.66$ | $46.58 \pm 5.44$ |
| | | | 0.01 | - | $52.92 \pm 4.66$ | $46.58 \pm 5.44$ |
| | | | 0.1 | - | $51.67 \pm 5.36$ | $45.96 \pm 5.94$ |
| | | | 1.0 | | $50.10 \pm 4.64$ | $44.60 \pm 3.98$ |
| | MHAD2 | $\ell_1$ | 0.001 | - | $41.56 \pm 6.21$ | $30.05 \pm 8.18$ |
| | | | 0.01 | - | $41.56 \pm 6.21$ | $30.05 \pm 8.18$ |
| | | | 0.1 | | $42.29 \pm 4.07$ | $30.05 \pm 8.18$ |
| | | | 1.0 | | $38.75 \pm 1.91$ | $26.05 \pm 3.45$ |
| | | $\ell_2$ | 0.1 | - | $29.36 \pm 2.29$ | $26.63 \pm 2.12$ |
| | | | 0.001 | | $28.05 \pm 1.64$ | $25.25 \pm 1.66$ |
| | | | 0.01 | | $28.05 \pm 1.64$ | $25.25 \pm 1.66$ |
| | | | 1.0 | | $27.68 \pm 1.84$ | $25.03 \pm 1.78$ |
| | MHAD1 | $\ell_1$ | 0.001 | - | $16.09 \pm 1.23$ | $9.28 \pm 1.39$ |
| | | | 0.01 | | $16.09 \pm 1.23$ | $9.28 \pm 1.39$ |
| | | | 0.1 | | $15.24 \pm 1.23$ | $7.66 \pm 1.70$ |
| | | | 1.0 | | $15.51 \pm 1.16$ | $8.31 \pm 1.52$ |

yielding improvements in approximately 90% of the cases. For instance, in the WHARF dataset, traditional regularization caused performance drops exceeding 50 percentage points, while maintaining or improving performance across all configurations.

In datasets with moderate or low baseline performance, such as MHAD1 and MHAD2, RIP delivered substantial gains: up to 30 percentage points in TS-Classifier and around 4 points in DClassifier. In contrast, $\ell_1$ and $\ell_2$ regularization failed to improve results and occasionally degraded them. These findings highlight the effectiveness as a generalization-enhancing mechanism, particularly in scenarios where conventional regularization techniques fall short.

Table 25: Regularization in Real Datasets.

| Model | Dataset | Protocol | Rate | $(\gamma, i)$ | Acc | F1 |
|-------|---------|----------|------|---------------|-----|-----|
| | | $\ell_1$ | 0.001 | - | 82.36±6.46 | 80.62± 6.21 |
| | WISDM | $\ell_2$ | 0.001 | - | 80.51±4.93 | 74.78± 7.45 |
| | | $\ell_1$ | 0.001 | - | 34.11 ± 7.97 | 20.82 ± 3.44 |
| TS-Classifier | WHARF | $\ell_2$ | 0.01 | - | 32.17 ± 8.85 | 18.33 ± 4.96 |
| | | $\ell_1$ | 0.001 | - | 30.47 ± 1.66 | 22.54 ±1.15 |
| | MHEALTH | $\ell_2$ | 0.001 | - | 31.31 ± 3.40 | 22.46 ± 3.96 |
| | | $\ell_1$ | 0.001 | - | 30.47 ± 1.66 | 22.64 ± 1.15 |
| | MHAD2 | $\ell_2$ | 0.001 | - | 31.31 ± 3.40 | 23.26 ± 3.96 |
| | | $\ell_1$ | 0.001 | - | 18.98 ± 1.34 | 13.79 ±1.55 |
| | MHAD1 | $\ell_2$ | 0.001 | - | 20.10 ± 2.11 | 15.10 ± 2.45 |
| | | $\ell_1$ | 0.001 | - | 96.94 ± 0.26 | **95.56 ± 0.37** |
| | WISDM | $\ell_2$ | 0.01 | - | 93.91 ± 0.46 | 91.29 ± 0.64 |
| | | $\ell_1$ | 1 | - | 78.38 ±2.48 | 70.44 ± 2.61 |
| DClassifier | WHARF | $\ell_2$ | 0.001 | - | 68.23 ± 1.75 | 52.96 ± 3.36 |
| | | $\ell_1$ | 0.01 | - | 91.78 ± 0.94 | 91.64 ± 1.30 |
| | MHEALTH | $\ell_2$ | 0.1 | - | 85.55 ± 1.54 | 80.82 ± 3.06 |
| | | $\ell_1$ | 0.1 | - | 52.75 ± 1.59 | 50.89 ± 1.34 |
| | MHAD2 | $\ell_2$ | 0.01 | - | 15.55 ± 1.40 | 6.96 ± 1.22 |
| | | $\ell_1$ | 0.01 | - | 69.27 ± 2.30 | 62.04 ± 3.11 |
| | MHAD1 | $\ell_2$ | 0.001 | - | 55.00 ± 5.09 | 43.87 ± 6.68 |

**Versatility Across Configurations**   Another key strength is its versatility across various configurations. Across all datasets, multiple combinations of hyperparameters ($\gamma$ and $i$ values) led to substantial performance gains, indicating that it is not overly sensitive to fine-tuning. For instance, $\gamma = -1$ and $\gamma = 1$ yielded strong results across datasets such as MHAD1, WHARF, and WISDM. This suggests broad applicability and that it typically requires only the identification of a reasonable $\gamma$ value to achieve competitive performance.

**Practicality Over Complexity**   In the DClassifier setting, datasets such as WISDM, WHARF, and USCHAD exhibited only marginal improvements, approximately one percentage point over the baseline, when using $\ell_2$ regularization. In one of these cases, it did not outperform the baseline; however, it achieved its best results with low values of $i$, still surpassing the baseline in several configurations. This observation reinforces a concern previously raised by Jiang et al. (2024) regarding the complexity of designing deep learning models. Traditional regularization techniques often require substantial expertise and careful hyperparameter tuning to be effective. In contrast, RIP offers a more practical and accessible alternative for practitioners, striking a balance between ease of use and strong empirical performance.

### G.1.1   REAL DATA

**Comparative Efficacy Across Models and Datasets**   Our experimental results across five diverse activity recognition datasets demonstrate that the RIP method generally outperforms traditional regularization techniques such as $\ell_1$ and $\ell_2$, particularly when applied to the TS-Classifier. In all datasets (WISDM, WHARF, MHAD2, USCHAD, and MHEALTH), RIP consistently achieves superior performance in both F1-score and accuracy. Notably, on the MHEALTH dataset, the configuration ($\gamma = 0, i = 1$) yielded approximately 30 percentage-point gains over the baseline. For the DClassifier, RIP also surpassed traditional regularization in most cases. In datasets such as WHARF, USCHAD, MHEALTH, and MHAD2, RIP consistently improved performance, whereas conventional methods either failed to improve performance or offered only marginal gains. The only exception was the WISDM dataset (the largest in our study, with over 10,000 training samples), where $\ell_2$ regularization outperformed RIP. This suggests that conventional techniques may still prove effective in specific scenarios, especially those involving large and well-structured datasets. Nonetheless, such cases were infrequent, reaffirming the general efficacy of RIP across models and datasets.

**RIP Outperforms Traditional Regularization**   In most scenarios, RIP delivered substantial performance gains over traditional regularization techniques. For example, on the MHEALTH dataset with the TS-Classifier, RIP improved the F1-score by approximately 50 percentage points and accuracy by around 30% compared to the best $\ell_1$ configuration. In the case of MHAD2, RIP enhanced performance in 13 out of 16 configurations, while traditional regularization methods showed no improvement. Furthermore, on datasets such as USCHAD and MHEALTH, traditional regularization consistently degraded performance, leading to reductions of up to 40 percentage points, whereas RIP improved both accuracy and F1-score across all tested configurations. These consistent gains across diverse datasets with varying levels of complexity underscore the robustness and broad applicability of RIP as an effective regularization strategy.

**Generalization Across Architectures**   One of the most promising aspects of RIP is its ability to generalize across different model architectures. When applied to both TS-Classifier and DClassifier, RIP proved effective across various configurations (e.g., other values of $\gamma$ and $i$), demonstrating flexibility in both tuning and deployment. Although its impact was more pronounced in TS-Classifier, the consistent improvements observed with DClassifier indicate that RIP adapts well to different architectures while maintaining its effectiveness. This versatility is particularly valuable in scenarios where traditional regularization techniques fail, offering a robust alternative that does not rely on architecture-specific or loss-based adjustments.

### G.1.2   SYNTHETIC VS. REAL DATA

Across all datasets and scenarios, constant data augmentation consistently outperformed both $\ell_1$ and $\ell_2$ regularization techniques. Even in cases where performance gains were marginal, constant data never degraded the model's baseline performance (unlike regularization), highlighting its effectiveness and reliability as a strategy to enhance synthetic data quality in Human Activity Recognition (HAR) tasks. RIP proved a reliable and adaptable technique, delivering consistent performance improvements across all tested datasets and configurations. Unlike standard regularization methods, RIP can boost weak baselines, preserve strong ones, and maintain robustness across data variations, making it a strong candidate for general-purpose regularization in deep learning pipelines.

The following discussion analyzes the outcomes reported in Table 25. Due to the breadth of results and potential analyses, our discussion emphasizes the key patterns and takeaways.

### G.2   GENERAL TIME-SERIES DATA RESULTS

We extended the previous experiments conducted on wearable sensor data to more general time-series datasets. The models employed were time-series classification methods, including TSBF, TSRF, and TS-Classifier. The only exception was D-Classifier, which is specific to HAR. We retained it for consistency with prior experiments.

Specifically, we evaluated the following datasets: the Hydraulic dataset Helwig et al. (2015), which comprises non-wearable sensor data used for predictive maintenance in hydraulic systems; the Eye dataset Roesler (2013), containing 14 EEG-derived features used to detect eye states; and the Occupancy dataset Candanedo (2016), composed of 14 environmental sensor features aimed at determining room occupancy. Since identifying the best generative models for each dataset lies beyond the scope of this study, we restricted our experiments to constant data augmentation within the TRTR protocol.

All datasets were preprocessed following the same methodology as in the main experiments. We applied a 10-fold stratified cross-validation split with 90% of the data for training and 10% for testing. Each dataset was segmented using the SNOW method, with 50-time-step windows and 50% overlap. This window size was selected based on preliminary validation: among a randomly selected dataset, TSBF (our best-performing model overall) achieved its baseline using this configuration.

We explored the same range of hyperparameters as in the main experiments, testing $\gamma \in 0, 1, -1, 5$ and $i \in 1, 5, 16, 32$. Table 26 presents only the best-performing configuration for each experiment.

Table 26 presents the results of extending RIP to non-wearable time-series datasets. Overall, the impact of RIP in these settings was less pronounced compared to the wearable-focused datasets, with mixed outcomes depending on the dataset and model architecture.

| Dataset | Model | Setup | F1 |
|---|---|---|---|
| Eye | TSBF | Baseline | $45.83 \pm 3.74$ |
| | | $(\gamma = 1, i = 5)$ | $\mathbf{47.77 \pm 6.65}$ |
| | TSRF | Baseline | $55.49 \pm 5.68$ |
| | | $(\gamma = 1, i = 1)$ | $57.22 \pm 4.84$ |
| | DClassifier | Baseline | $34.73 \pm 2.84$ |
| | | $(\gamma = -1, i = 16)$ | $35.17 \pm 2.74$ |
| | TS-Classifier | Baseline | $37.60 \pm 5.45$ |
| | | $(\gamma = 0, i = 5)$ | $37.18 \pm 6.11$ |
| Haydraulic | TSBF | Baseline | $72.00 \pm 1.97$ |
| | | $(\gamma = 1, i = 16)$ | $72.04 \pm 1.68$ |
| | TSRF | Baseline | $75.69 \pm 1.68$ |
| | | $(\gamma = 1, i = 5)$ | $75.60 \pm 1.92$ |
| | DClassifier | Baseline | $14.31 \pm 0.15$ |
| | | $(\gamma = 1, i = 1)$ | $13.43 \pm 0.03$ |
| | TS-Classifier | Baseline | $36.42 \pm 2.70$ |
| | | $(\gamma = -1, i = 5)$ | $14.13 \pm 2.36$ |
| Occupancy | TSBF | Baseline | $87.53 \pm 1.73$ |
| | | $(\gamma = 1, i = 5)$ | $87.49 \pm 1.74$ |
| | DClassifier | Baseline | $12.74 \pm 1.21$ |
| | | $(\gamma = -1, i = 1)$ | $10.46 \pm 2.11$ |
| | TS-Classifier | Baseline | $15.61 \pm 2.54$ |
| | | $(\gamma = 1, i = 1)$ | $20.74 \pm 3.87$ |

Table 26: Experiments with general time-series data

For the Eye dataset, marginal improvements were observed with the TSBF and TSRF models. Specifically, TSBF slightly increased the F1 score from 45.83% to 47.77%, and TSRF from 55.49% to 57.22%. While these gains suggest a potential benefit from CAR, the overlapping standard deviations indicate that the improvements may not be statistically significant. Conversely, D-Classifier and TS-Classifier exhibited minimal to no improvement, suggesting that RIP is ineffective for these models in this dataset.

In the Hydraulic dataset, RIP yielded performance nearly identical to the baseline for TSBF and TSRF, with negligible changes. Notably, TS-Classifier experienced a substantial decrease in performance, from 36.42% to 14.13%, indicating that RIP may negatively affect models when signal characteristics deviate substantially from those typically observed in wearable sensor data. D-Classifier also slightly underperformed in this setting when RIP was applied.

The Occupancy dataset produced mixed results. TSBF and D-Classifier matched or slightly underperformed relative to the baseline. Interestingly, TS-Classifier showed a notable improvement in F1 score, rising from 15.61% to 20.74%, suggesting that RIP can still offer benefits in specific scenarios, even outside wearable contexts. However, this improvement appears both dataset- and model-specific, limiting its generalizability.

These findings indicate that while RIP may retain some utility in general time-series scenarios, its effectiveness is considerably more evident in datasets derived from wearable sensors. Its applicability to non-wearable domains appears to be limited and highly dependent on the nature of the data and the model used. This supports the view that RIP is best suited as a regularization method for wearable sensor applications.

### G.3 TABULAR DATA RESULTS

To perform experiments using tabular data, we considered nine datasets: Absenteeism Martiniano & Ferreira (2018), Bank Moro et al. (2012), Diabetes Kahn, Glass German (1987), Iris Fisher (1988), Wine Quality Cortez et al. (2009) (comprising both white and red wine), and Sonar Rossi & Ahmed (2015). Further details on the datasets are provided in the Appendix.

We employed six classifiers: Decision Tree (DTree), Dummy, K-Nearest Neighbors (KNN), Multi-Layer Perceptron (MLP), Support Vector Machine (SVM), and Stochastic Gradient Descent (SGD).

The goal was to experimentally evaluate whether the proposed RIP technique positively affects classifier performance when applied to tabular data.

We assessed the technique using only real datasets, as generating synthetic data would require additional effort to determine the best-performing generative model, an objective beyond the scope of this work.

Initially, we tested the same $\gamma$ values used in previous experiments; however, they did not produce any variation from the baseline. Therefore, we conducted an Exploratory Data Analysis (EDA) on each dataset and adjusted the constant distributions accordingly. As a result, different values were tested for each dataset. Since RIP did not affect all datasets, we report only those where it had a measurable impact.

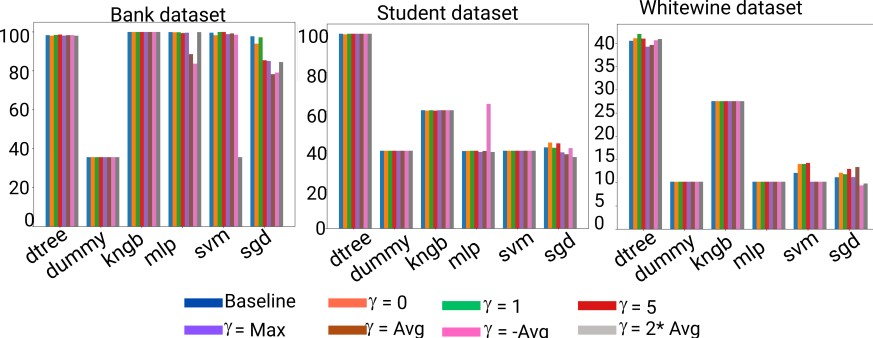

Figure 8: F1-scores per model for the Student, Wine Quality, White Wine, and Bank datasets. The x-axis displays the models, and the y-axis shows the corresponding F1 Scores. Each bar represents a different data distribution, distinguished by color and identified in the legend located to the left of the figure. The label "Avg" corresponds to the average of the features for the respective dataset.

Figure 8 reveals that although effective in some sensor-based and time-series contexts, this technique does not generalize as easily to structured tabular data. Our experiments with datasets such as Bank, White Wine Quality, and Student show that most classifiers either maintained or declined in performance when trained with constant-value data augmentations. In particular, Dummy classifiers consistently underperformed with RIP across all datasets and distributions, reinforcing the idea that naïve models are not robust to even minor changes in data structure. More importantly, SVM and MLP, typically more sensitive to data distribution changes, showed degraded F1 scores in several cases. While SGD exhibited more stable behavior, its performance was still unable to exceed the baseline in most settings. These findings suggest that RIP does not serve as a universal regularization strategy for tabular data, likely because such datasets are structured and often low-dimensional.

Although RIP can be applied to tabular datasets, its benefits are limited and highly context-dependent. Nonetheless, its application should be critically evaluated on a case-by-case basis, especially when dealing with classifiers sensitive to distributional changes.

### G.4    CUTMIX, MIXUP AND DRO

Our literature review found a notable lack of research focusing on regularization or domain generalization for Human Activity Recognition (HAR). The closest related work is Bento et al. (2023), which applied Mixup and Distributionally Robust Optimization (DRO) to accelerometer-based HAR. This direction aligns with our goal of enhancing model robustness.

While our proposed method (RIP) and DRO are not direct competitors, they aim to solve the same problem: robustness to shifts in data distribution. However, they operate in fundamentally different ways. DRO is an optimization-based approach that modifies the loss function to minimize performance on a worst-case distribution from a defined uncertainty set. Given that DRO is one of the few approaches specifically explored for robustness in the HAR literature, we deemed it an essential benchmark for our experiments.

Table 27: DClassifier - synthetic

| acc | recall | f1 | epsilon | method |
|-----|--------|-----|---------|--------|
| $45.38 \pm 2.83$ | $44.03 \pm 3.20$ | $41.80 \pm 4.05$ | 0.1 | cutmix |
| $44.04 \pm 3.85$ | $42.41 \pm 4.03$ | $40.25 \pm 4.62$ | 0.3 | cutmix |
| $44.93 \pm 3.21$ | $43.07 \pm 3.62$ | $40.86 \pm 4.68$ | 0.2 | cutmix |
| $31.58 \pm 3.28$ | $31.09 \pm 2.86$ | $24.90 \pm 3.61$ | 0.2 | dro |
| $32.70 \pm 4.71$ | $32.65 \pm 4.38$ | $26.88 \pm 4.99$ | 0.3 | dro |
| $29.94 \pm 2.77$ | $29.28 \pm 2.60$ | $22.85 \pm 2.85$ | 0.1 | dro |
| $39.67 \pm 6.03$ | $36.69 \pm 7.16$ | $32.02 \pm 9.51$ | 0.2 | mixup |
| $39.78 \pm 6.23$ | $36.78 \pm 7.33$ | $31.79 \pm 9.28$ | 0.3 | mixup |
| $39.88 \pm 6.14$ | $37.01 \pm 7.26$ | $32.33 \pm 9.63$ | 0.1 | mixup |

Table 28: D-Classifier - real

| acc | recall | f1 | epsilon | method |
|-----|--------|-----|---------|--------|
| $63.10 \pm 1.18$ | $62.26 \pm 1.37$ | $62.03 \pm 1.47$ | 0.1 | cutmix |
| $58.29 \pm 1.79$ | $57.19 \pm 1.88$ | $57.33 \pm 2.26$ | 0.3 | cutmix |
| $60.78 \pm 0.75$ | $59.50 \pm 0.76$ | $59.85 \pm 0.89$ | 0.2 | cutmix |
| $40.36 \pm 3.81$ | $39.13 \pm 4.28$ | $34.73 \pm 4.32$ | 0.2 | dro |
| $45.75 \pm 2.08$ | $43.44 \pm 2.34$ | $38.91 \pm 2.76$ | 0.3 | dro |
| $33.88 \pm 3.00$ | $31.80 \pm 3.09$ | $25.93 \pm 4.09$ | 0.1 | dro |
| $44.31 \pm 10.62$ | $41.71 \pm 11.80$ | $36.68 \pm 13.98$ | 0.2 | mixup |
| $46.49 \pm 9.79$ | $44.38 \pm 10.82$ | $39.71 \pm 12.76$ | 0.3 | mixup |
| $39.98 \pm 12.17$ | $36.95 \pm 13.48$ | $30.89 \pm 16.12$ | 0.1 | mixup |

The work by Bento et al. (2023) also leverages techniques like Mixup and Cutmix, data augmentation, and regularization methods proven highly effective in computer vision. Their main objective is to create new, synthetic training examples from existing data, forcing the model to learn more robust representations and generalize better, thereby preventing overfitting. Their adaptation to time series follows the same logic as in computer vision:

- **Mixup** creates a new training sample by combining two existing samples through a weighted linear interpolation.

- **Cutmix** creates a new training sample by cutting a temporal segment from one example and pasting it onto another.

Instead of operating on pixels and regions of a 2D image, these methods are applied to timesteps and segments of a 1D signal.

The source code for our implementation and experiments is publicly available on our GitHub repository. These comparison experiments were conducted exclusively on the DClassifier based on previous results. We used the MHAD2 dataset to ensure consistency with our supplementary experiments. Tables 27 and 28 show the full results. The comments are available in the main paper.

# H   ABLATIONS

In this section, we conduct ablation studies to evaluate the most sensitive components of our proposed method. These analyses aim to clarify our methodological choices and to justify the approach's effectiveness when the protocol is rigorously followed. We do not focus here on evaluating specific values of $\gamma$ or $i$, or on dataset-specific behaviors, as these aspects have already been thoroughly discussed in the Results and Discussion sections, where their individual effects on performance were highlighted.

Our method is grounded in the principle of effectively training models using synthetic data by narrowing the domain gap through domain randomization, as proposed by Tremblay et al. (2018) Tremblay et al. (2018). That study demonstrated that injecting random backgrounds into synthetic

images helped models focus on the relevant features (e.g., cars) by reducing their sensitivity to irrelevant details (e.g., backgrounds). Drawing on this, our RIP technique aims to construct a modular textual representation that directs the model's attention to semantically relevant information for classification. This is achieved by embedding the target signal within constant distributions defined by preselected $\gamma$ values, effectively "framing" the informative region of the input, much like a picture frame draws focus to the subject within an image. From this central idea, the key components of our approach were developed.

To further clarify the impact of these components, we conducted the following additional experiments:

1. We assessed RIP in $\sigma = (\gamma, \tilde{i}, \text{MHAD2}, \text{DClassifier})$ with $\gamma \in \{0, 1\}$ and $\tilde{i} = \frac{1}{2}i$ such $i \in \{1, 5, 16, 32\}$.

2. Instead of keeping $\gamma$ invariant across the entire shape $(\omega, 3)$, we replaced it with a random distribution $D$ of the same shape $(\omega, 3)$, under different value constraints. These values were randomly assigned while preserving the shape of the default temporal window. We evaluated the following configurations:

   - $D = \text{rand}()$: random values without any range restriction.
   - $D = \text{rand}(0, 4)$: random values in the range [0,4].
   - $D = \text{rand}(0, 1)$: random values in the range [0,1].
   - $D = \text{Dup}(x_i)$, where $Dup$ corresponds to make $i$ duplicates of the sample $x_i$, for all $i \in len(D)$.
     For these experiments, we have a setting $\tilde{\sigma} = (\text{D}, i, \text{MHAD2}, \text{DClassifier})$.

To assist interpretation of the results, the column labeled $\gamma$ is filled only when constant values are used, while the column D is filled only when random distributions are applied. The rows labeled Baseline and TSTR correspond to training using the standard synthetic dataset without RIP and serve as reference points for improvement. Since these experiments use only the synthetic MHAD2 dataset, the baselines are specific to it. A table with baseline results from synthetic and real datasets for each benchmark is available in the **??**. The column Experiment ID refers to the experiment numbers listed above.

**Are simple data copies sufficient?**  Not quite. Table 29 highlights the limitations of this approach. While the best result using direct duplication was achieved with $i = 5$ (replicating each original window five times), the observed improvement over the synthetic baseline (TSTR) remains marginal and falls within the range of standard deviation. This suggests that the gain is not statistically significant. Moreover, increasing the duplication factor beyond this point (e.g., $i = 16$ or $i = 32$) does not enhance performance and, in some cases, slightly degrades it.

This outcome is likely attributed to overfitting. Repeating identical samples increases the likelihood that the model memorizes specific training patterns instead of learning generalizable representations. This issue is particularly critical in time-series data, where minor variations in temporal dynamics often encode essential information for classification. Duplication fails to introduce such variations, potentially biasing the model toward narrow and overconfident hypotheses.

**Is $i$ as a duplication factor necessary?**  Yes. Setting $\tilde{i} = \frac{1}{2}i$, which implies using only a single copy when $i = 1$, results in a RIP structure of the form $\gamma_{i,j}, x_{i,j}, \gamma_{i,j}$, as opposed to the full configuration $\gamma_{i,j}, \gamma_{i,j}, x_{i,j}, \gamma_{i,j}, \gamma_{i,j}$. This minimal configuration performs worse than the synthetic baseline, as shown in Table 29. In contrast, incorporating duplication yields improvements of up to 4 percentage points in F1 score (see Figure **??**). For example, with the DClassifier model on the MHAD2 dataset using the full RIP setup, the F1 score reaches 55%, significantly narrowing the gap with the model trained on real data. Without duplication, such improvement does not occur; rather, the performance gap widens.

An additional observation is that even without duplication, increasing $i$ still yields performance gains. We hypothesize that this is due to a "widening" of the contextual frame, which further emphasizes the central input window and helps the model focus on relevant features. However, this effect alone does not match the benefits achieved through duplication. This may be because the

| Experiment ID. | $\gamma$ | D | $i$ | F1 |
|---|---|---|---|---|
| Baseline | - | - | - | 50.80±5.36 |
| 1. | 0 | - | 1 | 45.64±5.80 ↓ |
|  |  | - | 5 | 49.42±3.36 ↓ |
|  |  | - | 16 | 49.42±3.36 ↓ |
|  |  | - | 32 | 48.63±3.27 ↓ |
|  | 1 | - | 1 | 48.54±4.09 ↓ |
|  |  | - | 5 | **52.77±3.97** ↑ |
|  |  | - | 16 | 49.42±3.36 ↓ |
|  |  | - | 32 | 48.63±3.27 ↓ |
| 2. | - | rand() | 1 | 45.76±5.92 ↓ |
|  | - |  | 5 | 48.83±6.03 ↓ |
|  | - |  | 16 | 50.67±5.59 ↓ |
|  | - |  | 32 | 49.08±3.54 ↓ |
|  | - | rand(0,4) | 1 | 48.54±4.09 ↓ |
|  | - |  | 5 | **52.74±3.97** ↑ |
|  | - |  | 16 | 49.45±5.50 ↓ |
|  | - |  | 32 | 49.79±3.68 ↓ |
|  | - | rand(0,1) | 1 | 45.64±5.80 ↓ |
|  | - |  | 5 | 49.42±3.36 ↓ |
|  | - |  | 16 | **51.67±4.30** ↑ |
|  | - |  | 32 | 48.63±3.27 ↓ |
|  | - | Dup(x) | 1 | 50.69±6.00 ↓ |
|  | - |  | 5 | **52.43±4.50** ↑ |
|  | - |  | 16 | 50.06±4.61 ↓ |
|  | - |  | 32 | 50.07±5.26 ↓ |

Table 29: Ablation experiments.

surrounding synthetic context, without sufficient repetition, remains too weak to reliably steer the model's attention and enhance generalization.

**Is using a random distribution instead of a constant one sufficient?** No. While the configuration with $\gamma = \text{rnd}(0, 4)$ and $i = 5$ yielded the best performance among the randomized setups, this improvement is not statistically significant. The maximum value obtained (mean plus one standard deviation) only matches the best result from the baseline, rather than surpassing it. All other configurations failed to reach baseline performance, resulting in degraded results in most cases. We attribute this behavior to the model's difficulty in distinguishing non-stationary, randomly generated distributions from meaningful temporal patterns. This confusion hinders the model's ability to learn accurate representations and, therefore, to generalize effectively. Synthetic data often omits parts of the true data distribution, focusing disproportionately on specific aspects Shumailov et al. (2024). When random, non-stationary distributions are added, they may inadvertently resemble patterns belonging to other classes, leading the model to form misleading associations. As a result, rather than enhancing robustness, this strategy tends to compromise the model's generalization ability.

We also conducted additional experiments using RIP without adhering to the required ordering of $\gamma_{i,j}, \gamma_{i,j}, x_{i,j}, \gamma_{i,j}, \gamma_{i,j}$ for $\gamma$. Specifically, we tested configurations where $\sigma = (\gamma, i, \text{MHAD2}, \text{DClassifier})$ with $\gamma = \text{Avg(features)}$ and $i \in 1, 5, 16, 32$. Furthermore, we explored using non-integer values for $\gamma$ under the same setting.

Given that the results from these experiments were substantially lower than expected, we opted not to include them in the main paper. When the required sequence was not followed, performance deteriorated significantly, indicating that simply appending constant distributions in an unstructured manner can mislead the model. This is particularly problematic when the class labels remain unchanged, potentially causing the model to associate unrelated patterns with a specific class.

Similarly, non-integer values for $\gamma$, such as the average feature values, did not yield meaningful improvements. The performance was indistinguishable from or worse than the baseline in several cases. A comparable trend was also observed in experiments involving real data, where

$\gamma = \text{Avg(features)}$ failed to produce any noticeable performance gain. These results reinforce the importance of both the structure and the design of the contextual frame in RIP for achieving practical model training.