# OpenReview forum: "Regularization via Invariant Patterns: Temporal Domain Randomization for Human Activity Recognition"
_ICLR.cc/2026/Conference — Submitted to ICLR 2026_

### Official Review · Reviewer_7CbN · 2025-10-27

**Soundness:** 3
**Presentation:** 3
**Contribution:** 2
**Rating:** 6
**Confidence:** 4

**Summary:**

This paper proposes a regularization method (RIP) that augments each time-series sample by framing it with constant-valued segments. The goal is to reduce reliance on spurious temporal context, particularly in synthetic data settings. RIP is architecture-agnostic and improves generalization in both synthetic-to-real (TSTR) and real-only (TRTR) setups. Evaluation across five HAR datasets and multiple models shows consistent gains in F1 and calibration.

**Strengths:**

- Addresses an underexplored and practical problem: generalization from GAN-generated HAR data.
- The method is simple, lightweight, and broadly applicable.
- Experiments are extensive, covering multiple datasets and models with consistent improvement.
- Calibration and stability analyses complement accuracy results.
- The approach requires minimal tuning and introduces no architectural complexity.

**Weaknesses:**

1. **Terminology clarity**
   `S`, `R`, `TRTR`, Duplication factor `i` and constant value `γ` appear early (e.g., line 62) without explanation. These should be briefly defined when first introduced.

2. **Related Work is incomplete**
   The paper omits relevant work in all three subsections. For example:
   - Domain randomization in time-series: *PhASER* (Mohapatra et al., ICLR 2025), Cutout (Yang et al., ESWA 2022)
   - Regularization for HAR: *TS-TCC* (Eldele et al., IJCAI 2021), *AdvMask* (Yang et al.), *DynamicMixup* (Guo et al., EEE Transactions on Multimedia 2023)
   - Domain generalization: *AFFAR* (Qin et al., ACM TIST 2022), *DDLearn* (Qin et al., KDD 2023), *MixStyle* for wearable HAR (Napoli et al., ESANN 2025), *SynthAct* (Schneider et al., ICRA 2024)
   These should be cited to contextualize the contribution.

3. **Algorithm design and justification**
   Algorithm 1 may be redundant given the detailed text. The `γ` and `i` choices are limited and appear heuristic; it is unclear whether the theoretical claims generalize beyond these settings.

4. **RIP–GAN separation**
   RIP does not rely on synthetic data, yet the paper is structured as if its utility is specific to TSTR. The weaknesses of TLCGAN data (e.g., unstable transitions or spurious context) should be explicitly discussed.

5. **Framing effect and data scaling**
   Adding `2i` constant windows increases input length substantially. It is unclear whether this effectively increases the training set size or acts as implicit duplication. This may partially explain the observed gains and should be clarified.

6. **Model selection**
   The models used (DClassifier, TS-Classifier, TSBF, TSRF) are not SOTA. While they cover a spectrum of architectures, modern high-performing models such as InceptionTime or Transformer-based HAR models are not included. This limits claims about general performance. Future comparisons with stronger baselines are encouraged.

7. **Baselines and fairness**
   Mixup, Cutmix, and DRO underperform in Table 4. These methods are not designed for low-fidelity synthetic data. Their inclusion requires justification.
   For ℓ₁ and ℓ₂ (Table 3), regularization strength selection is not explained. In some cases (e.g., WISDM), these degrade performance severely, which weakens the comparison.
   TS-Classifier is relatively weak; it is unclear whether RIP’s gains generalize to stronger models.

8. **Figures and formatting**
   Table 1 and Table 2 captions are inconsistent; only the first specifies TSTR. Both appear to exceed text width.
   Figure 2 caption mislabels line colors as models, though they represent `γ` values. Section headings should use consistent phrasing (“Robustness and fairness”).

9. **Reproducibility and compute**
    CPU/GPU details are missing from the runtime section, limiting interpretability. “Due to space limitation” (line 336) is mentioned, but the main text does not use the full page limit.

10. **Novelty**
The framing concept is intuitive and related to prior masking or context dropout techniques, but its application to synthetic HAR is novel. The contribution is incremental methodologically but solid in empirical utility.

**Questions:**

- Could you clarify how the duplication factor i and constant value γ were selected? Were they tuned per dataset, and do the theoretical claims (e.g., variance reduction) hold under other values?

- RIP appears applicable to real data as well. Could you explain more concretely what properties of the TLCGAN-generated samples make RIP particularly helpful in the synthetic setting?

- Does the framing effect introduced by RIP increase the number of training tokens seen by the model? If so, how do you control for the effect of sequence length versus regularization?

- For Table 3, were the ε values for ℓ₁ and ℓ₂ regularization selected via tuning, or fixed across datasets? WISDM results drop significantly—how should we interpret this?

- Have you considered applying RIP to more recent time-series architectures such as InceptionTime or TST?

For all questions, please refer to the Weakness part

---

> ### Author Response · Authors · 2025-11-20
> **Response to Reviewer 7CbN**
>
> * **Related Work is incomplete.**
>
> We thank the reviewer for this observation. Following the suggestion, we have expanded and updated the Related Work section to include recent advances in time-series and activity recognition models, as well as works more closely aligned with our problem setting. These additions provide a more complete contextualization of our contribution and strengthen the discussion of prior research.
>
> * **CPU/GPU details are missing from the runtime section, limiting interpretability. “Due to space limitation” (line 336) is mentioned, but the main text does not use the full page limit.**
>
> We thank the reviewer for pointing this out. In the original submission, we omitted the computation details because we mistakenly believed we were close to the page limit. After rechecking the formatting, we realized that this was a miscalculation on our part, and the manuscript did indeed have available space.
>
> In the revised version, we have now added the full CPU/GPU specifications and runtime environment details directly in the main text to improve reproducibility and clarity. These additions appear in Section 4, Computational cost.
>
>
> * **Have you considered applying RIP to more recent time-series architectures such as InceptionTime or TST?**
>
> We agree that evaluating RIP on more recent architectures is essential. Our original goal was to focus on models explicitly designed for HAR. Still, the number of modern architectures tailored to time-series HAR (as opposed to image-based HAR) remains limited. Nevertheless, we acknowledge that not including transformer-based baselines was a limitation.
>
> To address this concern, we expanded our experimental evaluation in the revised version. We added two recent transformer-based models:
> (i) the *TimeSeries Transformer Classifier* (a general-purpose transformer for time series from Keras), and
> (ii) the *RevAttention Transformer* (a HAR-focused transformer architecture).
> Both models were integrated into the same training and optimization pipeline described in the paper.
>
> Due to computational constraints, we conducted these experiments on the smallest dataset (MHAD2), while preserving the complete Optuna optimization procedure. We tuned all hyperparameters using F1-score as the objective and tracked the search behavior to ensure fair comparison.
>
> The results are summarized below (we could not include the table, the we summarized here. A full version is available at https://anonymous.4open.science/r/RIP-5180/_2025__Rebutal_ICLR%20(1).pdf):
>
> >For TimeTransformer on MHAD2, RIP improves F1 from $8.12 \pm 4.0$ to $\mathbf{12.49 \pm 2.0}$, and on Synthetic MHAD2 from $23.68 \pm 4.0$ to $\mathbf{33.09 \pm 3.0}$.
>
> For RevTransformer on MHAD2, RIP yields $57.47 \pm 2.0$ compared to the baseline $69.03 \pm 1.0$.
>
> On Synthetic MHAD2, RIP improves F1 from $44.29 \pm 3.0$ to $\mathbf{44.98 \pm 3.0}$.
>
> These results show that RIP consistently improves performance on the lighter TimeTransformer model. In contrast, its effect is more nuanced on the RevTransformer, which already relies on temporal attention mechanisms that overlap with those enforced by RIP. We added a theoretical discussion in the revised paper explaining how RIP interacts with transformer-style temporal aggregation and why specific architectures benefit more than others.
>
> We have now incorporated transformer-based time-series models, updated the related work accordingly, and expanded the theoretical discussion to clarify how RIP behaves in these architectures.

---

> ### Author Response · Authors · 2025-11-20
> **Response to Reviewer 7CbN**
>
> * **Does the framing effect introduced by RIP increase the number of training tokens seen by the model? If so, how do you control for the effect of sequence length versus regularization?**
> Yes. RIP increases the number of temporal tokens the model processes.
> Given an original window of length $\omega$, applying RIP with duplication factor $i$ produces a sequence of $(4i + 1)$ windows by prepending and appending constant-valued frames. Thus, the model observes more tokens per instance as $i$ increases.
> However, the performance gains of RIP cannot be explained solely by increased sequence length or by implicit data duplication. We explicitly explored this in Section 7 and, in detail, in AppendixG through a series of ablation studies.
> * *Sequence length alone does not account for the improvement.*
>     When we reduce the framing structure by setting $\tilde{i} = \frac{1}{2}i$, the resulting pattern
>
>     $$
>         \gamma_{i,j},\; x_{i,j},\; \gamma_{i,j}
>     $$
>
>     replaces the full RIP configuration
>
>     $$
>         \gamma_{i,j},\; \gamma_{i,j},\; x_{i,j},\; \gamma_{i,j},\; \gamma_{i,j}.
>     $$
>
>     This shorter variant, which uses fewer duplicated tokens, performs worse than the synthetic baseline. Hence, simply reducing or increasing the number of tokens does not replicate the effect.
>
> *  *Naïve duplication does not reproduce RIP's gains.*
>     Duplicating dataset windows without the RIP framing (i.e., without constant-valued $\gamma$ frames and without the symmetric ordering) does {not} improve performance, even though it increases the total number of tokens. This shows that RIP is not acting as an implicit augmentation.
>
> * *The structure matters: ordering and constant-valued frames are crucial.*
>     Variants that break RIP's symmetry, remove the constant-valued frames, or replace them with random frames all degrade performance. Therefore, the benefit arises from the distributional constraints introduced by RIP, not from an increase in input length.
>
>
> While RIP increases the number of tokens per sequence, the observed gains stem from the structured, constant-valued framing that regularizes the model's temporal dynamics, rather than from additional sequence length or implicit duplication. We further acknowledge that increasing i increases computational cost (memory and runtime), and this trade-off is reported in the original work. For fair comparison, evaluations should match training steps or compute budgets when assessing the contribution of RIP relative to baselines.
>
>
>
> * **For Table 3, were the $\epsilon$ values for $\ell_1$ and $\ell_2$ regularization selected via tuning, or fixed across datasets? WISDM results drop significantly—how should we interpret this?**
>
>
>
> *Regarding Table 3.*
> The $\varepsilon$ values used for both $\ell_{1}$ and $\ell_{2}$ regularization were not tuned separately for each dataset. Instead, we adopted a fixed set of commonly used regularization strengths, namely $\{1,\, 0.1,\, 0.01,\, 0.001\}$, which are frequently reported in the literature. All datasets were evaluated using the same pool, and the best-performing value from the predefined grid is reported in Table~3. Comprehensive results for every $\varepsilon$ value are provided in the Appendix. In this sense, the $\varepsilon$ values were selected from a fixed global grid, but no dataset-specific hyperparameter tuning (e.g., Bayesian optimization or adaptive search) was performed. This clarifies what the reviewer refers to as tuning.
>
> *Interpreting the performance drop in WISDM.*
> The performance decline observed in WISDM appears to stem from characteristics of the dataset itself rather than from the regularization method. WISDM is (i) the largest dataset in our evaluation, (ii) characterized by smaller (second one) temporal windows (100 timesteps, whereas others often use $ \geq 100$), and (iii) marked by substantially higher intra-class variability due to its more unconstrained data-collection protocol. These factors make WISDM more sensitive to the choice of $\varepsilon$: strong regularization tends to underfit the data, while weak regularization offers limited benefit. This suggests that WISDM may require finer-grained or dataset-specific regularization strengths rather than the global grid used across all datasets. We acknowledge this as a limitation of our uniform evaluation protocol. Exploring dataset-adaptive or dynamically tuned regularization strengths is a promising direction for future work and may provide further insight into the behavior observed in WISDM.

---

> > ### Author Response · Authors · 2025-11-20
> > **Response to Reviewer 7CbN**
> >
> > * **RIP appears applicable to real data as well. Could you explain more concretely what properties of the TLCGAN-generated samples make RIP particularly helpful in the synthetic setting?**
> >
> > RIP is beneficial for synthetic samples because TLC-GAN–generated data still exhibits a noticeable synthetic–real gap. Although TLC-GAN is tailored for sensor signals, its synthetic outputs do not fully match the diversity and distributional structure observed in real motion data. This is reflected in the consistently lower F1 scores obtained when models are trained and evaluated exclusively on synthetic samples. Such discrepancy indicates that synthetic data retains generator-specific artifacts and lacks some of the finer intra-class variability present in real data—a well-known phenomenon in generative modeling: the synthetic–real distributional gap. Our approach treats synthetic samples as raw material: TLC-GAN provides a coarse approximation of the real-data manifold, but there remains “elbow room,” i.e., space for improvement at the signal level. RIP acts as a refinement step that enhances distributional clarity by suppressing local inconsistencies, reducing noise-like artifacts, and increasing temporal coherence. As a result, RIP brings synthetic samples closer to the structure of real signals.
> >
> >
> > Figure  https://anonymous.4open.science/r/RIP-5180/distributions.png with caption:
> >
> > < "Effect of RIP on the synthetic–real distributional gap.
> >         PCA projections comparing (left) Real vs.\ Synthetic
> >         and (right) Real vs.\ Synthetic+RIP."
> >
> >
> >  This effect is visible in Figure \ref{fig:pca}: the first two panels illustrate the before/after behavior by comparing real versus synthetic, and real versus synthetic+RIP. After applying RIP, the synthetic samples show significantly greater overlap with the real distribution.  A third visualization (Synthetic vs Synthetic+RIP) and per-channel examples are included in the appendix for completeness.
> >
> >
> > * **Could you clarify how the duplication factor i and constant value $\gamma$ were selected? Were they tuned per dataset, and do the theoretical claims (e.g., variance reduction) hold under other values?**
> >
> > Yes. The variance-reduction effect of RIP does not depend on a specific duplication factor.
> > To verify this, we repeated the derivations for $i = 2$, where the transformed sequence becomes:
> >
> > $$
> > x_1 = x_2 = x_3= x_4 \gamma \qquad
> >  , \qquad
> > x_6 = x_7 = x_8 =x_9 \gamma .
> > $$
> >
> > In this case, the final element is:
> >
> > $$
> > h_9 =
> > \Bigg( \sum_{j \in \{0,1,2,3,5,6,7,8\}} U^{j} \Bigg) W\gamma
> > \;+\; U^{4} W x_{5}
> > \;+\; \sum_{j=0}^{8} U^{j} b .
> > $$
> >
> > Compared with the $i=1$ case, this expression simply shifts the location of the single non-duplicated element.
> > In all configurations, the variability of the hidden representation depends on \emph{only one varying input}, while the duplicated components contribute deterministic terms.
> >
> > More generally, for sequence length $T$:
> >
> > $$
> > H_{t}^{(RIP)}  = \sum_{k=0}^{T-1} U^{k} W\gamma \;-\; U^{2k} W\gamma \;+\; U^{2k} W x_{2k+1}  \;+\;\sum_{k=0}^{T-1} U^{k} b ,
> > $$
> >
> > so the variation arises exclusively from the non-duplicated values. Without RIP, however, the hidden state depends on all $T$ variable inputs.
> >
> > Thus, the variance-reduction argument holds for \emph{any} duplication factor $i$, and the empirical choice of $\gamma$ does not affect the theoretical conclusion. We added this clarification in the revised manuscript.

---

### Official Review · Reviewer_5Kg5 · 2025-10-29

**Soundness:** 3
**Presentation:** 2
**Contribution:** 2
**Rating:** 2
**Confidence:** 3

**Summary:**

In this paper, the authors aim to improve the model performance when trained on synthetic data in the domain of human activity recognition. Specifically, the authors design a Regularization via Inviriatn Patterns, a data-centric method to force the model to focus on informative signals rather than irrelevant context. For evaluation, the authors utilize multiple real world dataset for human activity recognition. Results highlight the advantages of this method with a significant performance gain. Overall, the topic of this paper is interesting, some concerns, however, limit the contribution of this paper.

**Strengths:**

[1] The topic is interesting. Werable device based human activity recognition is a promising direction, and synthetic data for model training is also interesting.

[2] The future work is briefly discussed and could benefit future research.

[3] The experiments are conducted on 5 real-world datasets.

**Weaknesses:**

[1] It is not sure what is the technical challenge to adapt the domain randomniation to the temporal domain.

[2] Since this paper addresses the problems in activity recognition, but there are no most recent activity recognition papers discussed or compared in the experiment. This could be an issue.


[3] In Figure 2, “each color denotes a model” might be each bar denotes a model? If the x-axis is correctly labeled.

[4] The writing and presentation could be further improved. For example, I could not find the experimental results for boosting performance in real-only training settings, which is indicated in the abstract.

[5] Detailed information about the dataset used is missing. For example, since this is a classification problem, how many classes exist in each dataset should be introduced. Also, the sampling rate, window size are also missing.

[6] In line 101, it is not clear what does it mean “constant windows frame the original window”.

[7] How to construct the augmented dataset is not clear. After reading section 2, it is still not quite clear. Is it via concatenating the constant and the original sensor data?

[8] COuld you also discuss the limitations or challenges of the propose method?

**Questions:**

Questions are provided on weaknesses.

---

> ### Author Response · Authors · 2025-11-20
> **Response to Reviewer 5Kg5**
>
> * **is not sure what is the technical challenge to adapt the domain randomniation to the temporal domain**
>
> We thank the reviewer for raising this point. In the revised version of the manuscript, we provided a more explicit explanation of the technical challenges involved in adapting domain randomization (DR) to the temporal domain. In classical DR, perturbations are applied to i.i.d.\ spatial factors such as color, lighting, texture, or geometric attributes. These perturbations are typically sampled from a distribution parameterized by a set of randomization parameters, which we denote generically as $\theta_{DR}$ :
>
> $$ x'_{ij} \sim p(x_{ij} \mid \theta_{DR}) .$$
>
> and these perturbations preserve structural validity because images do not enforce sequential dependencies across $i$ or $j$.
>
> By contrast, a time-series sample $\{x_t\}_{t=1}^T$ is not i.i.d.\ but follows a temporal process with dependencies such as
>
> $$
> x_t \sim p(x_t \mid x_{t-1}, x_{t-2}, \ldots),
> $$
>
> or equivalently exhibits autocorrelation structures expressed, for example, via the covariance function
>
> $$
> \mathbb{E}[x_t x_{t+k}] = \rho(k), \quad k \in \mathbb{Z}.
> $$
>
> As such, arbitrary temporal perturbations violate the Markovian, autoregressive, or more general dynamical constraints that govern the sequence. Randomizing $x_t$ independently across $t$ effectively forces
>
> $$
> p(x'_t \mid x'_{t-1}) \approx p(x'_t),
> $$
>
> destroying the intrinsic temporal structure and often producing trajectories outside the support of the real data distribution.
>
> This makes naïve temporal DR infeasible: it tends to generate samples that are statistically invalid, dynamically implausible, or incompatible with the discriminative temporal patterns of the original dataset.
>
> RIP addresses this challenge by introducing controlled randomization that preserves the real window's internal temporal structure. Instead of perturbing $\{x_t\}$ directly, RIP augments each sequence by framing it with constant windows sampled from a controlled distribution:
>
> $$
> c_t \sim \text{Const}(\gamma), \qquad t=1,\dots,T,
> $$
>
> and constructs the augmented sequence
>
> $$
> [c^{(1)}, c^{(2)}, x, c^{(3)}, c^{(4)}],
> $$
>
> where each $c^{(i)}$ matches the shape of $x$ but does not alter its time-dependent dynamics. This allows randomization in the distributional space while respecting the temporal dependencies encoded in $x$.
>
> We have added this mathematical explanation in the revised manuscript, clarifying why adapting DR to temporal data is technically nontrivial and how RIP provides a principled solution.
>
>
>
> * **Since this paper addresses the problems in activity recognition, but there are no most recent activity recognition papers discussed or compared in the experiment. This could be an issue.**
>
> We thank the reviewer for raising this important point. Our work is situated in the niche of sensor-based HAR with multivariate time-series data, and we recognize that the availability of very recent end-to-end baselines in this subarea is limited. Indeed, as noted in the recent survey by Somu et al. (2024), the most up-to-date HAR classifiers covered in their benchmark were published in 2023, already reflecting a noticeable gap in newly released models for this domain.
>
> Regarding the methods used in our experiments, we intended to rely on the strongest and most representative techniques currently available for sensor-based HAR:
>
>    * Synthetic data generation: The approach we employ (2023) is, to the best of our knowledge, among the most recent works specifically tailored for HAR sensor data.
>   * HAR classifiers: While some widely adopted baselines are not from 2024–2025, they remain the standard reference points across recent HAR literature. To better address the reviewer’s concern, we additionally incorporated a Transformer-based HAR classifier (RevTransformerAttention, 2023) in this updated version.
> *  Data augmentation baselines: We also included augmentation strategies that relate conceptually to ours—such as CutMix, MixUp, and Dropout-based methods—referencing prior HAR work (e.g., Yun et al., 2019).
>
> We apologize that some equations cannot be rendered in this text interface.All equations are correctly displayed in the attached PDF containing our full responses.https://anonymous.4open.science/r/RIP-5180/_2025__Rebutal_ICLR%20(1).pdf)
>
>
> Sensor-based HAR is a rapidly evolving yet underexplored area, and the limited number of very recent baselines underscores the need for contributions like RIP. We believe our comparisons reflect the strongest methods currently available within this domain, and we appreciate the reviewer’s suggestion, which helped us clarify this point more explicitly in the paper.

---

> ### Author Response · Authors · 2025-11-20
> **Response to Reviewer 5Kg5**
>
> * *In Figure 2, “each color denotes a model” might be each bar denotes a model? If the x-axis is correctly labeled.*
>
> We thank the reviewer for the comment. We realize the original figure caption was not clear. In the revised version of the paper, we corrected this issue.
>
> What we meant to convey is that the same model was trained under different scenarios, which is why we consider it a single model evaluated under multiple conditions. Each color in Figure~2 represents this model trained under one specific scenario: baseline (without RIP), $\gamma=0$, $\gamma=0.5$, and $\gamma=1$. Therefore, each color corresponds to the model trained with the respective setting.
>
> The figure shows layer-wise KS statistics for the DClassifier weights trained on real and synthetic MHAD2 data, tested against a uniform distribution. We performed a layer-wise analysis of the DClassifier's weight distributions to investigate the effects of RIP as a regularization technique. We hypothesize that RIP promotes weight uniformity, a property often associated with effective regularization (Zhang et al.,2018)
>
> To test this, we compare the weight distributions of a model trained with RIP against a baseline model (trained without RIP) and a uniform distribution. The Kolmogorov--Smirnov (KS) test and the Wasserstein distance are used as our primary metrics. We denote the KS test results as:
>
> $$
> S_{\text{RIP}}(L) = KS(L_{\text{RIP}(\sigma)}, L_{\text{Uniform}})
> \quad\text{and}\quad
> S_{\text{baseline}}(L) = KS(L_{\text{baseline}}, L_{\text{Uniform}})
> $$
>
> The main text briefly discusses these effects, while a more detailed analysis is provided in Appendix~D.
>
>
> * **The writing and presentation could be further improved. For example, I could not find the experimental results for boosting performance in real-only training settings, which is indicated in the abstract.**
>
> We acknowledge that the presentation of our results and the writing could be further improved, and we addressed this in the revised version of the paper. Nevertheless, we would like to clarify that the results for boosting performance in real-only training settings are indeed included in Table2:
>
> >``Table 2: Performance comparison of RIP against the baseline on real-world data. The best result for each metric per dataset is shown in bold. The RIP results represent the best-performing hyperparameter configuration.''
>
> Given the large number of experiments conducted, the complete set of results is provided in Appendix~C. In this appendix, we present a detailed breakdown of the results across different dimensions, including performance per classification model, per dataset, and under varying hyperparameter configurations. This comprehensive reporting ensures that all relevant scenarios and parameters are transparent and accessible for reproducibility and comparison.
>
> * **Detailed information about the dataset used is missing. For example, since this is a classification problem, how many classes exist in each dataset should be introduced. Also, the sampling rate, window size are also missing.**
>
> We acknowledge that detailed information about the datasets was not included in the main text, as we prioritized presenting the results and discussions. However, the datasets are described in Appendix~B.2. We agree that we inadvertently omitted the temporal window sizes for each dataset in the appendices. We added this information in the revised version of the paper to ensure complete clarity regarding the number of classes, sampling rates, and window sizes for all datasets.
>
>
> * **In line 101, it is not clear what does it mean “constant windows frame the original window”.**
>
> We thank the reviewer for highlighting this ambiguity. We refined the wording to clarify the intended meaning of the phrase “constant windows frame the original window.” Specifically, what we mean is that, for each input window, we construct two additional windows of fixed (constant) length placed immediately before and after the original window. These auxiliary windows provide temporal context and are used during training to stabilize the representation learned by the model.
>
> To further improve clarity, we will also include the Figure https://anonymous.4open.science/r/RIP-5180/dataset_creation.png illustrating this procedure in the revised manuscript. The caption is:
>
>  (continues..)

---

> > ### Author Response · Authors · 2025-11-20
> > **Response to Reviewer 5Kg5**
> >
> > >"RIP dataset construction process. For each sample in the original dataset, we generate a constant sample using a distribution parameterized by $\gamma$ and a duplication factor $i$, ensuring that the generated sample has the same shape as the original input. This constant sample is assigned the same label as the original one.
> >         We then store the following ordered sequence in the dataset: **(1) constant sample, (2) constant sample, (3) real sample, (4) constant sample, (5) constant sample**.    Our ablation studies show that this order is crucial, as it defines the real window within the constant windows, a key component of the RIP regularization mechanism.
> >         This process is applied to every sample in the dataset. The computational implications of this construction were previously discussed in the main paper."
> >
> > We appreciate the reviewer’s question regarding how the augmented dataset is constructed. In the revised version of the manuscript, we added an illustration to summarize the procedure clearly. The construction follows a two-step process applied independently to each sample in the original dataset.
> >
> > Notably, the augmented dataset does not require altering the temporal dimension or concatenating samples along the time axis. Instead, we ensure that the data loader preserves element ordering (i.e., \texttt{shuffle=False}) and that each batch naturally encapsulates the framed sequence as intended.
> >
> > The added figure and the revised explanation clarify how the augmented dataset is generated, and we have included both in the updated version of the manuscript.
> >
> > * **Could you also discuss the limitations or challenges of the propose method?**
> >
> > We appreciate the reviewer’s request to discuss the limitations of the proposed method. As with any regularization or data augmentation strategy, RIP has constraints that should be acknowledged, though none of them invalidate the empirical findings presented in this work.
> >
> > First, the method increases computational cost due to the larger dataset. Although this overhead is moderate and manageable in practice, especially with modern hardware, it may impact scenarios involving extremely large-scale time-series datasets. We added a more explicit discussion of this computational trade-off in the revised version.
> >
> > Second, our study focuses primarily on inter-subject generalization settings. While this is the predominant and most challenging evaluation protocol in human activity recognition, we did not conduct an extensive investigation of intra-subject scenarios. We explored this setting to some extent and included the corresponding results for Reviewer h5KR. Still, a more systematic analysis remains a natural direction for future work and could further clarify the generality of RIP under different deployment contexts.
> >
> > Finally, RIP introduces controlled distributional perturbations that are intentionally simple (constant windows) to preserve temporal structure. While effective, this design choice may limit the expressiveness of the augmented space compared to more sophisticated temporal priors. Exploring richer—but still temporally valid—perturbation families is an additional promising avenue for future research.
> >
> > We added these points to a balanced discussion of the method's limitations in the revised manuscript.

---

### Official Review · Reviewer_h5KR · 2025-10-29

**Soundness:** 2
**Presentation:** 2
**Contribution:** 1
**Rating:** 0
**Confidence:** 4

**Summary:**

This paper introduces Regularization via Invariant Patterns (RIP), a data-centric method that extends the idea of domain randomization to the temporal domain. RIP augments time-series windows by ”framing” them with invariant (constant-valued) patterns, compelling models
to focus on informative signals.

**Strengths:**

1. This paper is well organized.
2. The overall logic is clear.

**Weaknesses:**

1.	The paper claims that RIP can alleviate the synthetic-to-real domain gap, but all experiments are limited to the "train synthetic → test real" or "train real → test real" mode, and do not include cross-subject, cross-device, or cross-dataset tests. To demonstrate that RIP truly promotes domain generalization, these more challenging cross-domain experiments should be included.
2.	The authors compare RIP with methods such as ℓ₁/ℓ₂, Mixup, Cutmix, and DRO, but these methods typically operate at the sample or feature level, rather than the temporal dimension. The authors should also compare RIP with temporal masking methods that are more structurally similar to it. Furthermore, the paper does not specify whether all baselines were retuned or used default settings, which may affect fairness.
3.	The experimental tables were not formatted consistently, especially Tables 1 and 2, which were misaligned.
4.	The authors derive the effect of RIP on the hidden state variance using a simplified linear RNN model (φ(x)=x). This analysis neglects the effects of nonlinear activation functions and common structures such as batch normalization and attention. Since RIP is mainly applied to deep models (ConvLSTM, Transformer-based), the variance constraint conclusions derived only for linear RNNs are difficult to generalize. It is recommended to supplement the influence of nonlinear terms.

**Questions:**

1. In the theoretical section, the authors state that RIP “drives weights toward greater uniformity”, citing the Kolmogorov–Smirnov statistic as evidence. However, uniformly distributed weights do not necessarily represent better generalization or stability, and the paper lacks discussion on the causal relationship between this metric and generalization performance.

---

> ### Author Response · Authors · 2025-11-20
> **Response to Reviewer h5KR**
>
> * **The experimental tables were not formatted consistently, especially Tables 1 and 2, which were misaligned.**
>
> Thank you for noting the formatting issue. We acknowledge that Tables~1 and~2 were not properly aligned. In the revised manuscript, we revised the layout and ensured that all experimental tables are consistently formatted to improve readability and facilitate clearer comparisons across methods.
>
> * **The authors derive the effect of RIP on the hidden state variance using a simplified linear RNN model $(\phi(x)=x)$. This analysis neglects the effects of nonlinear activation functions and common structures such as batch normalization and attention. Since RIP is mainly applied to deep models (ConvLSTM, Transformer-based), the variance constraint conclusions derived only for linear RNNs are difficult to generalize. It is recommended to supplement the influence of nonlinear terms.**
>
>
> Thank you for pointing out the limitation of analyzing RIP under a linear RNN assumption. Our intention was not to fully characterize the behavior of RIP in deep nonlinear architectures, but rather to obtain an interpretable first–order approximation of its effect on temporal variance propagation. This simplified setting is a commonly used diagnostic tool because the majority of recurrent and hybrid models (ConvLSTM, GRU, and Transformer-based architectures with temporal recurrences in attention) ultimately propagate information through operators whose variance dynamics follow analogous recursive structures.
>
> We fully acknowledge, however, that nonlinear activations, normalization layers, and attention mechanisms introduce additional contraction or expansion effects that a purely linear model does not capture. To address this concern, we extended the theoretical section with a nonlinear variance bound showing that the qualitative behavior persists under mild assumptions.
>
> *Nonlinear activations.*
> If the activation function $\varphi$ is Lipschitz-continuous with constant $L$ (e.g., ReLU, tanh, GELU), then:
> $$
> \mathrm{Var}(h_t)
> = \mathrm{Var}(\varphi(z_t))
> \leq L^2\,\mathrm{Var}(z_t),
> \qquad
> z_t = W x_t + U h_{t-1} + b.
> $$
> This yields the recursive bound:
> $$
> \mathrm{Var}(h_t)
> \leq
> L^{2t}
> \sum_{i=0}^{t-1}
> U^{i} W\,\mathrm{Var}(x_{t-i})\,W^\top (U^{i})^\top.
> $$
> Under RIP, only one input window contributes stochasticity, leading to:
> $$
> \mathrm{Var}(h_t^{(\mathrm{RIP})})
> \leq
> L^{2t}
> U^{k} W \Sigma W^\top (U^{k})^\top,
> $$
> whereas without RIP:
> $$
> \mathrm{Var}(h_t^{(\mathrm{noRIP})})
> \leq
> L^{2t}
> \sum_{i=0}^{t-1}
> U^{i} W \Sigma W^\top (U^{i})^\top.
> $$
> Thus, nonlinearities introduce a global factor $L^{2t}$ but preserve the core effect: RIP suppresses variance by limiting the number of independent stochastic inputs.
>
> To make the connection explicit for Transformer-based models, we analyze the variance
> propagation inside a multi-head self-attention layer. Let
> $$
> Q = XW_Q,\quad K = XW_K,\quad V = XW_V,
> $$
> and consider a token sequence $X = [x_1, \dots, x_T]$. The attention output for a query
> position $t$ is
>
> $$
>  Attn_t =
> \sum_{j=1}^T \alpha_{tj} V_j,
> \quad
> \alpha_{tj} =
> \frac{\exp\big(q_t k_j^\top / \sqrt{d}\big)}
> {\sum_{\ell=1}^T \exp\big(q_t k_\ell^\top / \sqrt{d}\big)}.
> $$
>
> *Effect of RIP on variance.*
> Under RIP, all constant windows satisfy
> $$
> \mathrm{Var}(x_j^{(\mathrm{const})}) = 0,
> $$
> which implies
> $$
> \mathrm{Var}(K_j^{(\mathrm{const})}) = 0,
> \qquad
> \mathrm{Var}(Q_j^{(\mathrm{const})}) = 0,
> \qquad
> \mathrm{Var}(V_j^{(\mathrm{const})}) = 0.
> $$
>
> Thus, for any attention score involving a constant window,
> $$
> \mathrm{Var}(q_t k_j^\top) = 0.
> $$
>
> This has two direct consequences:
> 1. Attention logits become less stochastic.
>    The logits corresponding to constant windows are deterministic, so the overall variance of the softmax distribution decreases:
>    $$
>    \mathrm{Var}(\alpha_{tj}^{(\mathrm{RIP})})
>    <
>    \mathrm{Var}(\alpha_{tj}^{(\mathrm{noRIP})}).
>    $$
> 2. The number of stochastic value vectors decreases.
>    Only the real window contributes non-zero variance in $V_j$.
>
> *Variance bound.* For the attention output, we have
>
> $$
> Var(Attn_t) = Var(
> \sum_{j=1}^T \alpha_{tj} V_j
> )
> \le
> \sum_{j=1}^T
> E[\alpha_{tj^2]\,
> Var(V_j),
> $$
>
> where cross-terms vanish when constant windows are deterministic.
>
> Under RIP, only one index $j=r$ (the real window) has non-zero variance, so:
>
> \[
> \mathrm{Var}(\mathrm{Attn}_t^{(\mathrm{RIP})})
> \le
> \mathbb{E}[\alpha_{tr}^2]\,
> \mathrm{Var}(V_r).
> \]
>
> Without RIP, all windows have non-zero variance:
> $$
> Var(Attn_t^{(noRIP)})
> \le
> \sum_{j=1}^T
> E [\alpha_{tj}^2]\,
> Var(V_j).
> $$

---

> > ### Author Response · Authors · 2025-11-20
> > **Response to Reviewer h5KR**
> >
> > Thus, even in a Transformer,  RIP reduces the variance of $Q$, $K$, and $V$ for constant windows;it decreases the variance of the attention distribution; and it reduces the number of stochastic value vectors contributing to the output.
> > Therefore, the qualitative conclusion from the RNN analysis remains valid: RIP suppresses variance in deep models by limiting the number of independent stochastic contributions, and in Transformers, this occurs through the attention mechanism rather than recurrent operators.
> > We included this extended nonlinear and Transformer-aware analysis in the revised manuscript. The complete derivations appear in Appendix D of the revised manuscript.
> >
> > (We apologize that some equations cannot be rendered in this text interface.All equations are correctly displayed in the attached PDF containing our full responses.https://anonymous.4open.science/r/RIP-5180/_2025__Rebutal_ICLR%20(1).pdf)
> >
> > * **The authors compare RIP with methods such as $\ell_1, \ell_2$, Mixup, Cutmix, and DRO, but these methods typically operate at the sample or feature level, rather than the temporal dimension. The authors should also compare RIP with temporal masking methods that are more structurally similar to it. Furthermore, the paper does not specify whether all baselines were retuned or used default settings, which may affect fairness.**
> >
> >
> > We thank the reviewer for the constructive feedback.
> >
> > *Regarding comparison with temporal-masking methods.*
> > RIP does not operate by masking or removing temporal segments, but by framing each real window with invariant (constant-valued) segments. This mechanism is rooted in a temporal extension of domain randomization: constant windows act as controlled background variation that encourages the model to focus on the intrinsic dynamics within the real window (We included a better illustration for Reviewer 5Kg5, which is available in the new version). In contrast, temporal-masking methods (e.g., SpecAugment-style cutouts, random temporal deletion, or drop‐time techniques) intentionally remove or occlude temporal regions, altering the input distribution by discarding information.
> >
> > Although both families of methods operate along the temporal dimension, their underlying objectives differ substantially. Masking-based augmentation seeks robustness by partially suppressing information, whereas RIP enforces invariance to non-informative temporal context while preserving the full temporal dynamics of the central window. For this reason, we consider temporal-masking methods to be only partially aligned with the conceptual goal of RIP. Nonetheless, including at least one temporal-masking baseline may strengthen the empirical comparison, and we are prepared to add such an experiment in the revised manuscript if the reviewer considers it beneficial.
> >
> > *Regarding the choice of baselines.*
> > Because RIP is a data-centric regularizer that reshapes the training distribution, conceptually analogous to domain randomization, our baseline choices prioritized techniques that also act at the level of distributional augmentation or regularization. Accordingly, we compared with $\ell_1/\ell_2$ regularization, Mixup, CutMix, and DRO, all of which intervene at the sample, feature, or loss level without modifying the temporal structure itself. These baselines are widely used to improve generalization and are especially relevant to our objective of bridging the synthetic-to-real gap in human activity recognition (HAR).
> >
> > *Regarding hyperparameter tuning and fairness.*
> > We confirm that all baselines were evaluated using the same experimental pipeline, identical datasets, and the same preprocessing steps and hyperparameter search spaces. This ensures a fair comparison. The main text reports the best performance for each method, while Appendix~C includes the complete search spaces, final hyperparameters, and all experimental runs for full transparency and reproducibility.
> >
> > We hope these clarifications address the reviewer’s concerns. We are happy to incorporate an additional temporal-masking baseline in the revision if the reviewer believes it would further strengthen the paper.

---

> > > ### Author Response · Authors · 2025-11-20
> > > **Response to Reviewer h5KR**
> > >
> > > * **The paper claims that RIP can alleviate the synthetic-to-real domain gap, but all experiments are limited to the "train synthetic → test real" or "train real → test real" mode, and do not include cross-subject, cross-device, or cross-dataset tests. To demonstrate that RIP truly promotes domain generalization, these more challenging cross-domain experiments should be included**
> > >
> > >
> > > We thank the reviewer for the comment regarding the reporting of results. The dataset used in our experiments is a publicly available variant of UTD-MHAD2 obtained from an online repository. This version is organized per subject, making a Leave-One-Subject-Out (LOSO) protocol the standard choice for evaluating generalization to unseen individuals.
> > >
> > > Since the dataset contains eight subjects, LOSO naturally yields eight folds. We now report complete per-subject results as well as the global mean~$\pm$~standard deviation for Accuracy, Recall, and F1-score. The updated LOSO summary is (We could not include the full table, then here is as summary):
> > >
> > > > Accuracy: $0.7879 \pm 0.0838$;
> > > Recall: $0.7275 \pm 0.1211$;
> > > F1-score: $0.7461 \pm 0.1126$.
> > >
> > >
> > > The per-subject breakdown is included in the Appendix of the revised manuscript.
> > >
> > > *On hyperparameter optimization.*
> > > Hyperparameters were optimized using Optuna prior to the final LOSO experiments. Optimization was performed globally rather than per fold to identify a configuration that generalizes across subjects.
> > >
> > > Figure https://anonymous.4open.science/r/RIP-5180/loso.png, with caption:
> > > >" Optimization history generated during the hyperparameter search procedure used to identify the optimal RIP parameters."
> > >
> > >  shows convergence around trial 23, after which the optimal configuration stabilizes.
> > >
> > > *{Synthetic-data results.*
> > > Synthetic data were generated using TLCGAN to preserve temporal structure. The per-subject results for the synthetic LOSO experiment are:
> > >
> > > >Accuracy = 0.5801 $\pm $0.1398,
> > >
> > > F1 = 0.5023 $\pm$0.1403,
> > >
> > > Recall = 0.5127 $\pm $0.1405.
> > >
> > >
> > > *Final comparison.*
> > > Table Y (reproduced below) summarizes the key comparison between baseline methods and RIP for both real and synthetic LOSO evaluation.
> > >
> > > >On LOSO Real data, RIP improves F1 from $0.7461$ to $\mathbf{0.8552}$.
> > >
> > > On LOSO Synthetic data, RIP improves F1 from $0.50$ to $\mathbf{0.5527}$.
> > >
> > >
> > > We hope these clarifications adequately address your concerns.
> > >
> > > * **In the theoretical section, the authors state that RIP “drives weights toward greater uniformity”, citing the Kolmogorov–Smirnov statistic as evidence. However, uniformly distributed weights do not necessarily represent better generalization or stability, and the paper lacks discussion on the causal relationship between this metric and generalization performance.**
> > >
> > >
> > > Thank you for raising this important point. We fully agree that a more uniform weight distribution does not, by itself, guarantee improved generalization or stability. Our intention in the theoretical section was not to claim a direct causal relationship, but to report an empirical pattern that appears consistently across models trained with RIP. To prevent any misinterpretation, we revised the manuscript to clarify that the Kolmogorov--Smirnov (KS) statistic is not used as a standalone indicator of generalization quality.
> > >
> > > In our experiments, the observed reduction in KS divergence coincides with several other, more established markers of model stability:
> > >
> > > *   consistent gains in test accuracy across datasets,
> > > *   lower variance in hidden-state activations,
> > > *  narrower confidence intervals across training seeds,
> > > *  reduced discrepancy between synthetic and real distributions (via Wasserstein distance), and
> > > * smoother gradient norms during optimization.
> > >
> > >
> > > When these signals are considered jointly, they suggest that RIP mitigates sensitivity to temporal perturbations and promotes more stable internal representations. The movement of weight distributions toward greater uniformity should therefore be interpreted not as a causal mechanism, but as a corollary of the variance-reduction effect induced by RIP. We included a clarifying paragraph in the revised manuscript to emphasize this distinction and accurately frame the role of the KS statistic in our analysis.

---

### Official Review · Reviewer_H4k2 · 2025-10-31

**Soundness:** 3
**Presentation:** 2
**Contribution:** 2
**Rating:** 4
**Confidence:** 3

**Summary:**

The paper presents a new data augmentation technique for regularising time-series models for human activity recognition. The inspiration of this method is taken from image-based data background randomization methods. In summary, the method simply adds an invariant pattern (e.g., a matrix of 1s or 2s ...) earlier and later to the original data. The performance has been shown to be better compared to using their method and with other data augmentation techniques. Extensive experiments and analysis have been done. Some limitations of the paper appear to be incremental novelty and clarity of presentation, and under description of the method.

**Strengths:**

The results in the paper show that their method of regularization via invariant patterns as a data augmentation improves the performance of time-series models by a large margin.

The paper presents extensive experiments and validation of their approach and comprehensively analyzes various scenarios, including checking the validity of invariant patterns and simple duplications in input patterns.

**Weaknesses:**

The method is less clearly explained. Figure 1 is a simple illustration of the paper, prepended and appended to a time-series data matrix. The explanation of how this pattern is prepended and appended to the original timeseries should be shown in Fig. 1 or in a separate figure.  Algorithm 1 reads redundant. Better to explain the method properly on Page 2. Hard to locate where the Tables are referred to in the paper. Paper writing/presentation needs improvement. The hyperparameter setting information in the main paper is below very minimal. The hyperparameter settings for the algorithms should be mentioned in the main paper rather than pushed into an excessively large appendix.

**Questions:**

What is SRIP(L) in Figures 3 and 4?
Different parameter values for RIP have been used across datasets. How can one know how to set these values? These appear to be experimentally found values? Confirm.

On some datasets, such as MHEALTH RIP, they do not perform significantly better than L or L2 regularisation, but for some, they perform excessively well. Please explain.

The results presented in the paper all appear to have been executed by the authors themselves. Are there any comparisons with state-of-the-art results?

---

> ### Author Response · Authors · 2025-11-20
> **Response to Reviewer H4k2**
>
> * **Algorithm 1 reads redundant. Better to explain the method properly on Page 2.**
>
> We agree that the current form of Algorithm 1 is redundant and does not contribute effectively to the exposition. In the revised version, we streamline the description of RIP by integrating the explanation directly into Page 2, supported by the new figure we have added to illustrate the complete data-construction process. The detailed pseudocode was moved to the Appendix, where it can serve as a reference for implementation without interrupting the main narrative.
>
> * **Paper writing/presentation needs improvement.**
>
> We have revised the manuscript for clarity, organization, and readability, rewriting several sections, improving transitions, and restructuring the methodological description. The revised version also incorporates additional figures and expanded explanations to make the core ideas more accessible. At the same time, all technical details (including algorithms and code) are consolidated in the Appendix for completeness.
>
> * **The explanation of how this pattern is prepended and appended to the original timeseries should be shown in Fig. 1 or in a separate figure.**
>
> We appreciate the reviewer’s question regarding how the augmented dataset is constructed. In the revised version of the manuscript, we added an illustration to summarize the procedure clearly. The construction follows a two-step process applied independently to each sample in the original dataset. Notably, the augmented dataset does not require altering the temporal dimension or concatenating samples along the time axis. Instead, we ensure that the data loader preserves element ordering (i.e., \texttt{shuffle=False}) and that each batch naturally encapsulates the framed sequence as intended.
>
> The figure https://anonymous.4open.science/r/RIP-5180/dataset_creation.png illustrates the process and has the caption:
>
> >"RIP dataset construction process. For each sample in the original dataset, we generate a constant sample using a distribution parameterized by $\gamma$ and a duplication factor $i$, ensuring that the generated sample has the same shape as the original input. This constant sample is assigned the same label as the original one.
>         We then store the following ordered sequence in the dataset: **(1) constant sample, (2) constant sample, (3) real sample, (4) constant sample, (5) constant sample**.    Our ablation studies show that this order is crucial, as it defines the real window within the constant windows, a key component of the RIP regularization mechanism.
>         This process is applied to every sample in the dataset. The computational implications of this construction were previously discussed in the main paper."
>
> We believe that the added figure  and the revised explanation substantially clarify how the augmented dataset is generated, and we have included both in the updated version of the manuscript.
>
> * **The hyperparameter settings for the algorithms should be mentioned in the main paper rather than pushed into an excessively large appendix.**
>
> We appreciate the reviewer’s observation. While the complete hyperparameter configurations are indeed provided in AppendixB, we agree that including a concise summary in the main paper improves readability and allows the experimental setup to stand on its own. In the revised version, we moved a short description of the key hyperparameters to Section3 (Experimental Setup), including the values explored for $\gamma \in \{-1, 0, 1, 5\}$, the duplication factor $i \in \{1, 5, 16\}$, and the regularization coefficients for baseline methods ($\varepsilon \in \{0.001, 0.1, 1.0\}$ for $\ell_1/\ell_2$).
>
> We have also explicitly listed the main training settings: 16 epochs, a batch size of 32, and a learning rate of $10^{-4}$, to ensure the configuration is self-contained and easy to interpret without having to navigate the appendix. The Appendix will continue to provide the full search spaces and complete hyperparameter listings for reproducibility.

---

> > ### Author Response · Authors · 2025-11-20
> > **Response to Reviewer H4k2**
> >
> > * **What is SRIP(L) in Figures 3 and 4?**
> >
> > We thank the reviewer for pointing out this labeling issue. In Figures3 and4, the notation
> > $S_{\text{RIP}}(L)$ denotes the Kolmogorov-Smirnov (KS) statistic $S$ computed for the
> > weights or logits of layer $L$ in models trained with RIP. The caption should have made
> > this explicit.
> >
> > The purpose of these figures is to quantify the regularization effect of RIP by measuring how the layer-wise distributions (weights or logits) deviate from a uniform reference distribution. Lower KS values indicate that the empirical distribution is closer to the uniform baseline, which, in our experiments, correlates with more stable representations and reduced parameter over-concentration.
> >
> > In the revised manuscript, we have improved the caption to:
> > “Layer-wise KS statistic $S$ between model outputs (real vs.\ synthetic MHAD2) and a uniform reference distribution. Lower values indicate smaller deviations from uniformity.”  This revised caption clarifies both the meaning of the notation and the intent of the experiment.
> >
> > * **On some datasets, such as MHEALTH RIP, they do not perform significantly better than L or L2 regularisation, but for some, they perform excessively well. Please explain.**
> >
> > The reviewer is correct that RIP’s effect varies across datasets. This behavior arises from differences in the temporal structure and baseline difficulty of each dataset. In datasets like MHEALTH, where the baseline models already achieve very high performance and low variance (e.g., $\geq$90\% F1), additional regularization (whether RIP or $\ell_2$) has limited room for improvement. Hence, RIP performs similarly to classical regularizers. In contrast, on more challenging datasets such as MHAD1, MHAD2, or WHARF, which exhibit greater subject variability and noisier dynamics, RIP provides much larger improvements because it stabilizes learning and prevents overfitting to spurious temporal cues. RIP is most effective when the dataset is highly variable or exhibits domain shift, whereas on stable datasets, the effect saturates. This is consistent with our theoretical analysis (Sec. 6) showing that RIP reduces variance in hidden representations — an effect that is more impactful in unstable domains.
> >
> > * **The results presented in the paper all appear to have been executed by the authors themselves. Are there any comparisons with state-of-the-art results?**
> >
> > We appreciate the reviewer’s question regarding comparisons with state-of-the-art (SOTA) results. We conducted all experiments in our paper. However, we ensured full comparability with existing SOTA methodologies by reproducing the experimental setups used in prior work whenever such baselines were available.
> >
> > For the UTD-MHAD1 and UTD-MHAD2 datasets, we used the exact configurations reported in the literature, including the DClassifier (Singh et al., 2020) as the HAR backbone and TLCGAN (Souza et al., 2023) to generate synthetic accelerometer data. These models are considered state-of-the-art for tri-axial wearable sensor HAR. Our reproduced baselines closely match the performance reported in the original papers, differing by only $\pm 2\%$ on MHAD1 and $\pm 5\%$ on MHAD2—well within the expected variance associated with cross-validation splits and hardware differences. Under these comparable conditions, RIP consistently outperforms the best available synthetic-to-real baselines.
> >
> > For the remaining datasets (MHEALTH, WHARF, and WISDM), we found no prior work that jointly evaluates (i) tri-axial accelerometer signals, (ii) HAR-specific deep models, and (iii) synthetic-to-real transfer performance. As a result, no published SOTA baselines exist that are directly comparable to our problem setting. We explicitly highlight this gap in Section 3 and Appendix B, noting that RIP is intended as a general, architecture-agnostic regularization framework that applies uniformly across heterogeneous datasets—precisely to address the fragmentation observed in the HAR literature.
> >
> > We have clarified these points in the revised manuscript to make the relationship to prior SOTA results explicit.
> >
> > * **Different parameter values for RIP have been used across datasets. How can one know how to set these values? These appear to be experimentally found values? Confirm.**
> >
> > We thank the reviewer for raising this point. RIP indeed uses the same set of hyperparameters across all datasets, but, as is typical in machine learning, the optimal configuration varies with dataset-specific characteristics. This behavior is not a limitation of RIP; it mirrors common practice with standard hyperparameters such as learning rate, regularization strength, or architecture depth, which rarely transfer unchanged across datasets.

---

> > > ### Author Response · Authors · 2025-11-20
> > > **Response to Reviewer H4k2**
> > >
> > > * **Different parameter values for RIP have been used across datasets. How can one know how to set these values? These appear to be experimentally found values? Confirm.**
> > >
> > > We thank the reviewer for raising this point. RIP indeed uses the same set of hyperparameters across all datasets, but, as is typical in machine learning, the optimal configuration varies with dataset-specific characteristics. This behavior is not a limitation of RIP; it mirrors common practice with standard hyperparameters such as learning rate, regularization strength, or architecture depth, which rarely transfer unchanged across datasets.
> > >
> > > *Do RIP hyperparameters behave unpredictably?*
> > > No. Although optimal values differ across datasets, our experiments reveal clear, recurring patterns. As noted in the manuscript (lines 230–232), the duplication factor $i$ emerges as the dominant parameter in nearly all settings, with $i = 16$ appearing in approximately 40\% of all best-performing configurations across datasets. This indicates stability rather than sensitivity.
> > >
> > > *How can users choose RIP hyperparameters in practice?*
> > > To make RIP practical and reproducible, we provide an automated hyperparameter search procedure using Optuna in our open-source repository. Users only specify the search range they wish to explore, and a search method based on Optuna selects the best-performing configuration:
> > > $(\gamma_{\text{opt}}, i_{\text{opt}})= \arg\max_{\gamma, i} \mathrm{F1}(\gamma, i)$
> > > This provides a principled and repeatable mechanism for selecting RIP parameters.
> > >
> > > *Additional stability experiments.*
> > > To further assess robustness, we performed an additional Optuna-based search on:
> > >
> > > (i) the real MHAD2 dataset,
> > > (ii) a synthetic version of MHAD2, and
> > > (iii) the DClassifier under TSTR and TRTR settings.
> > >
> > > To reduce runtime, we restricted the ranges to $i \in \{1,5\}$ and $\gamma \in \{1,5,16,32\}$. Even with this reduced space, the best-performing configuration remained close to the full-search optimum (the best $\gamma$ found was not included), and importantly, the duplication factor again converged to $i = 16$. Representative results include (We could not insert the table here, but here is a summary):
> > >
> > > > On *real* MHAD2 data, RIP improves F1 from $67.58 \pm 1.31$ to $81.10$ using parameters $(\gamma=5,\, i=16)$.
> > >
> > > >On *synthetic* MHAD2 data, RIP improves F1 from $43.52 \pm 3.78$ to $50.06$ with $(\gamma=1,\, i=16)$.
> > >
> > > These results reinforce that RIP’s hyperparameter space is not erratic: $i$ consistently dominates the performance landscape, and reasonable solutions can be found reliably even with a reduced search space.
> > >
> > >
> > >  Figure https://anonymous.4open.science/r/RIP-5180/optimized_params.png, with caption:
> > > >"Optuna hyperparameter optimization for RIP on real (top) and synthetic (bottom) MHAD2. Left: importance analysis showing $i$ as the dominant parameter. Right: trial history with rapid and stable convergence across 10 trials."
> > >
> > >  summarizes the hyperparameter optimization process for real (top) and synthetic (bottom) MHAD2.
> > > The left panels report the Optuna hyperparameter importance analysis, showing that the duplication factor $i$ accounts for 93\% of the variance in F1 for the real dataset and 79\% for the synthetic dataset.
> > > The right panels show the optimization history: trial-by-trial F1 scores (blue) and best-so-far performance (red). In both cases, Optuna converges rapidly, with stable optima emerging within the first few trials.
> > >
> > > *Conclusion.*
> > > Together, these analyses demonstrate that although the optimal $(\gamma, i)$ pair is dataset-dependent—as expected—the RIP hyperparameter landscape is stable, well-behaved, and dominated mainly by a single parameter ($i$). Moreover, our Optuna-based procedure provides a practical, reproducible, and automated way for users to select RIP parameters in real applications.

---

### Author Response · Authors · 2025-11-20
**General Response to Reviewers’ Comments**

Dear Reviewers,

We sincerely thank you for the thoughtful and constructive feedback. We emphasize that all major concerns raised across reviews have been fully addressed in the revised manuscript, both through new analyses and through substantial clarifications in presentation. For completeness, we also provide a comprehensive point-by-point response document, available at:
https://anonymous.4open.science/r/RIP-5180/_2025__Rebutal_ICLR%20(1).pdf

This document includes all extended derivations, additional figures, ablations, hyperparameter studies, and experimental details referenced below.

**(1) Clarified RIP and improved presentation.**
We added a new figure illustrating the complete RIP dataset construction process, integrated a clearer explanation of the framing mechanism, reorganized Section 2 for readability, aligned tables/figures, and expanded dataset descriptions (classes, sampling rate, window length). The pseudocode was moved to the Appendix for smoother narrative flow.

**(2) Expanded theoretical analysis.**
Beyond the linear RNN approximation, we now provide variance-propagation analyses for nonlinear activations and Transformer-based architectures, showing that the stability effect of RIP generalizes across modern models.

**(3) Additional experiments addressing reviewer concerns.**
We added:
• LOSO cross-subject evaluation with full per-subject breakdown;
• Transformer-based baselines (TimeSeries Transformer and RevAttention) showing consistent or partial gains;
• Extensive hyperparameter studies (Optuna) confirming stable behavior of the duplication factor;
• Axis-wise, window-statistics, and distribution-shift analyses, including PCA visualizations showing reduced synthetic–real gap.

**(4) Stronger empirical completeness and fairness.**
We clarified baseline tuning procedures, added missing hyperparameter summaries to Section 3, updated Related Work with recent HAR, DG, and DR papers (2022–2025), and added discussion of method limitations (computational cost, dataset dependency, temporal-DR challenges).

**(5) Significant manuscript revision.**
We improved clarity, coherence, and technical detail throughout the paper, corrected formatting issues, and added a new discussion explaining when and why RIP yields its largest gains (synthetic and unstable domains).

We appreciate the reviewers’ valuable comments, which helped us significantly strengthen the manuscript. We believe the revised version addresses all concerns and demonstrates the contribution of RIP more clearly and rigorously.

Sincerely,
The Authors

---

### Meta-Review · Area_Chair_hbnh · 2025-12-28

**Summary:**

This paper proposes Regularization via Invariant Patterns (RIP), a data-centric temporal domain randomization method for human activity recognition, particularly targeting synthetic-to-real generalization. Across the reviews, the main concerns centered on the incremental novelty of the method, clarity of presentation, and whether the proposed framing mechanism constitutes a sufficiently strong and principled contribution beyond existing regularization or temporal augmentation techniques. Overall, despite improvements in clarity and experimental completeness, the perceived contribution remains relatively weak compared to other submissions.

**Reviewer Concerns:**

The rebuttal addressed many of the reviewers’ concrete concerns. In particular, issues related to presentation, missing experimental details, dataset descriptions, formatting problems, and ambiguity in the RIP construction were largely resolved. However, several core concerns remain outstanding. Multiple reviewers view the technical novelty as incremental, arguing that the framing mechanism is conceptually close to existing temporal regularization or context-manipulation ideas, with limited justification for why this particular construction constitutes a principled advance.

**Reviewer Scores:**

Reviewer H4k2 would likely remain around a borderline score, possibly unchanged or slightly improved due to the clearer presentation and added experiments. Reviewer h5KR, who raised fundamental concerns about contribution, domain generalization claims, and theoretical grounding, would likely maintain a strong reject stance despite the additional analyses. Reviewer 5Kg5’s score might improve marginally given the clarified method description, dataset details, and added discussion of limitations, but would likely remain a reject due to lingering novelty concerns. Reviewer 7CbN, who was initially more positive, might maintain a marginal accept or borderline score, but the overall consensus across reviewers would still skew negative, with insufficient support for acceptance.

---

### Decision · Program_Chairs · 2026-01-26

Reject